# 👭 FairMedFM: Fairness Benchmarking for Medical Imaging Foundation Models

**Ruinan Jin**[1,4*], **Zikang Xu**[2*], **Yuan Zhong**[3*], **Qingsong Yao**[2],
**Qi Dou**[3†], **S. Kevin Zhou**[2†], **Xiaoxiao Li**[1,4†]
[1]The University of British Columbia
[2]University of Science and Technology of China
[3]The Chinese University of Hong Kong
[4]Vector Institute

## Abstract

The advent of foundation models (FMs) in healthcare offers unprecedented opportunities to enhance medical diagnostics through automated classification and segmentation tasks. However, these models also raise significant concerns about their fairness, especially when applied to diverse and underrepresented populations in healthcare applications. Currently, there is a lack of comprehensive benchmarks, standardized pipelines, and easily adaptable libraries to evaluate and understand the fairness performance of FMs in medical imaging, leading to considerable challenges in formulating and implementing solutions that ensure equitable outcomes across diverse patient populations. To fill this gap, we introduce `FairMedFM`, a fairness benchmark for FM research in medical imaging. `FairMedFM` integrates with 17 popular medical imaging datasets, encompassing different modalities, dimensionalities, and sensitive attributes. It explores 20 widely used FMs, with various usages such as zero-shot learning, linear probing, parameter-efficient fine-tuning, and prompting in various downstream tasks – classification and segmentation. Our exhaustive analysis evaluates the fairness performance over different evaluation metrics from multiple perspectives, revealing the existence of bias, varied utility-fairness trade-offs on different FMs, consistent disparities on the same datasets regardless FMs, and limited effectiveness of existing unfairness mitigation methods. Checkout `FairMedFM`'s project page and open-sourced codebase, which supports extendible functionalities and applications as well as inclusive for studies on FMs in medical imaging over the long term.

## 1 Introduction

Foundation Model (FM) facilitated medical image analysis is playing a pivotal role in healthcare [2, 3]. These models, which leverage large-scale pretraining and fine-tuning [6], have demonstrated remarkable capabilities in various medical imaging tasks, including classification [69, 41] and segmentation [55, 39]. As the use of FMs proliferates in medical imaging, addressing the challenges of evaluating and ensuring their fairness and utility becomes increasingly critical [28, 24, 77], where biases in model performance can result in significant disparities in patient care and outcomes.

Creating benchmarks for algorithm fairness in medical imaging can lead to consistent experiment settings and ensure standardization. There are efforts to benchmark fairness algorithms in non-FM-based traditional machine learning for medical imaging [77, 50, 23, 76, 13, 72]. However, the fairness

---

[*] Equal contribution. Authors are listed in alphabetical order.
[†] Co-corresponding authors.

38th Conference on Neural Information Processing Systems (NeurIPS 2024) Track on Datasets and Benchmarks.

Table 1: Comparisons between `FairMedFM` and other medical imaging fairness literature and benchmarks.

| Category | Subcategory | Items | FairMedFM (ours) | Khan et al. [28] | MedFA-IR [77] | Iurada et al. [23] | RadFusion et al. [76] | CXR-Fairness [72] | Fair-Tune[13] |
|---|---|---|---|---|---|---|---|---|---|
| **Models** | | Study includes FM | ✔ | ✔ | | | | | |
| **Foundation Models**[1] Sec. 3.2 | Domains | General-purpose | ✔ | ✔ | | | | | |
| | | Medical-specific | ✔ | ✔ | | | | | |
| | Types | Vision Models | ✔ | ✔ | | | | | |
| | | Vision-language Models | ✔ | | | | | | |
| **Functionalities** Sec. 2 and 3 | Tasks | Classification | ✔ | ✔ | ✔ | ✔ | ✔ | ✔ | ✔ |
| | | Segmentation | ✔ | | | | | | |
| | Usages | Zero-shot | ✔ | | | | | | |
| | | Linear Probing | ✔ | ✔ | | | | | |
| | | CLIP-Adaptation | ✔ | | | | | | |
| | | Prompt-based Segmentation | ✔ | | | | | | |
| | | Parameter-efficient Fine-tuning | ✔ | | | | | | ✔ |
| | | Full Training [2] | ✔ | ✔ | ✔ | ✔ | ✔ | ✔ | ✔ |
| | Debias Algorithms | Group Rebalancing | ✔ | | ✔ | | | | |
| | | Adversarial Training | ✔ | | ✔ | ✔ | | ✔ | |
| | | Fairness Constraint | ✔ | ✔ | ✔ | | ✔ | | |
| | | Subgroup-tailored Modeling | ✔ | | ✔ | | | | |
| | | Domain Generalization | ✔ | | ✔ | ✔ | | ✔ | |
| **Data** Sec. 3.1 | Dimensions | 2D | ✔ | ✔ | ✔ | ✔ | | ✔ | ✔ |
| | | 2.5D | ✔ | | | | | | |
| | | 3D | ✔ | | ✔ | | ✔ | | ✔ |
| | Modalities | X-ray | ✔ | ✔ | ✔ | ✔ | ✔ | ✔ | ✔ |
| | | CT | ✔ | | ✔ | | | | ✔ |
| | | MRI | ✔ | | ✔ | | | | |
| | | Ultrasound | ✔ | | | | | | |
| | | Fundus | ✔ | | ✔ | | | | ✔ |
| | | OCT | ✔ | | ✔ | | | | ✔ |
| | | Dermatology | ✔ | | ✔ | ✔ | | | ✔ |
| | Sensitive Attributes | Sex | ✔ | ✔ | ✔ | ✔ | ✔ | ✔ | ✔ |
| | | Age | ✔ | | ✔ | | ✔ | ✔ | ✔ |
| | | Race | ✔ | ✔ | ✔ | | ✔ | ✔ | ✔ |
| | | Preferred language | ✔ | | | | | | |
| | | Skin tone | ✔ | | | ✔ | ✔ | | ✔ |
| | | Marital states | ✔ | | | | | | |
| | | Handedness | ✔ | | | | | | |
| | | BMI | ✔ | | | | | | |
| **Evaluation Metrics Taxnonmy** Sec. 3.4 | | Utility | ✔ | ✔ | ✔ | ✔ | ✔ | ✔ | ✔ |
| | | Outcome-consistency Fairness | ✔ | ✔ | ✔ | ✔ | ✔ | ✔ | ✔ |
| | | Predictive-alignment Fairness | ✔ | | ✔ | | | ✔ | |
| | | Fairness-utility Tradeoff | ✔ | | | | | | |
| | | Positive-parity Fairness [2] | ✔ | | ✔ | | ✔ | ✔ | ✔ |
| | | Representation Fairness [2] | ✔ | | | | | | |
| | | Statistics Test [2] | ✔ | | ✔ | | | | |

[1] Only studies that involve FMs are ticked in this category.
[2] Results presented in the Appendix.

of modern FMs differs due to their extensive pre-training on diverse and often large-scale datasets. The varied nature of general-purpose and medical-specific FMs, as well as their application to medical imaging downstream tasks, introduces unique fairness challenges. A growing body of literature has begun to explore various aspects of fairness in FMs for medical imaging, including developing bias mitigation strategies [24, 67], and fairness evaluation [28]. However, these studies are often limited in scope, e.g., focusing on a single category of FMs, data modality, or tasks.

**Why is our benchmark needed?** *First*, no existing literature nor framework provides standardized pipeline to investigate fairness on comprehensive FMs (domains and types), comprehensive functionalities (tasks, applications, and debiasing algorithms), comprehensive data (dimensions, organs, modalities, and sensitive attributes (SA)), and comprehensive evaluation aspects in medical imaging, as shown in Tab. 1. *Second*, insufficient understanding of the fairness issues and utility trade-offs associated with the development and deployment of FMs for medical imaging persists due to a lack of comprehensive analysis based on extensive experimentation. *Lastly*, there is a pressing need for a versatile fairness evaluation codebase that is easily extensible to essential segmentation tasks and adaptable to FMs for various uses in medical imaging. Existing libraries, though acknowledged by the fair machine learning community [5, 4, 77], do not adequately fulfill these requirements.

To fill these gaps, we propose the first comprehensive pipeline, `FairMedFM`, along with benchmarking observations for the fairness of FMs in medical imaging. Our contribution mainly includes the following two folds:

1. We offer a comprehensive evaluation pipeline covering 17 diverse medical imaging datasets, 20 FMs, and their usages (see Tab. 1). This benchmark addresses the need for a consistent evaluation and standardized process to investigate FMs' fairness in medical imaging.

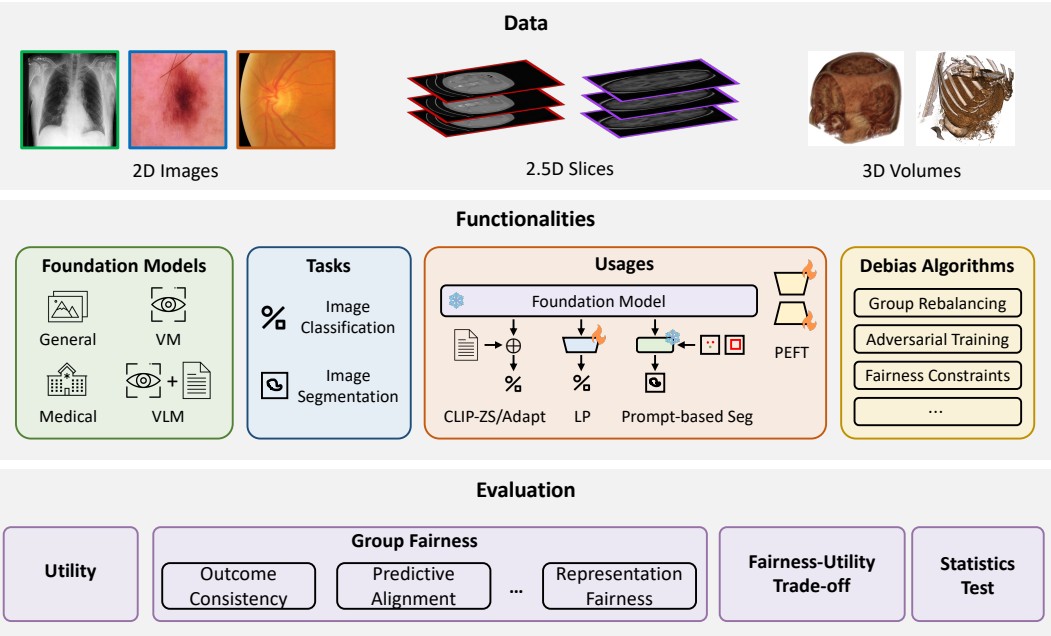

Figure 1: Overview of the `FairMedFM` framework, a standardized pipeline to investigate fairness on diverse datasets (2D, 2.5D, and 3D), comprehensive functionalities (various FMs, tasks, usages, and debias algorithms), thorough evaluation metrics. The details are explained in Sec. 3.

2. With `FairMedFM`, we conducted a thorough analysis from various perspectives, where we found that (1) Bias is prevalent in using FMs for medical imaging tasks, and the fairness-utility trade-off in these tasks is influenced not only by the choice of FMs but also by how they are used; (2) There is significant dataset-aware disparities between SA groups in most FMs; (3) Consistent disparities in SA occur across various FMs on the same dataset; and (4) Existing bias mitigation strategies do not demonstrate strong effectiveness in FM parameter-efficient fine-tuning scenarios.

3. We open-source `FairMedFM`, an extensible implementation for launching the FMs for medical image analysis, to prompt the study of FMs for medical imaging and fairness evaluation in the community.

The scope of our work is to establish a more comprehensive benchmark for medical imaging, focusing on classification and segmentation tasks, binarized SAs and commonly used FM strategies from the literature. Our objective is to raise awareness of fairness issues within the medical imaging community and assist in developing fair algorithms in the machine learning community, by promoting more accessible and reproducible methods for fairness evaluation.

## 2 Preliminaries on Foundation Models, Medical Imaging, and Fairness

### 2.1 FMs in Medical Imaging

FMs have recently garnered widespread interest due to their powerful generalization capabilities. These large models are designed to learn from large-scale unsupervised data. In medical applications, FMs are particularly valuable because massive amounts of unlabeled medical data are easier to obtain than labeled data, which requires costly expert annotations. Typically, FMs are pre-trained on broad datasets to acquire medical knowledge using two primary training objectives: recovering masked words or vision patches [66], and aligning features of paired text and images through contrastive learning [49]. Their pre-trained representations can be successfully applied to various downstream medical tasks with minimal or no reliance on expert labels. In this paper, we focus primarily on classification and segmentation tasks.

**Classification FMs in Medical Imaging.** Classificaiton FMs vary in architecture, which can be roughly categorized into two groups. The first one is *vision models (VMs)*, which takes images as input and learns a general representation across diverse datasets by different self-supervised proxy tasks. MedMAE [66] and MedLVM [43] are trained by masked imaging modeling [17] and graph matching, respectively; DINOv2 [45], MoCo-CXR [59], and C2L [75] decrease the feature disparities of the paired and augmented vision patches. Another group is *vision-language models (VLMs)* (e.g., CLIP [16], MedCLIP [65], PubMedCLIP [15], BiomedCLIP [74]), which are trained to attract the feature of paired text and images, while BLIP [33] design a $q$-former for vision-text alignment.

In classification, we follow the common workflow [8] to consider $D = (\mathcal{X}, \mathcal{Y}, \mathcal{A})$ to be a set of distributions where we have input $x$ in space $\mathcal{X}$, the disease label $y \in \{0, 1\}^C$ in space $\mathcal{Y}$, and sensitive attributes $a$ in space $\mathcal{A}$. Let $f_v(\cdot) : \mathcal{X} \to v \in \mathbb{R}^k$ denote the vision encoder of a foundation model which embeds the inputs into feature $v$ with dimension $k$.

**Segmentation FMs in Medical Imaging.** Following the trend of the large model, the Segment Anything Model (SAM) [30] shows great potential in segmentation task, which is trained on 1B labeled natural data and offers the zero-shot ability to generate segmentations mask only based on point or box prompts as input. In terms of both data structure and context, medical images exhibit significant differences from natural images. First, medical images vary in modalities, including 2D grayscale images (X-ray), 2D RGB images (dermatology), and 3D volumes (CT). Especially for 3D images, MedSAM [38] processes 3D volumes with slice-wise operations, while SAM-Med3D [63] is a 3D model trained on massive labeled 3D medical volumes. Furthermore, considering the domain gap between medical and natural images, fine-tuning is necessary for applying Segmentation FMs (SegFMs) (pre-trained on natural images like SAM [30]) in medical to learn medical context. Similar to classification, we consider $D = (\mathcal{X}, \mathcal{S}, \mathcal{A})$, where we have the segmentation mask $s \in \{\mathbb{R}^{w \times h \times d}\}^C$ in label space $\mathcal{S}$.

## 2.2 Fairness in Medical Imaging

**Defintion of Fairness.** Fairness, a critical aspect of AI ethics, urging that deep learning models should not have skewed outcomes towards personalities with diverse demographics, has been widely studied in computer vision [26] and natural language processing [35]. Among various fairness definitions, *group fairness* is the most common one, which ensures that the model's performance is consistent across different groups. Let $Y, \hat{Y}$ be the ground truth label and prediction of the model, respectively, and $A \in \{0, 1\}$ be the SA. One of the group fairness metrics, Accuracy Parity, is defined as $P(\hat{Y} = Y | A = 0) = P(\hat{Y} = Y | A = 1)$.

**Fairness in Medical Imaging.** Fairness issues *do* exist in deep learning-based medical image analysis, especially for the marginalized populations. For example, Seyyed-Kalantari *et al.* [52] find that their Chest X-ray classifier has a higher underdiagnosis rate for Hispanic female patients. Similar phenomenons also occur in other medical tasks, including regression [47], segmentation [48], reconstruction [11], etc.

## 2.3 FMs Meet Fairness in Medical Imaging

Previous studies have shown that unfairness can be induced to FMs from the pre-training datasets, the fine-tuning process, the application of downstream tasks [35]. However, most of the current studies on fairness in medical imaging mainly focus on the traditional models, while evaluating and mitigating unfairness in FMs is still in its infancy. Khan *et al.* [28] benchmarked six classification FMs on sex and race. However, the fairness metrics and datasets involved in their study are limited. Therefore, extra efforts is required to evaluate whether these FMs in medical imaging have fair outcomes before applying them in real clinical scenarios.

# 3 FairMedFM

Fig. 1 presents the pipeline of `FairMedFM` framework, which offers an easy-to-use codebase for benchmarking the fairness of FMs in medical imaging. `FairMedFM` contains 17 datasets (9 for classification and 8 for segmentation) and 20 FMs (11 for classification and 9 for segmentation). It also integrates 9 fairness metrics (5 for classification and 4 for segmentation) and 6 unfairness

mitigation algorithms (3 for classification and 3 for segmentation), trying to provide a relatively comprehensive benchmark for fairness in medical imaging FMs.

## 3.1 Datasets

The following 17 publicly avaliable datasets are included in `FairMedFM` to evaluate fairness in FMs in medical imaging: CheXpert [22], MIMIC-CXR [25], HAM10000 (CLS) [61], FairVLMed10k [36], GF3300 [37], PAPILA [31], BRSET [42], HAM10000 (SEG) [61], TUSC [53], FairSeg [60], Montgomery County X-ray [7], KiTS 2023 [18], CANDI [27], IRCADb [58], SPIDER [62]. These datasets vary in the following aspects: (1) *Task type:* classification and segmentation; (2) *Dimension:* 2D and 3D; (3) *Modality:* OCT, X-ray, CT, MRI, Ultrasound, Fundus, OCT, and dermatology; (4) *Body part:* brain, eyes, skin, thyroid, chest, liver, kidney, and spine; (5) *Number of classes:* ranging from 2 to 15; (6) *Number of samples:* ranging from 20 to more than 350k; (7) *Sensitive attribute:* sex, age, race, preferred language, skin tone, etc. (8) *SA skewness (Male : Female):* ranging from 0.19 to 1.67. The details of these datasets can be found in the Appendix B.

## 3.2 Models

**Classification FMs.** Eleven FMs from two categories are used for evaluation: (1) *vision models (VMs)*: C2L [75], DINOv2 [45], MedLVM [43], MedMAE [66], MoCo-CXR [59]; (2) *vision-language models (VLMs)*: CLIP [16], BLIP [34], BLIP2 [33], MedCLIP [65], PubMed-CLIP [15], BiomedCLIP [74]. For all models, *LP* is used for fine-tuning; for VLMs, we also conduct *CLIP-ZS* and *CLIP-Adapt*. Since these FMs are 2D models, we use 2.5D slices for 3D data and report volume-wise results.

`FairMedFM` evaluate the fairness of FMs under three commonly used protocols for classification:

• *Linear probing (LP).* A classification head $h(\cdot) : v \to \mathcal{Y}$ is trained to map the FM's embedding $v$ to the prediction $\hat{y} = h(f_v(x))$.

• *Parameter-efficient fine-tuning (PEFT).* PEFT aims to fine-tune FMs with a classification head to new downstream tasks with minimal computational overhead. `FairMedFM` evalute fairness of FMs in the fine-tuning setting with modern PEFT strategies (e.g., LoRA [20] and pruning).

• *CLIP-ZS and CLIP-Adapt.* Vision-language models offer the zero-shot classification ability for FMs, which compares the similarities of the vision embedding $f_v(x)$ between different class-wise prototypes text embeddings $f_t(x)$ of positive and negative prompts (e.g., "There is no pneumonia.") for each class. We consider both zero-shot *(CLIP-ZS)* inference as well as a simple adaptation strategy *(CLIP-Adapt)* [57] which fine-tunes the class prototypes initialized with CLIP zero-shot prototypes.

**Segmentation FMs.** Nine SegFMs are selected from three categories for evaluation: (1) *general-SegFMs:* SAM [30], MobileSAM [71], TinySAM [56]; (2) *2D Med-SegFMs:* MedSAM [38], SAM-Med2D [9], and FT-SAM [9]; (3) *3D Med-SegFMs:* SAM-Med3D [63], FastSAM3D [54]. All the four segmentation prompts described in Sec. 2.1 are used for 2D SegFMs. we adopt *rand* and *rands* for SAM-Med3D and FastSAM, *rands* and *bbox* for SegVol following their official implementation.

For SAM-family models, extra point prompts $p_{\text{point}}$ or bounding box prompt $p_{\text{bbox}}$ are required in the inference stage. Therefore, `FairMedFM` examine the fairness of SegFMs with different types of prompts including (1) *center:* the center point of the mask; (2) *rand:* 1 random point inside the mask; (3) *rands:* 5 random points inside the mask; (4) *bbox:* the bounding box of the mask. These prompts can be generated either directly from the ground truth mask or manually annotated. To thoroughly evaluate the fairness of SegFMs, `FairMedFM` provides the interface for both 2D and 3D SAM, and the access to fine-tune the SAM on the specific medical dataset with full supervision.

## 3.3 Unfairness Mitigation Methods

`FairMedFM` provides popular and generalizable bias mitigation strategies and integrates them with the FMs. Following literature specialized in bias mitigation algorithms [77], we categorize them into the following:

(1) *Group rebalancing* is a technique used to address bias in datasets by adjusting the representation of different subgroups [21, 48], ensuring that minority groups have equal representation during training. This helps to mitigate biases that can arise from imbalanced datasets. (2) *Adversarial training* is a method for reducing bias by training models in a way that they learn to make predictions while simultaneously being penalized for recognizing SA [70, 29]. This promotes fairness by minimizing the influence of biased features. (3) *Fairness constraints* are used to ensure that models are trained to produce fair outcomes for different subgroups. This approach involves adding the differentiable form of fairness metrics to the training objective directly [44], or adjusting weights of the loss function for different subgroups to penalize the model for making biased predictions [60]. (4) *Subgroup-tailored modeling* is a method that allows subgroups to have different model parameters, enabling the model to learn different representations for subgroups. This specialized modification can be applied on part of the model, i.e. fairness-aware adaptors [68], or the entire model [64]. (5) *Domain generalization* aims to improve a model's ability to perform well across various domains, including those not encountered during training [64]. This approach seeks to create models that generalize better by finding robust solutions that work well in different scenarios [51].

### 3.4 Evaluation Metrics Taxnonmy

**Utility** refers to the effectiveness of the model in making accurate predictions. Examples include the **Area Under the receiver operating characteristic Curve (AUC)** for classification and **Dice similarity score (DSC)** for segmentation.

**Group fairness** is evaluated from four aspects following common practice [8]: (1) *Outcome-consistency fairness*, which evaluates discrepancies in the model's performance (e.g., accuracy, components of confusion matrix, etc.) between different sensitive groups. In classification, we include **delta AUC ($AUC_\Delta$)**, which is measured as the maximum AUC gap among subgroups; and **Equalized Odds (EqOdds)**, which measures the differences in true positive and false positive rates between advantaged and disadvantaged groups. In segmentation, we include **delta DSC ($DSC_\Delta$)**, which assesses if both groups receive approximately equal predictive performance; and **DSC skewness ($DSC_{Skew}$)**, which measures the degree of skewness of DSC between advantaged and disadvantaged groups. (2) *Predictive alignment fairness*, which focuses on the alignment between predicted probabilities and actual outcomes. It ensures that predicted scores accurately reflect true likelihoods, providing a reliable basis for decision-making across different groups. We report the **expected calibration error gap ($ECE_\Delta$)** in classification, where a high value indicates an optimal decision threshold; (3) *Positive-parity fairness*, which ensures that the positive classification rate is equal for both unprivileged and privileged groups, preventing any group from being overlooked. We note that this is an optional evaluative aspect that may not be applicable to all scenarios. For example, positive parity is compromised in diagnosing glaucoma, where morbidity rates differ between males and females; (4) *Representation fairness*, which evaluates fairness from the aspects of feature representation learned in the latent space, by estimating either group-wise feature separability among subgroups. We report the last two metrics in the Appendix E.2.

**Utility-fairness trade-off** takes both utility and fairness into account. It can be evaluated by combining utility and fairness metrics. Besides, equity scaling measurements that involve both aspects could also be used. In classification, we measure the **equity-scaled AUC ($AUC_{ES}$)**, which takes both utility and fairness into account. In segmentation, we report the **equity-scaled DSC ($DSC_{ES}$)**, which measures the tradeoff between overall utility and utility variations [60].

The mathematical definitions of the above metrics are presented in the Appendix C.

## 4 Results

In this section, we highlight the representative observations and takeaways from benchmarking the fairness of FMs on image classification and segmentation tasks utilizing `FairMedFM` framework. We choose to present the fairness results concerning sex since it is the most common SA shared across datasets. However, our method supports a broader range of SAs as listed in the Tab. 1. *We direct readers to Appendix E.1 for additional results on more SAs and extensive evaluations, which corroborate the observations discussed in the main text.* We also present the evaluation for *positive-parity-fairness*, *representation-fairness*, and *statistics test* in Appendix E.2.

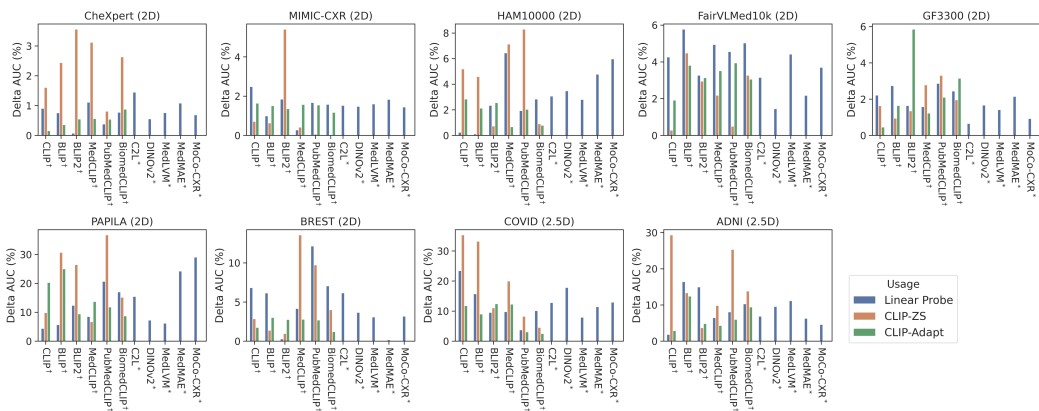

Figure 2: Bias in classification tasks. $AUC_\Delta$ is the fairness evaluation metric. "†" denotes vision-language models, and "∗" denotes pure vision models, where CLIP-ZS and CLIP-Adapt are not applicable.

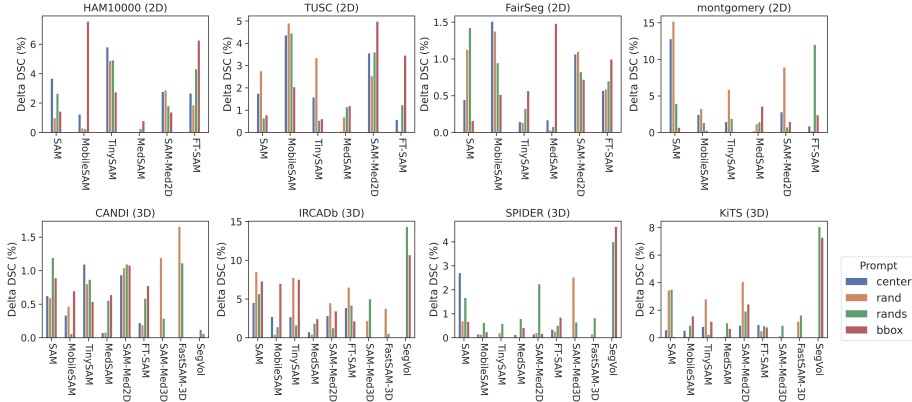

Figure 3: Bias in segmentation tasks. $DSC_\Delta$ is the fairness evaluation metric. Note that SAM-Med3D, FastSAM-3D, and SegVol are only applicable for 3D datasets.

**Bias widely exists in FMs for medical imaging**. Fig. 2 and Fig. 3 present the fairness metrics on various classification and segmentation tasks as stated in Sec. 1. In classification, the $AUC_\Delta$ has an average value larger than 5% over all methods but presents large variations across methods. Similarly, in segmentation, our results in both 2D and 3D datasets reveal similar disparities, with notable high $DSC_\Delta$ of SegVol and FT-SAM reaching up to 10%.

**Careful selection and use of FMs are needed for ensuring a good fairness-utility trade-off.** These pervasive biases challenge the fairness-utility tradeoff of FMs in medical applications. We observe that the fairness-utility trade-off in these tasks is influenced not only by the choice of FMs but also by how they are used as shown in Fig. 4. We report trade-off scores $AUC_{ES}$ and $DSC_{ES}$, for each datasets on the selected models for both tasks respectively. Compared with CLIP-ZS models, a simple adaptation, CLIP-Adapt, has proven effective in significantly boosting the fairness-utility trade-off in medical applications. Further, we evaluate the effects of segmentation tasks on the choice of prompts (including *center*, *rand*, *rands*, and *bbox*), and the types of SegFMs (including *2D General-SegFMs* and *2D Med-SegFMs*). Compared to *center* and *rand*, models using *rands* and *bbox* tend to be fairer across different datasets. This might be due to the tighter constraints on the segmentation provided by *rands* and *bbox*. Regarding models, General-SegFMs achieve a better fairness-utility trade-off than Med-SegFMs.

**Consistent disparities in SA occur across various FMs on the same dataset.** The performance of FMs shows dataset-specific biases, favoring one category of the given SA over the other, depending on the dataset. We present the four datasets in classification tasks and use sex as an example, presented

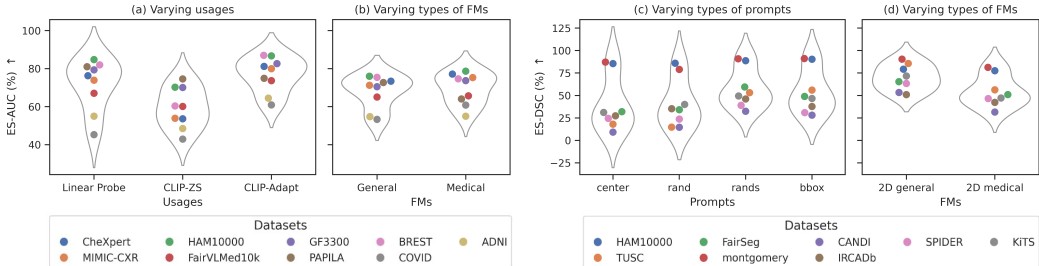

Figure 4: Fairness-utility tradeoff in FMs for different components on (a-b) classification and (c-d) segmentation tasks. We use $AUC_{ES}$ and $DSC_{ES}$ as evaluation metrics. (a) Fairness-utility tradeoff for different classification usages (using CLIP as an example); (b) Fairness-utility tradeoff for general-purpose and medical-specific FMs in classification; (c) Fairness-utility tradeoff for different prompts in segmentation (using SAM-Med2D as an example); (d) Degree of fairness for different SegFM categories in segmentation.

in Fig. 5. The BREST and MIMIC-CXR dataset shows a higher performance for females compared to males across various models. Conversely, the FairVLMed10k and GF3300 dataset indicates better performance for males. Results on other SAs and segmentation tasks can be found in Appendix E.1.

**Existing unfairness mitigation strategies are not always effective.** While various unfairness mitigation methods for traditional neural networks have been proposed, their effectiveness on FMs for medical imaging remains underexplored. This study evaluates three mitigation algorithms for both classification and segmentation tasks. For classification, we apply two PEFT strategies (LP and LoRA) on three datasets (MIMIC-CXR, HAM10000 (CLS), and FairVLMed10k), while experiments for segmentation are conducted on the HAM10000 (SEG) dataset following the SAMed pipeline [73]. As shown in Tab. 2, although some mitigation strategies show better fairness metrics compared to the baseline, their utility-fairness tradeoffs do not always exceed (for example, LoRA + GroupDRO vs. LoRA). Besides, some mitigation algorithms show both lower utility and worse fairness (SAMed + InD vs. SAMed), which means that existing unfairness mitigation strategies are not always effective for FMs. Potential reasons could come from the scaling gap in training data and model parameters between the 'small models' and large-scale FMs. Although there have been studies that focus on unfairness mitigation for FMs [24, 67], extra efforts are required to guarantee the fairness of FMs.

## 5 Conclusion

In this work, we introduced `FairMedFM`, a pioneering benchmark aimed at comprehensively evaluating the fairness of FMs in healthcare. Our pipeline demonstrates versatility by supporting various

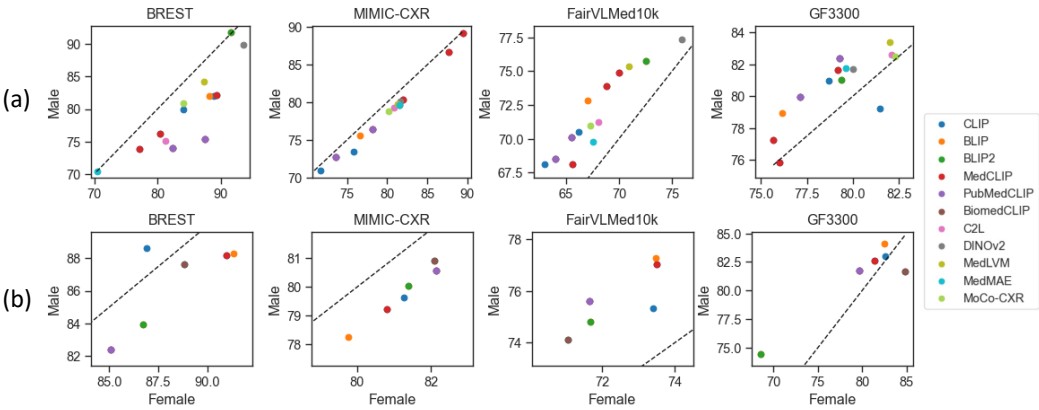

Figure 5: Consistent disparities in SA occur across various FMs on the same dataset. (a) LP; (b) CLIP-Adapt. Points on the black dashline represent equal utility.

Table 2: Evaluation of bias mitigation methods. Best and second best results are highlighted. All results are in percentage. Results for classification (CLS) are averaged across MIMIC-CXR, HAM10000 (CLS), and FairVLMed10k; Results for segmentation (SEG) are computed on HAM10000 (SEG).

| Category | Method | $AUC_{Avg} \uparrow$ | $AUC_{Female} \uparrow$ | $AUC_{Male} \uparrow$ | $AUC_{\Delta} \downarrow$ | $EqOdds \uparrow$ | $ECE_{\Delta} \downarrow$ | $AUC_{ES} \uparrow$ |
|---|---|---|---|---|---|---|---|---|
| CLS | LP | 76.07 | 75.51 | 76.04 | 0.53 | 96.08 | 0.75 | 75.87 |
| | LP + GroupDRO [51] | 75.70 | 75.15 | 75.68 | 0.54 | 95.93 | 0.83 | 75.50 |
| | LP + Resampling [77] | 75.66 | 76.01 | 75.45 | 0.56 | 94.65 | 1.78 | 75.45 |
| | LP + LAFTR [40] | 75.80 | 76.15 | 75.69 | 0.46 | 95.13 | 0.93 | 75.63 |
| | LoRA | 82.52 | 83.25 | 81.30 | 1.95 | 96.18 | 0.35 | 81.72 |
| | LoRA + GroupDRO [51] | 79.54 | 79.95 | 78.73 | 1.22 | 96.74 | 0.17 | 79.06 |
| | LoRA + Resampling [77] | 83.27 | 84.32 | 82.46 | 1.86 | 95.65 | 1.01 | 83.04 |
| | LoRA + LAFTR [40] | 80.50 | 80.08 | 80.70 | 0.62 | 97.87 | 0.22 | 80.25 |

| Category | Method | $DSC_{Avg} \uparrow$ | $DSC_{Female} \uparrow$ | $DSC_{Male} \uparrow$ | $DSC_{\Delta} \downarrow$ | $DSC_{STD} \downarrow$ | $DSC_{Skew} \downarrow$ | $DSC_{ES} \uparrow$ |
|---|---|---|---|---|---|---|---|---|
| SEG | SAM (best) | 83.55 | 84.47 | 82.74 | 1.73 | 0.86 | 1.11 | 82.83 |
| | SAMed [73] | 90.92 | 90.98 | 90.87 | 0.10 | 0.05 | 1.01 | 90.87 |
| | SAMed + FEBS [60] | 91.63 | 91.64 | 91.62 | 0.02 | 0.01 | 1.00 | 91.62 |
| | SAMed + Resampling [77] | 91.51 | 91.60 | 91.43 | 0.17 | 0.09 | 1.02 | 91.43 |
| | SAMed + InD [64] | 88.85 | 89.61 | 88.19 | 1.42 | 0.71 | 1.14 | 88.22 |

tasks, such as classification and segmentation, and by adapting to 20 different FMs, including both general-purpose and medical-specific models. By integrating a wide range of functionalities, such as LP, CLIP-Adapt, prompt-based segmentation, PEFT, and bias mitigation strategies, our framework enables comprehensive evaluations that are essential for developing fair and effective medical imaging solutions. With `FairMedFM`, we conducted in-depth analysis and revealed four key findings and takeaways: (1) Bias widely exists in FMs for medical imgaing tasks; (2) Different FMs and their variant usages present different fairness-utility trade-offs, therefore careful selection and proper use of FMs are crucial for ensuring a good fairness-utility trade-off; (3) The performance of various FMs exhibits consistent dataset-specific biases, which aligns with the SA distribution in individual datasets; and (4) The existing unfairness mitigation strategies are not always effective in FM settings.

**Future development plan.** Despite our efforts to include a wide range of datasets, FM methods, and fairness algorithms in our work to greatly enhance the comprehensiveness of existing benchmarks, there remains potential for further refinement and expansion. Our future work will continue to improve the comprehensiveness of our benchmark based on the framework and codebase of `FairMedFM`. The current pipeline `FairMedFM` is sufficiently flexible to extend; therefore, we will continue to incorporate new medical imaging datasets and emerging FM architectures over time. In addition to medical image classification and segmentation, the scope of our study will encompass predictive modeling, object detection, and vision-based question answering. Moreover, we intend to incorporate a broader range of fairness definitions in our evaluations and to investigate a wider array of bias-mitigation algorithms. We will ensure that `FairMedFM`'s open-sourced codebase remains actively maintained and updated at the forefront of promoting equitable healthcare technologies.

# Acknowledgments and Disclosure of Funding

Y. Zhong and Q. Dou are supported by the Research Grants Council of Hong Kong Special Administrative Region, China (Project No. T45-401/22-N). Z. Xu, O. Yao, and S.K. Zhouare supported in part by the Natural Science Foundation of China under Grant 62271465, SuzhouBasic Research Program under Grant SYG202338, and Open Fund Project of Guangdong Academy of Medical Sciences, China (No. YKY-KF202206). R. Jin and X. Li are supported by the UBC Advanced Research Computing and Digital Research Alliance of Canada. We thank the reviewers for their insightful feedback and comments.

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

# A    Related Work

## A.1    Fairnesss in Medical Imaging

Bias widely exists in deep learning-based medical image analysis and has been studied by several recent studies. In the FM domain, [67] proposed to add perturbations with adversarial training on the latent embedding space to mitigate bias in segmentation while [24] adds the debiased edit on the input image to mitigate the bias when FM's gradient is inaccessible. In the non-FM deep learning tasks, [10] investigates the trade-off among the fairness, privacy and utility in medical data; [32] uses diffusion model to generate the synthetic images and argument the training data for mitigating the bias; [77] benchmarks the commonly used bias mitigation strategies on medical image classifications.

However, with the quick development of FM in medical image analysis, FM-based diagnostics become more and more popular. In the fairness domain, except the very recent study tastes the bias mitigation strategies in them [67, 24], few literatures provide a comprehensive overview of FMs in medical image analysis in the perspective of fairness. This gap motivates us to create the comprehensive benchmark, `FairMedFM`, which offers an evaluation pipeline covering 17 diverse medical imaging datasets, 20 FMs, and their usages. This benchmark addresses the need for a consistent evaluation and standardized process to investigate FMs' fairness in medical imaging. To restate, our objective is to raise awareness of fairness issues within the medical imaging community and assist in developing fair algorithms in the machine learning community, by promoting more accessible and reproducible methods for fairness evaluation.

## A.2    Foundation Models

`FairMedFM` focuses on benchmarking the FMs for classification and segmentation as details in A.2.1 and A.2.2. Fig. 6 visualized the usages of FMs in this study.

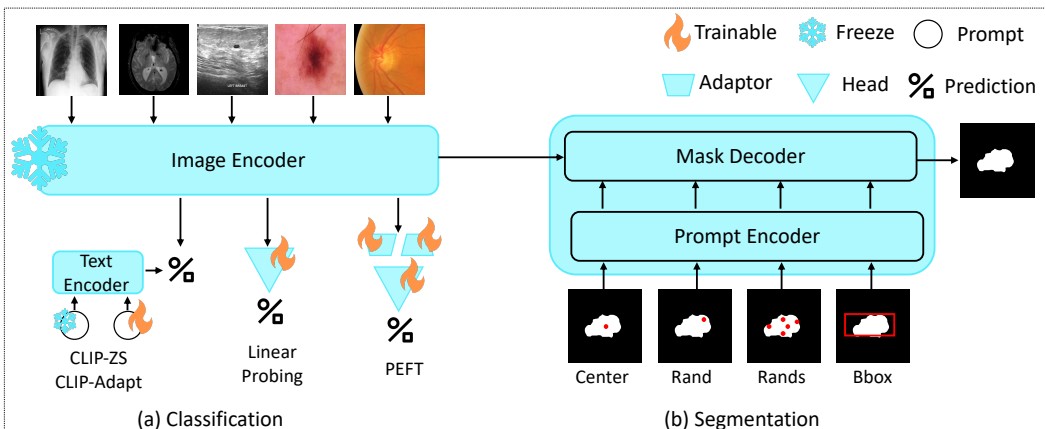

Figure 6: (a) Usages for classification, where the embedding of FM's image encoder is applied for CLIP-ZS, CLIP-Adapt, LP and PEFT; (b) Usages of SAM-family for segmentation, where the embedding of the image is passed to the mask decoder for generating the segmentation mask.

### A.2.1 Classification

In classification, `FairMedFM` implements both the VM and the VLM ranging from general-purpose and medical-specific FMs as detailed in Tab. 3. The usages of FMs for classification are outlined in Fig. 6, where CLIP-ZS, CLIP-Adapt, LP, and PEFT are included in `FairMedFM` as shown in Fig. 6 and detailed in Sec. 3.2 in the main paper.

Table 3: FMs used in classification.

| Category | Domain | Model | Link |
|---|---|---|---|
| VM | General | DINOv2 [45] | https://github.com/facebookresearch/dinov2 |
| | Medical | LVM-Med [43]
MedMAE [66]
MoCo-CXR [59]
C2L [75] | https://github.com/duyhominhnguyen/LVM-Med
https://github.com/lambert-x/medical_mae
https://github.com/stanfordmlgroup/MoCo-CXR
https://github.com/funnyzhou/C2L_MICCAI2020 |
| VLM | General | BLIP [34]
BLIP2 [33]
CLIP [49] | https://github.com/salesforce/BLIP
https://github.com/salesforce/LAVIS/tree/main/projects/blip2
https://github.com/openai/CLIP |
| | Medical | MedCLIP [65]
PubMedCLIP [15]
BiomedCLIP [74] | https://github.com/RyanWangZf/MedCLIP
https://github.com/sarahESL/PubMedCLIP/tree/main/PubMedCLIP
https://huggingface.co/microsoft/BiomedCLIP-PubMedBERT_256-vit_base_patch16_224 |

### A.2.2 Segmentation

Tab. 4 details the segmentation FMs used in this paper, which can be categorized into 2D natural SegFMs, 2D medical SegFMs, and 3D medical SegFMs. These models are based on the origin SAM architecture as shown in Fig. 6 (b), which consists of an image encoder, a prompt encoder and a mask decoder.

Table 4: FMs used in segmentation.

| Category | Domain | Model | Link |
|---|---|---|---|
| 2D | Natural | SAM [30]
MobileSAM [71]
TinySAM [56] | https://github.com/facebookresearch/segment-anything
https://github.com/ChaoningZhang/MobileSAM
https://github.com/xinghaochen/TinySAM |
| | Medical | MedSAM [38]
SAM-Med2D [9]
FT-SAM [9] | https://github.com/bowang-lab/MedSAM
https://github.com/OpenGVLab/SAM-Med2D
https://drive.google.com/file/d/1J4qQt9MZZYdv1eoxMTJ4FL8Fz65iUFM8/view |
| 3D | Medical | SAM-Med3D [63]
FastSAM3D [54]
SegVol [12] | https://github.com/OpenGVLab/SAM-Med2D
https://github.com/arcadelab/FastSAM3D
https://github.com/BAAI-DCAI/SegVol |

# B  Datasets Details

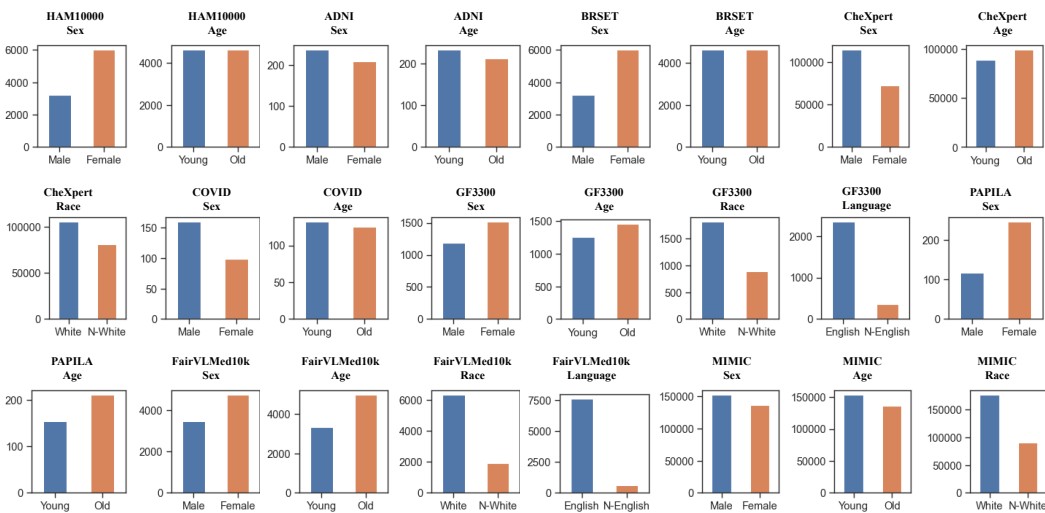

Figure 7: SA distribution of classification datasets.

## B.1  Classification Datasets

**CheXpert** is a large dataset of chest X-rays labeled for the presence of 14 different observations as well as uncertainty labels. It includes 221006 chest X-ray images, making it widely used for the development and evaluation of medical imaging models. Many FMs, like MedCLIP, adapt it as part of the training data.

**MIMIC-CXR** is another publicly available dataset of chest radiographs with corresponding radiology reports. This dataset, consisting of 357542 chest X-ray images, is part of the MIMIC family, providing rich clinical context and metadata alongside imaging data.

**HAM10000** dataset consists of 9948 dermatoscopic images of pigmented skin lesions. These images are categorized into seven different types of skin conditions. It is used for both classification and segmentation fairness evaluation.

**FairVLMed10k** is a medical dataset containing 10000 diverse visual-linguistic pairs designed to address fairness in medical AI, where it was first used to train the FairCLIP [36]. It includes various types of medical images of the eye paired with descriptive text annotations.

**GF3300** also a dataset designed for evaluating medical fairness, which includes 3,300 subjects with both 2D and 3D imaging data of the retinal nerve, helping to promote fairness in medical AI for glaucoma detection.

**PAPILA** is a dataset of papillary thyroid carcinoma images, accompanied by detailed clinical and pathological information. This dataset includes 420 color fundus images of the eye, aiming to aid in the development of diagnostic tools for thyroid cancer.

**BRSET** consists of 16266 breast ultrasound images, annotated with benign and malignant labels. These color fundus images of the eye are crucial for advancing breast cancer detection and classification algorithms.

**COVID-CT-MD** is a dataset of computed tomography scans for patients diagnosed with COVID-19. It includes 305 lung CT images with annotations for COVID-19 manifestations, aiding in the development of diagnostic tools for the pandemic.

**ADNI-1.5T** is part of the Alzheimer's Disease Neuroimaging Imaging and includes MRI scans acquired at 1.5 Tesla. It consists of 550 brain MRI images, used extensively in research focused on early detection and progression tracking of Alzheimer's disease.

Tab. 5 lists the references, modality, body part, size, and SAs information of these datasets.

Table 5: Classification datasets details.

| Type | Dataset | Modality | Body Part | # Images | Sensitive Attribute |
|------|---------|----------|-----------|----------|---------------------|
| 2D | CheXpert [22] | Chest X-ray | Chest | 221,006 | Sex, Age, Race |
| | MIMIC-CXR [25] | Chest X-ray | Chest | 357,542 | Sex, Age, Race |
| | HAM10000 [61] | Dermatoscopy | Skin | 9,948 | Sex, Age |
| | FairVLMed10k [36] | SLO Fundus | Eye | 10,000 | Sex, Age, Race, Language |
| | GF3300 [37] | OCT RNFL thickness | Eye | 3,300 | Sex, Age, Race, Language |
| | PAPILA [31] | Color Fundus | Eye | 420 | Sex, Age |
| | BRSET [42] | Color Fundus | Eye | 16,266 | Sex, Age |
| 3D | COVID-CT-MD [1] | Lung CT | Chest | 305 | Sex, Age |
| | ADNI-1.5T [46] | Brain MRI | Brain | 550 | Sex, Age |

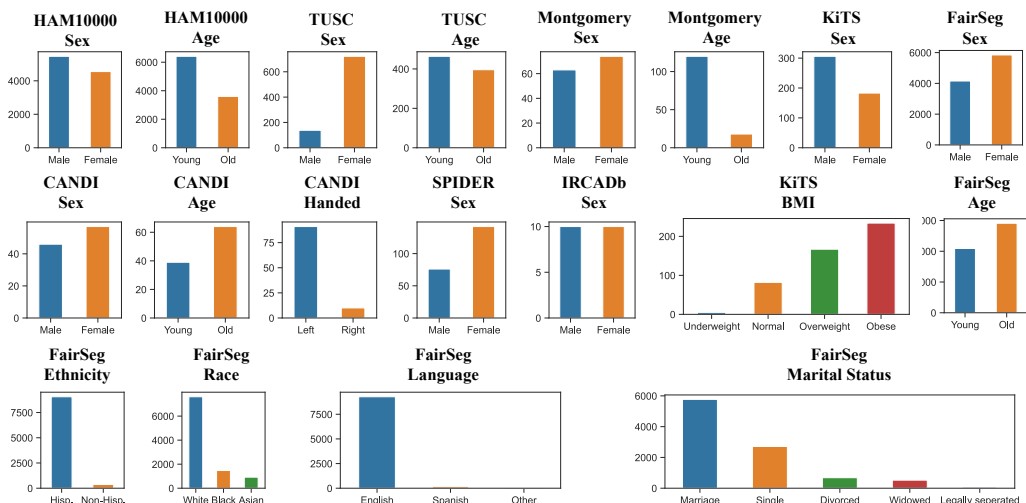

Figure 8: SA distribution of segmentation datasets.

## B.2 Segmentation Datasets

**HAM10000** contains 10,000 2D RGB dermatology images, with binary segmentation masks for skin lesion. We use sex and age as the sensitive attribute. For age, we regard age larger than 60 as the *old* group, and age smaller than 60 as the *young* group.

**TUSC** contains 860 2D thyroid ultrasound images, with binary segmentation masks for thyroid nodule. We use sex and age as the sensitive attribute. For age, we regard age larger than 60 as the *old* group, and age smaller than 60 as the *young* group.

**FairSeg** contains 10,000 2D OCT images, with three-class segmentation masks for optic cup and rim. We use sex, age, race, language, and marital status as the sensitive attribute. For age, we regard age larger than 60 as the *old* group, and age smaller than 60 as the *young* group. We categorize race into *White*, *Black*, and *Asian*. We categorize language into *English*, *Spanish*, and *Other*. We categorize marital status into *Marriage or Partnered*, *Single*, *Divorced*, *Widowed* and *Legally seperated*. We categorize ethnicity into *Hispanic* and *Non-Hispanic*.

**Montgomery County X-ray (montgomery)** contains 137 2D chest X-ray images, with three-class segmentation masks for left lung and right lung. We use sex and age as the sensitive attribute. For age, we regard age larger than 60 as the *old* group, and age smaller than 60 as the *young* group.

**KiTS2023 (KiTS)** contains 489 3D kidney CT volumes, with four-class segmentation masks for kidney, tumor, and cyst. We use sex and bmi as the sensitive attribute. For bmi, we categorize into underweight, normal, overweight, and obese following [14].

**CANDI** contains 103 3D brain MRI volumes, with multi-class segmentation masks for many brain structures. We select six classes including left/right brain white matter, left/right cerebral cortex, and left/right ventricle for segmentation. We use sex, age, and handedness as the sensitive attribute. For

Table 6: Segmentation datasets details.

| Type | Dataset | Modality | Body Part | # Images | Sensitive Attribute |
|---|---|---|---|---|---|
| 2D | HAM10000 [61] | Dermatology | Skin | 10,015 | Sex, Age |
| | TUSC [53] | Ultra Sound | Thyroid | 860 | Sex, Age |
| | FairSeg [60] | OCT | Eye | 10,000 | Sex, Age, Race, Marital Status, etc. |
| | Montgomery County X-ray [7] | X-ray | Chest | 137 | Sex, Age |
| 3D | KiTS 2023 [18] | CT | Kidney | 489 | Sex, Body Mass Index (BMI) |
| | CANDI [27] | MRI | Brain | 103 | Sex, Age, Handedness |
| | IRCADb [58] | CT | Liver | 20 | Sex |
| | SPIDER [62] | MRI | Spine | 218 | Sex |

age, we regard age larger than 10 as the *old* group, and age smaller than 10 as the *young* group as the maximum age in CANDI dataset is about 16. For handedness, we categorize it into *left-handed* and *right-handed*.

**IRCADb** contains 20 3D liver CT volumes, with multi-class segmentation masks for many abdomen organs. We select six classes including class 1, 17, 33, 65, 129, and 193 for segmentation. We use sex as the SA.

**SPIDER** contains 218 3D spine MRI volumes, with multi-class segmentation masks for vertebra and disc. We use all the fifteen classes for segmentation, and use sex as the sensitive attribute.

Tab. 6 lists reference, modality, body part, size, and SAs information of these datasets.

# C Metrics

## C.1 Utility

We use Area Under the receiver operating characteristic Curve (AUC) and Accuracy for evaluating utility in classification and Dice Similarity Coefficient (DSC) for segmentation. The formulas for these metrics are:

$$\text{AUC} = \frac{1}{n_1 n_0} \sum_{i=1}^{n_1} \sum_{j=1}^{n_0} \mathbf{1}(s_i > s_j)$$

where $n_1$ is the number of positive samples, $n_0$ is the number of negative samples, $s_i$ is the score for the $i$-th positive sample, and $s_j$ is the score for the $j$-th negative sample. The indicator function $\mathbf{1}(s_i > s_j)$ is 1 if $s_i$ is greater than $s_j$, and 0 otherwise.

$$\text{ACC} = \frac{TP + TN}{TP + TN + FP + FN}$$

where $TP$ is the number of true positives, $TN$ is the number of true negatives, $FP$ is the number of false positives, and $FN$ is the number of false negatives.

$$\text{DSC} = \frac{2 \cdot |\hat{Y} \cap Y|}{|\hat{Y}| + |Y|}$$

where $\hat{Y}$ is the set of predicted positive samples, and $Y$ is the set of actual positive samples. The DSC measures the overlap between the predicted and actual positive samples.

## C.2 Fairness

Fairness measurements are categorized into three criteria: Positive-Parity Fairness, Outcome-Consistency Fairness, and Predictive-Alignment Fairness [8].

*Positive-Parity Fairness* metrics primarily consider the positive rate, ensuring equal consideration in positive classifications for both unprivileged and privileged groups. For example, in disease screening, it is crucial that both groups have an equal chance of being flagged as positive cases, ensuring no group is overlooked. We apply Demographic Parity (DP) as the criterion for this group. The formula for Disparity Impact is given by:

$$\text{DP} = \left| \Pr(\hat{Y} = 1 | A = 0) - \Pr(\hat{Y} = 1 | A = 1) \right|$$

where $Y$ is the prediction conditioned by SA. This ratio measures the disparity between the group with the minimum performance and the group with the maximum performance.

*Outcome-Consistency Fairness* measures assess the discrepancies in confusion matrix components between different sensitive groups. Common metrics include Equal Opportunity and Equal Odds. Equal Opportunity assesses whether both groups receive approximately equal scores by calculating the gaps in Accuracy ($\text{Acc}_\Delta$), AUC ($\text{AUC}_\Delta$), and DSC ($\text{DSC}_\Delta$). Equalized Odds ensures that the model performs equally well across SA groups in terms of both TPR and FPR. In our study, we measure equal odds by calculating the differences in TPR and FPR between the advantaged and disadvantaged groups. The formulas for these metrics are:

$$\text{AUC/ACC/DSC}_\Delta = \text{AUC/ACC/DSC}_{\max} - \text{AUC/ACC/DSC}_{\min}$$

$$\text{AUC/ACC/DSC}_{\min} = \min \text{AUC/ACC/DSC}$$

$$\text{AUC/ACC/DSC}_{\text{Skew}} = \frac{\max_i \left( 1 - \text{AUC/ACC/DSC}_a \right)}{\min_i \left( 1 - \text{AUC/ACC/DSC}_a \right)}$$

$$\text{EqOdds} = \frac{1}{2} \left| \Pr(\hat{Y} = 1 | Y = 1, A = 0) - \Pr(\hat{Y} = 1 | Y = 1, A = 1) \right|$$
$$+ \frac{1}{2} \left| \Pr(\hat{Y} = 1 | Y = 0, A = 0) - \Pr(\hat{Y} = 1 | Y = 0, A = 1) \right| \tag{1}$$

*Predictive-Alignment Fairness* metrics focus on the predicted probabilities or scores, aiming to evaluate how well the predicted probabilities of outcomes align with the actual outcomes. The Expected Calibration Error (ECE) evaluates if the predicted scores are indicative of true likelihoods, thus providing a reliable basis for decision-making across different groups, where a high value leads to an optimal decision threshold. We measure the gap of ECE between different SA group, $\text{ECE}_\Delta$, where a higher gap indicates strong bis in terms of the predictive alignment. The formulas for it is:

$$\text{ECE}_\Delta = \left| \frac{1}{N} \sum_{i=1}^{N} |p_i - o_i| - \frac{1}{N'} \sum_{j=1}^{N'} |p'_j - o'_j| \right|$$

where $N$ is the total number of samples, $p_i$ is the predicted probability for sample $i$, and $o_i$ is the actual outcome for sample $i$. $N$ and $N'$ belong to two different subgroups.

*Representation Fairness* inspects the integration between the model and the dataset, trying to figure out the relationship between fairness and the feature distribution in the latent space. Generally, a feature distribution that is hard to separate will result in lower bias. In this paper, we visualize the representation fairness by t-SNE [19].

## C.3 Fairness-utility Tradeoff

The fairness-utility tradeoff pursues a balance between fairness and utility. Following [37, 60], we use Equity Scaling measurements of AUC ($\text{AUC}_{\text{ES}}$) and DSC ($\text{DSC}_{\text{ES}}$) for classification and segmentation, respectively. The equations are as follows.

$$\text{AUC}_{\text{ES}} = \frac{\overline{\text{AUC}}}{1 + \text{AUC}_\Delta} \tag{2}$$

$$\text{DSC}_{\text{ES}} = \frac{\overline{\text{DSC}}}{1 + \text{DSC}_\Delta} \tag{3}$$

where $\overline{\text{AUC}}$ and $\overline{\text{DSC}}$ are the average AUC and DSC over all data samples. $\text{AUC}_\Delta$ and $\text{DSC}_\Delta$ are the standard deviation of AUC and DSC across all subgroups defined by sensitive attributes, respectively.

## C.4 Statistics Test

We perform statistical significance tests to ensure that any observed performance in benchmarking FMs' s performance in medical imaging is not due to occasion on specific datasets. It assesses their performance across various datasets to draw meaningful and robust conclusions. We adapt the statistics evaluation in [77], where the relative ranks for each FM's performance are calculated on individual datasets and then averaged. The Friedman test is executed and the Critical Difference (CD) figure is plotted.

# D   Experimental Details

## D.1   Classification

The classification pipeline follows a straightforward approach, employing common data pre-processing strategies. During hyper-parameter selection, we first determine the optimal learning rate using LP. This learning rate is then applied to CLIP-Adapt and other PEFT methods, given the similar parameter scales of the adapters.

### D.1.1   Data Pre-processing

For 2D datasets, we resize all images to $256 \times 256$ and then apply `CenterCrop` to achieve a size of $224 \times 224$. All datasets are normalized using the ImageNet mean and standard deviation, as most FMs are initialized with these parameters. For 3D datasets, we utilize a 2.5D loading approach. Initially, the volumes are resized to a longitudinal axis size of 32. Subsequently, slices are processed independently through the 2D pre-processing pipeline and input into 2D foundation models. The final volume-wise prediction is obtained by maximizing the predictions of all slices in the volume.

### D.1.2   Subgroup Definition

We follow previous works to binarize sensitive attributes and define subgroup pairs [22, 77]. The sensitive attributes included in `FairMedFM` are listed in Tab. 5 and Tab. 6.

**Sex**. We follow the metadata in the original dataset to binarize the data into Male and Female subgroups.

**Age**. We use different thresholds to distinguish between Young and Old data points. By default, we use a threshold of 60 for all datasets except COVID-CT-MD and ADNI-1.5T, where the thresholds are 50 and 75, respectively. These threshold choices are primarily aimed at constructing a balanced testing set and ensuring a sufficient number of data points.

**Race**. We split data samples into White and Non-white subgroups.

**Language**. We split data samples into English and Non-English subgroups.

**BMI & Handedness & Marital Status**. The subgroup splitting of these sensitive attributes is introduced in Sec. B.2.

### D.1.3   Hyper-parameters

In classification, we initially use LP to identify the optimal learning rate and batch size. Once the loss converges and the training and testing performance align, we apply the same set of hyper-parameters for CLIP-Adapt, LP, and full fine-tuning. For all experiments, we use the AdamW optimizer with cosine annealing schedule configured with batch size of 128 and weight decay of 0.05. All models are trained for 100 epochs, during which we observe the convergence of loss and the alignment of training and testing metrics. Our experiments are conducted on a single NVIDIA A100 GPU.

We also include multiple unfairness mitigation algorithms, of which the hyper-parameters are grid-searched and listed as follows:

**LP & CLIP-Adapt & Resampling.** Learning rate: $2.5 \times 10^{-4}$.

**Group DRO.** Learning rate: $2.5 \times 10^{-4}$; Group adjustments: 1.

**LAFTR.** Learning rate: $3 \times 10^{-4}$; Adversarial coefficients: 0.1.

## D.2   Segmentation

The evaluation of SegFMs consists of two step, i.e. data pre-processing, prompt generation, and network inference. Note that for multi-class tasks, we process each class seperately, and average the results across different classes.

### D.2.1 Data Pre-processing

As there are 2D datasets and 3D datasets used in this paper, and different SegFMs can only accept either 2D or 3D input, we first pre-process the datasets as follows:

**2D models + 2D datasets.** The RGB grayscale images and grayscale images are resize to (1024, 1024, 3), and directly sent to 2D SegFMs including SAM, MobileSAM, TinySAM, MedSAM, SAM-Med2D, and FT-SAM.

**2D models + 3D datasets.** The 3D MRI/CT volumes are firstly normalized to [0, 1]. Then, the slices along the Axial plane are splited, and only slices that have ground truth segmentation masks are resized to (1024, 1024, 3) and saved for evaluation using 2D SegFMs.

**3D models + 3D datasets.** The 3D MRI/CT volumes are firstly normalized to [0, 1], and directly sent to 3D SegFMs including SAM-Med3D, FastSAM-3D, and SegVol.

### D.2.2 Prompt Generation

In this paper, the prompts are generated from the ground truth mask to obtain better utilities. For *rand* and *rands*, we use Random Number Generator with equal weights for each point. Following the official implementation, we use *center*, *rand*, *rands* and *bbox* for all the 2D SegFMs, use *rand* and *rands* for SAM-Med3D and FastSAM-3D, and use *rands* and *bbox* for SegVol.

### D.2.3 Network Inference

The DSC scores are computed based on the origin shape of the input image. For 2D models + 2D datasets and 3D models + 3D datasets, we directly compute the sample-wise DSC, for 2D models + 3D dataset, we first aggregate slice-wise results to get sample-wise prediction, and then compute the sample-wise DSC.

### D.2.4 t-SNE Visualization

t-SNE are presented for only 2D datasets + 2D models. We use feature map after the image encoder, which is of shape (256, 64, 64). We average the feature map across the second and the third channel to get a feature vector with the shape of 256, and t-SNE is computed using Python scikit-learn package.

### D.2.5 Hyper-parameters

We finetune the original SAM using the implementation of SAMed on HAM10000 dataset. The HAM10000 dataset is randomly split into train and test with a ratio of 8:2. Earlystop strategy is applied in the training procedure. The hyper-parameters for unfairness mitigation are as follows:

**SAMed.** Learning rate: 0.005; Optimizer: AdamW; Max epoch: 100; Batchsize: 16.

**FEBS.** Learning rate: 0.005; Optimizer: AdamW; Max epoch: 20; Batchsize: 16; Dice loss coefficient: 0.8. FEBS loss temperature coefficient: 1.

**Resampling.** Learning rate: 0.005; Optimizer: AdamW; Max epoch: 20; Batchsize: 8.

**InD.** Learning rate: 0.005; Optimizer: AdamW; Max epoch: 20; Batchsize: 16.

# E Additional Benchmarking Results

## E.1 Results of More Sensitive Attributes

In this section, we validate our conclusions in Sec. 4 with experiments on more sensitive attributes, including age, race, language, BMI, etc. The results demonstrate consistency in our findings across various sensitive attributes.

**Bias widely exists in FMs for medical imaging**. Similar to Fig. 2, we report classification results on more sensitive attributes in Fig. 9 and Fig. 10, where $AUC_\Delta$ and DP are fairness metrics, respectively. Fig. 11 reports segmentation results on more sensitive attributes, where $DSC_\Delta$ is the fairness metric.

**Careful selection and use of FMs are needed for ensuring a good fairness-utility trade-off**. Similar to Fig. 3, we report the fairness-utility trade-off with additional sensitive attributes regarding various usages, prompts, FM types on classification and segmentation in Fig. 12 and Fig. 13, respectively.

**Consistent disparities in SA occur across various FMs on the same dataset**. Similar to Fig. 5, we further investigate the utility skewness between the Male and the Female on segmentation tasks. As shown in Fig. 14, the utility skewness is basically consistent within each dataset, despite of the type of SegFMs. For example, SegFMs perform worse on the Male group than the Female group in the HAM10000 dataset, while it is easier for SegFMs to segment lung for the Male group in the Montgomery dataset. Besides, compared to 3D models, this trend is more consistent for 2D models. Potential reasons for this phenomenon could be the differences in the size of the masks (the mask in 3D datsets are larger) and the number of classes of masks (3D datasets have more than 2 classes of masks). This task complexity may lead to performance variations for different SegFMs.

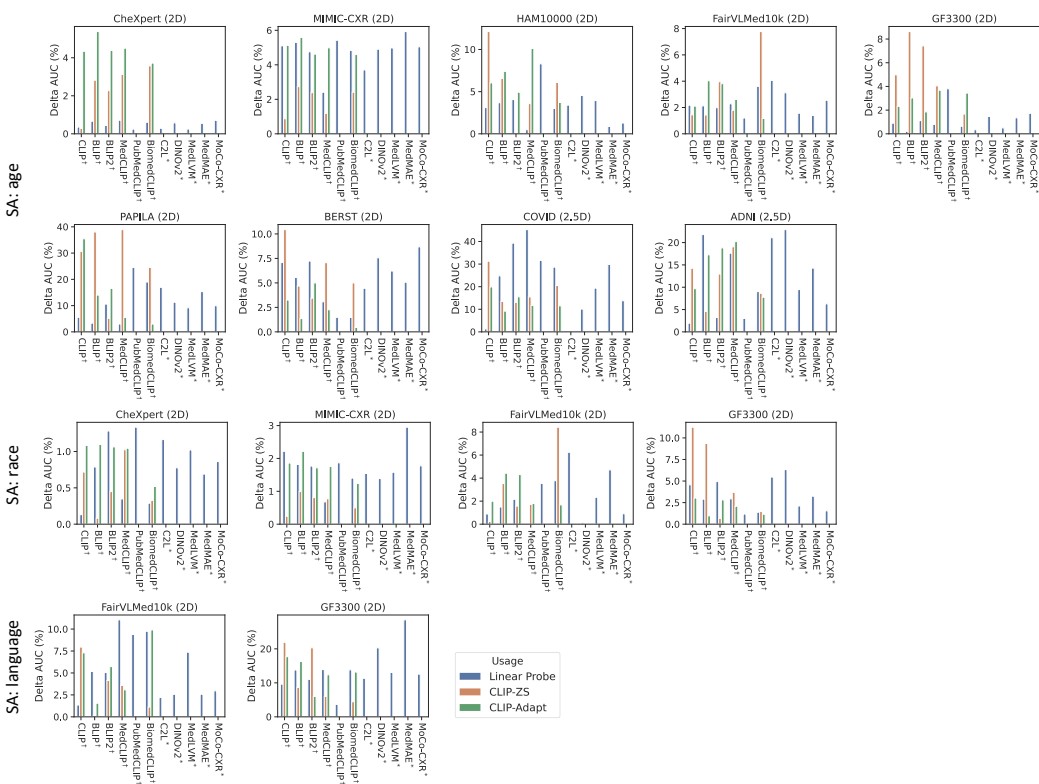

Figure 9: Bias in classification tasks with more SAs. $AUC_\Delta$ is the fairness evaluation metric. "†" denotes vision-language models, and "∗" denotes pure vision models, where CLIP-ZS and CLIP-Adapt are not applicable.

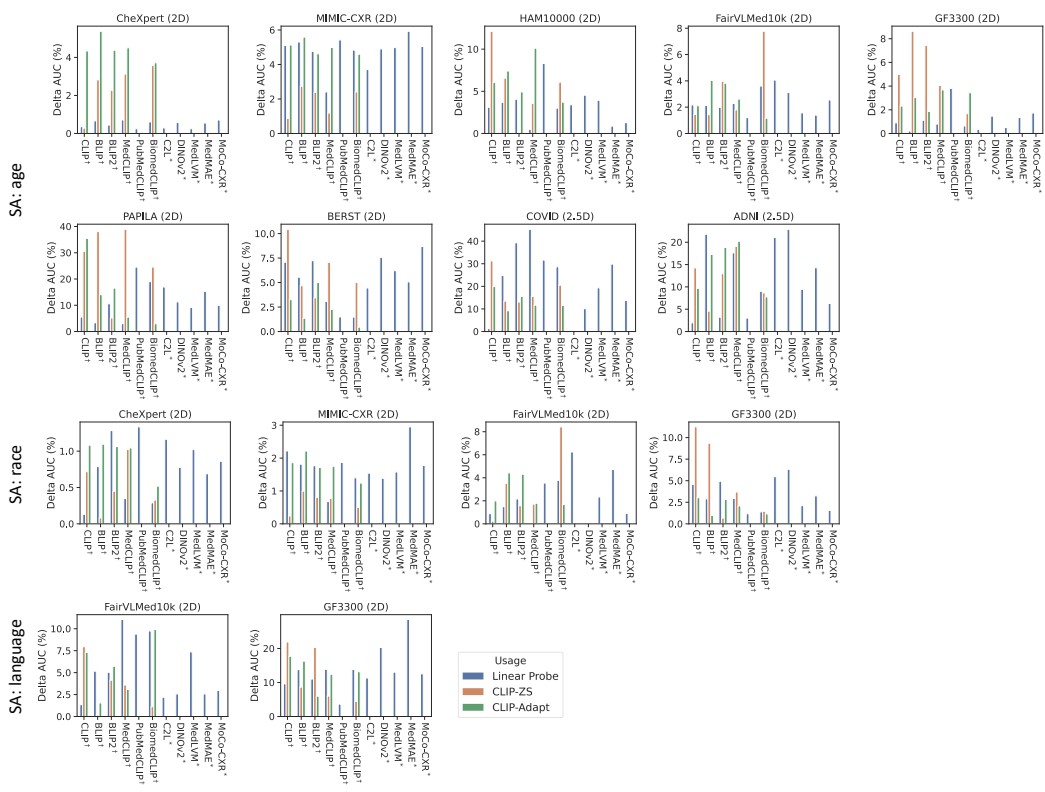

Figure 10: Bias in classification tasks with more SAs. DP is the fairness evaluation metric. "†" denotes vision-language models, and "∗" denotes pure vision models, where CLIP-ZS and CLIP-Adapt are not applicable.

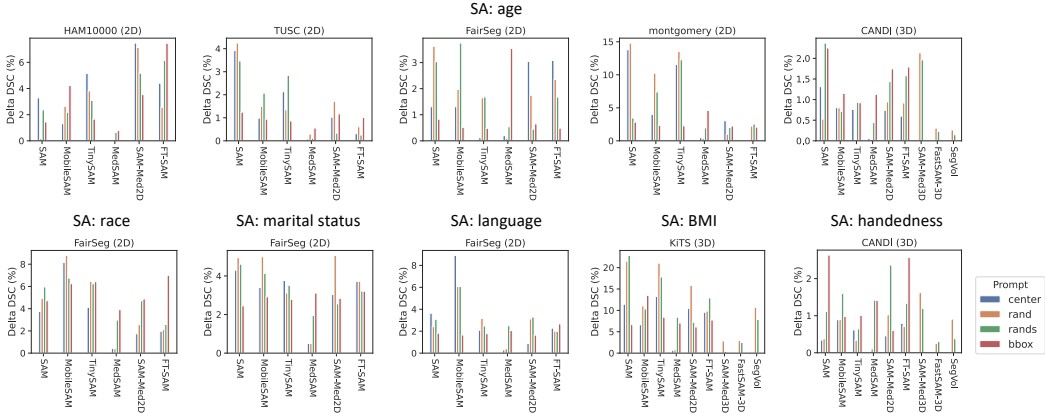

Figure 11: Bias in segmentation tasks with more SAs. $DSC_\Delta$ is the fairness evaluation metric. Note that SAM-Med3D, FastSAM-3D, and SegVol are only applicable for 3D datasets.

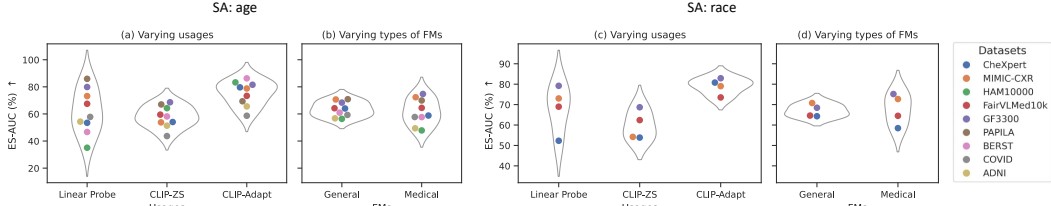

Figure 12: Fairness-utility tradeoff in FMs for different components on classification with more SAs. We use AUC$_{ES}$ and DSC$_{ES}$ as evaluation metrics. (a, c) Fairness-utility tradeoff for different classification usages (using CLIP as an example); (b, d) Fairness-utility tradeoff for general-purpose and medical-specific FMs.

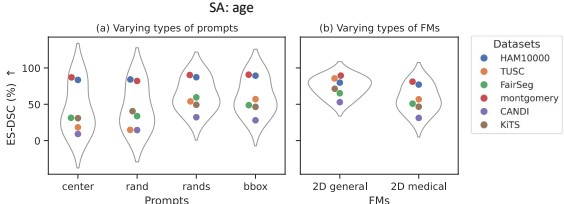

Figure 13: Fairness-utility tradeoff in FMs for different components on segmentation with more SAs. We use AUC$_{ES}$ and DSC$_{ES}$ as evaluation metrics. (a) Fairness-utility tradeoff for different prompts in segmentation (using SAM-Med2D as an example); (b) Degree of fairness for different SegFM categories.

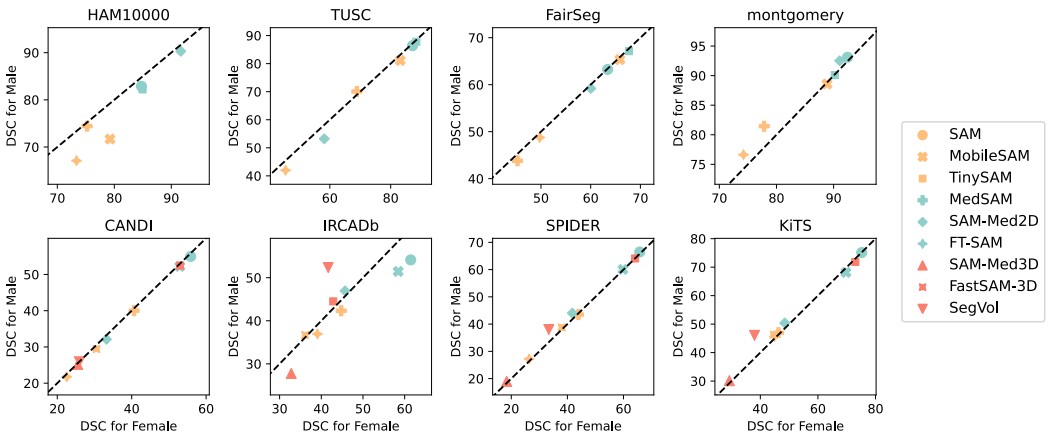

Figure 14: ●:General purpose 2D SegFMs; ●: Medical 2D SegFMs; ●: Medical 3D SegFMs.

### E.2 More Observations

**No FM significantly outperforms others in terms of utility and fairness**. Fig. 15 presents the *statistical test* using a CD diagram. In CD diagrams, FMs connected by a horizontal line belong to the same group, indicating no significant difference based on the p-value. As shown, all models belong to the same group in terms of *overall AUC* in (a), meaning no FM outperforms the others in utility. Similarly, for fairness metrics, *EqOdds* in (c), AUC$\Delta$ in (e), and ECE$\Delta$ in (f) also show all models grouped in a single fold. Additionally, the utility-fairness tradeoff indicated by AUC$_{ES}$ in (d) shows that no FM outperforms the others. The plots indicate that no FM consistently achieves a superior tradeoff; all models are statistically similar and fall within the same performance group.

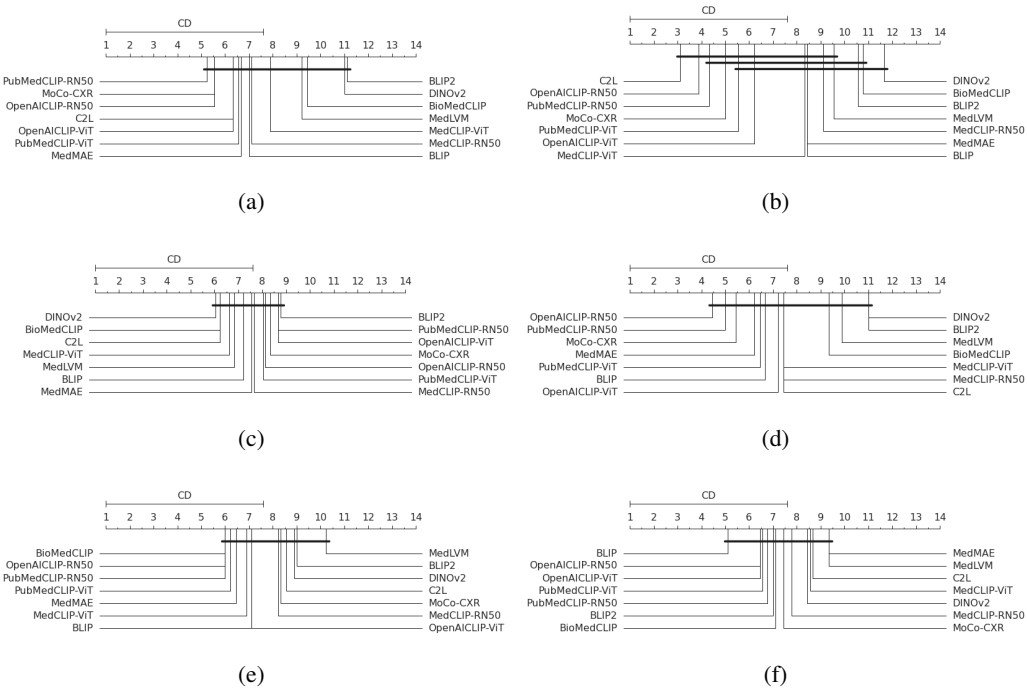

Figure 15: The CD diagram on LP for (a) Overall AUC; (b) Overall ECE; (c) EqOdds; (d) AUC$_{ES}$; (e) AUC$_{\Delta}$; (f) ECE$_{\Delta}$

**Representation fairness**. The t-SNE of 2D SegFMs on the four 2D datasets are shown in Fig. 16. Here we use sex as an example. As shown in Fig. 16, compared to the TUSC dataset and FairSeg dataset, the latent space of the HAM10000 dataset is more separatable. This is roughly aligned with the results in Fig. 3, where the DSC$_\Delta$ of the HAM10000 dataset are larger than the rest datasets. This finding provides potential for us to mitigate unfairness for SegFMs by manipulating the latent space, which has been explored in APPLE [67]. On the other hand, considering that the group-wise separablity is similar acrosss different SegFMs, these four tasks might suffer more unfairness from the data than from the model.

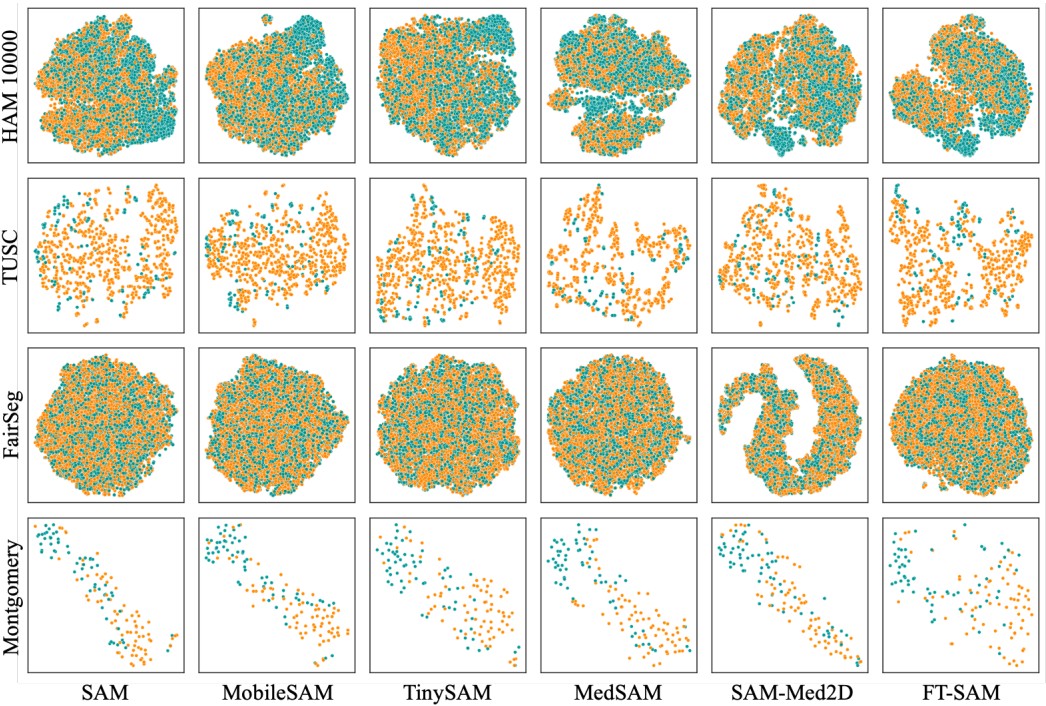

Figure 16: Unfairess potentially exists in the latent space of SegFMs. •: data points of the Male; •: data points of the Female.

**Visualization of the fairness-utility trade-off.** In Fig. 17 and Fig. 18, we visualize the fairness-utility trade-off of various foundation models across different datasets, using different tasks or prompts for classification and segmentation tasks. Our results demonstrate that careful selection and use of foundation models are crucial for achieving a favorable fairness-utility trade-off in both tasks. This observation is consistent with the results using fixed equity-scaling metrics, as shown in Fig. 4. Additionally, the analysis from Sec. 4 is reflected in these new visualization results.

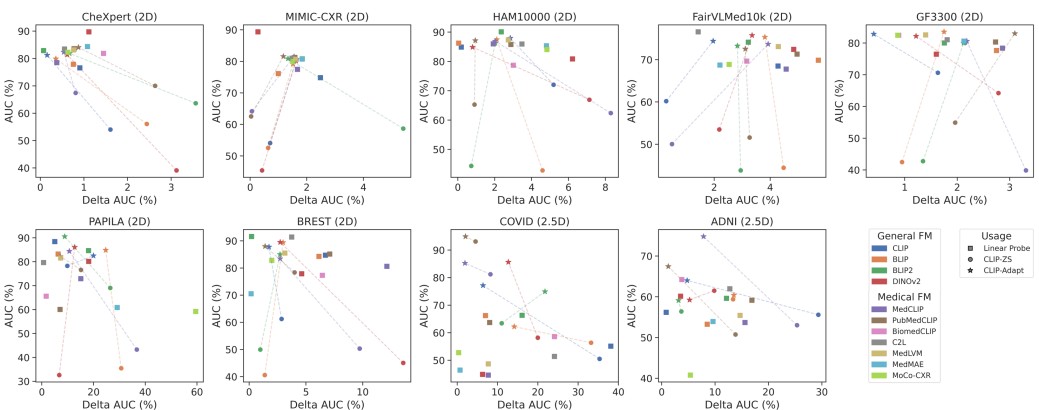

Figure 17: The fairness-utility trade-off for classification tasks across different datasets, usages, and foundation models, using sex as the sensitive attribute. The dotted line represents the adaptation of CLIP models. The upper left corner of each plot signifies optimal fairness-utility trade-offs.

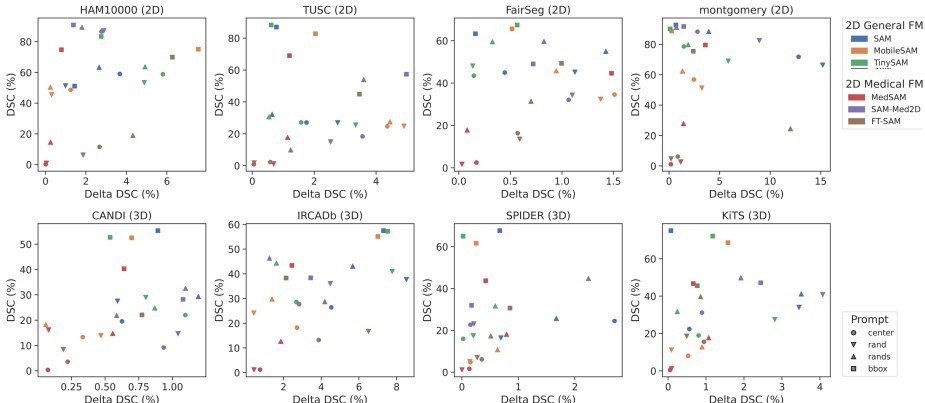

Figure 18: The fairness-utility trade-off for segmentation tasks across different datasets, prompts, and foundation models, using sex as the sensitive attribute. The upper left corner of each plot signifies optimal fairness-utility trade-offs.

## E.3 Detailed Results

In this part, we provide the detailed numeric results in this paper. Table 7-9 list the results of the classification task, and Table 10-18 list the results of the segmentation task. Please note that the codebase gets refactored and continuously evolves to adapt new models and tasks, thus we highly recommend you use the open-sourced FairMedFM following App. F to launch the code.

Table 7: Classification results for ADNI, BERST, and COVID datasets, *sex* as the sensitive attribute. All experiments are repeated three times and mean $\pm$ std are reported (%).

| Dataset | Usage | Model | AUC | ACC | BCE | ECE | ΔAUC | ΔECE | DP | EOD | AUC$_{Male}$ | AUC$_{Female}$ | ECE$_{Male}$ | ECE$_{Female}$ |
|---|---|---|---|---|---|---|---|---|---|---|---|---|---|---|
| ADNI | Linear Probe | BLIP | 55.79±2.52 | 47.76±0.45 | 69.32±0.29 | 16.27±2.38 | 16.44±8.24 | 4.87±2.75 | 5.77±3.33 | 12.01±4.39 | 50.03±7.07 | 60.78±10.49 | 18.28±3.28 | 16.90±2.26 |
| | | BLIP2 | 56.75±4.53 | 49.04±6.23 | 68.29±1.20 | 13.15±1.68 | 14.95±4.61 | 3.68±2.72 | 10.90±6.18 | 9.91±5.18 | 60.34±13.18 | 53.69±3.94 | 17.99±1.38 | 14.32±2.57 |
| | | BiomedCLIP | 56.54±2.27 | 48.40±6.58 | 69.89±1.02 | 14.08±2.56 | 10.29±7.16 | 3.09±1.23 | 9.62±5.77 | 10.11±7.16 | 58.55±8.96 | 55.79±5.83 | 16.22±0.95 | 17.13±2.20 |
| | | C2L | 56.13±7.35 | 46.79±9.10 | 69.50±0.41 | 14.50±2.51 | 6.86±6.81 | 3.51±1.84 | 2.56±1.11 | 3.99±2.79 | 57.53±4.84 | 54.61±12.15 | 18.31±1.62 | 18.42±2.60 |
| | | CLIP | 55.49±0.68 | 41.03±0.45 | 69.22±0.15 | 14.34±3.93 | 1.84±1.04 | 4.50±3.34 | 2.56±1.11 | 5.33±1.22 | 54.99±1.28 | 56.27±0.96 | 14.09±2.25 | 18.59±4.95 |
| | | DINOv2 | 57.87±4.22 | 54.17±5.05 | 67.97±0.66 | 12.79±1.23 | 9.58±3.69 | 4.21±5.16 | 1.92±0.00 | 8.53±4.41 | 58.58±9.02 | 55.97±2.85 | 16.82±4.53 | 19.32±0.94 |
| | | MedCLIP | 57.17±3.70 | 45.83±4.32 | 69.37±0.22 | 14.54±1.47 | 6.45±4.41 | 2.50±0.88 | 3.21±2.94 | 4.53±1.64 | 60.37±1.00 | 53.92±5.13 | 19.39±3.11 | 18.10±0.98 |
| | | MedLVM | 55.98±7.33 | 42.31±2.36 | 68.64±1.45 | 14.46±1.48 | 11.16±9.59 | 3.73±2.03 | 5.13±7.28 | 6.40±5.62 | 55.53±14.18 | 56.71±7.80 | 17.92±0.82 | 15.32±3.93 |
| | | MedMAE | 56.95±5.16 | 48.40±6.68 | 69.06±0.25 | 11.67±0.39 | 6.30±4.64 | 5.63±1.62 | 8.33±1.11 | 7.50±0.49 | 55.48±4.50 | 61.11±6.71 | 17.11±4.95 | 17.99±2.09 |
| | | MoCo-CXR | 47.60±10.58 | 45.51±7.94 | 69.85±0.96 | 14.66±1.57 | 4.58±2.74 | 3.25±0.44 | 10.26±16.13 | 11.73±14.81 | 47.11±11.19 | 47.08±10.37 | 19.23±2.88 | 20.02±0.34 |
| | | PubMedCLIP | 59.30±5.26 | 55.45±8.65 | 69.17±0.03 | 15.30±2.10 | 8.06±7.60 | 2.64±2.85 | 8.33±11.10 | 10.93±9.29 | 55.07±8.75 | 63.13±1.41 | 19.17±2.71 | 16.60±1.72 |
| | CLIP-ZS | BLIP | 59.41±0.00 | 64.42±0.00 | 68.88±0.00 | 16.23±0.00 | 13.36±0.00 | 8.17±0.00 | 13.46±0.00 | 84.87±0.00 | 66.21±0.00 | 52.84±0.00 | 17.53±0.00 | 25.69±0.00 |
| | | BLIP2 | 56.41±0.00 | 42.31±0.00 | 80.20±0.00 | 25.47±0.00 | 3.69±0.00 | 0.68±0.00 | 3.85±0.00 | 96.01±0.00 | 58.37±0.00 | 54.69±0.00 | 26.91±0.00 | 27.59±0.00 |
| | | BiomedCLIP | 51.30±0.78 | 39.74±0.56 | 338.74±50.21 | 58.58±1.41 | 11.83±2.94 | 0.10±0.10 | 1.28±1.11 | 97.08±0.93 | 56.53±1.08 | 44.70±1.87 | 58.69±1.17 | 58.65±1.32 |
| | | CLIP | 55.61±0.00 | 39.42±0.00 | 70.44±0.00 | 14.81±0.00 | 29.34±0.00 | 11.51±0.00 | 1.92±0.00 | 97.62±0.00 | 41.32±0.00 | 70.66±0.00 | 22.50±0.00 | 10.99±0.00 |
| | | MedCLIP | 55.25±7.56 | 49.68±6.75 | 75.45±2.56 | 22.72±1.59 | 8.04±1.55 | 2.19±1.25 | 4.49±2.22 | 93.06±2.31 | 53.51±2.94 | 56.84±11.96 | 23.48±2.63 | 24.74±0.75 |
| | | PubMedCLIP | 53.03±0.00 | 41.35±0.00 | 75.07±0.00 | 18.19±0.00 | 25.35±0.00 | 0.50±0.00 | 5.77±0.00 | 94.39±0.00 | 40.09±0.00 | 65.44±0.00 | 19.18±0.00 | 19.68±0.00 |
| | CLIP-Adapt | BLIP | 54.71±5.30 | 47.76±4.74 | 69.65±2.16 | 12.47±2.32 | 12.42±1.65 | 8.29±2.23 | 1.92±1.92 | 93.63±2.38 | 53.12±4.22 | 56.73±11.93 | 16.21±4.14 | 14.90±7.57 |
| | | BLIP2 | 58.60±3.11 | 49.36±8.40 | 67.54±0.63 | 13.22±4.13 | 4.84±3.82 | 1.48±1.67 | 10.26±10.59 | 86.69±7.26 | 59.06±6.35 | 57.76±2.03 | 14.37±3.07 | 13.60±1.86 |
| | | BiomedCLIP | 60.55±6.02 | 57.05±8.18 | 76.88±5.66 | 17.39±1.38 | 9.42±8.54 | 6.97±7.77 | 10.26±1.11 | 88.02±2.34 | 64.93±3.84 | 55.50±10.68 | 21.10±5.31 | 25.37±5.68 |
| | | CLIP | 65.40±2.70 | 57.05±6.26 | 65.49±1.16 | 10.48±2.37 | 2.87±2.30 | 5.19±2.26 | 1.28±1.11 | 97.08±3.22 | 63.39±2.77 | 66.26±3.15 | 15.16±3.46 | 18.63±1.98 |
| | | MedCLIP | 57.33±1.68 | 50.32±2.42 | 67.41±0.20 | 10.18±3.65 | 4.30±0.86 | 3.36±3.12 | 8.33±9.87 | 90.66±7.30 | 57.86±4.22 | 55.91±1.87 | 14.88±5.34 | 12.98±5.14 |
| | | PubMedCLIP | 74.45±0.56 | 69.23±2.54 | 62.60±0.33 | 13.22±2.57 | 5.99±4.02 | 4.34±4.76 | 17.95±1.11 | 82.64±1.18 | 71.02±2.31 | 77.01±1.98 | 19.57±2.20 | 15.23±3.28 |
| BERST | Linear Probe | BLIP | 84.21±0.11 | 92.15±0.57 | 50.84±0.09 | 32.02±0.07 | 6.15±0.10 | 3.99±0.05 | 0.62±0.24 | 9.43±2.03 | 82.07±0.14 | 88.22±0.07 | 30.04±0.08 | 34.03±0.08 |
| | | BLIP2 | 91.61±0.08 | 94.64±0.13 | 38.09±0.04 | 23.89±0.04 | 0.27±0.06 | 3.02±0.04 | 1.06±0.19 | 7.35±0.85 | 91.81±0.10 | 91.53±0.09 | 22.36±0.05 | 25.38±0.03 |
| | | BiomedCLIP | 85.09±0.19 | 92.42±0.46 | 43.81±0.03 | 25.87±0.07 | 7.06±0.28 | 2.86±0.16 | 0.23±0.18 | 4.59±0.27 | 82.21±0.07 | 89.26±0.34 | 24.42±0.05 | 27.28±0.12 |
| | | C2L | 77.33±0.17 | 85.27±1.09 | 65.76±0.32 | 40.94±0.19 | 6.16±0.29 | 2.74±0.02 | 3.13±0.57 | 12.10±2.35 | 75.09±0.31 | 81.26±0.16 | 39.56±0.20 | 42.30±0.18 |
| | | CLIP | 84.76±0.12 | 91.00±0.41 | 53.53±0.06 | 34.06±0.03 | 6.83±0.15 | 3.12±0.14 | 0.69±0.08 | 9.19±0.74 | 82.03±0.16 | 88.85±0.11 | 32.68±0.05 | 35.80±0.16 |
| | | DINOv2 | 91.35±0.11 | 94.95±0.22 | 33.05±0.07 | 19.55±0.06 | 3.67±0.08 | 2.70±0.10 | 0.54±0.13 | 5.60±1.36 | 89.92±0.08 | 93.59±0.15 | 18.18±0.10 | 20.88±0.03 |
| | | MedCLIP | 78.05±0.18 | 83.18±4.10 | 50.51±0.24 | 30.41±0.17 | 4.17±0.40 | 1.96±0.01 | 1.80±0.96 | 1.88±1.10 | 76.26±0.36 | 80.42±0.11 | 29.41±0.17 | 31.38±0.17 |
| | | MedLVM | 85.56±0.09 | 91.50±0.00 | 49.16±0.13 | 30.01±0.10 | 3.09±0.05 | 1.69±0.01 | 1.54±0.42 | 4.74±2.47 | 84.26±0.10 | 87.34±0.07 | 29.16±0.10 | 30.84±0.10 |
| | | MedMAE | 70.58±0.20 | 71.08±2.86 | 64.09±0.04 | 39.45±0.02 | 0.17±0.15 | 1.34±0.04 | 3.38±0.59 | 3.76±2.98 | 70.37±0.13 | 70.53±0.28 | 38.78±0.01 | 40.12±0.04 |
| | | MoCo-CXR | 82.17±1.22 | 91.52±0.54 | 56.54±0.74 | 35.49±0.57 | 3.18±1.10 | 2.52±0.16 | 0.81±0.25 | 2.14±2.13 | 80.85±1.34 | 84.03±1.26 | 34.22±0.51 | 36.74±0.62 |
| | | PubMedCLIP | 80.64±0.02 | 90.08±1.02 | 56.35±0.06 | 35.45±0.04 | 12.15±0.09 | 2.19±0.04 | 0.71±0.53 | 9.67±0.72 | 75.38±0.06 | 87.53±0.04 | 34.42±0.04 | 36.61±0.05 |
| | CLIP-ZS | BLIP | 40.59±0.00 | 7.52±0.00 | 453.89±0.00 | 91.42±0.00 | 1.38±0.00 | 2.61±0.00 | 0.12±0.00 | 99.02±0.00 | 41.43±0.00 | 40.05±0.00 | 90.10±0.00 | 92.71±0.00 |
| | | BLIP2 | 50.00±0.05 | 61.45±27.39 | 335.77±0.00 | 89.86±0.00 | 1.05±0.12 | 2.51±0.00 | 0.54±0.25 | 99.14±0.38 | 49.53±0.00 | 50.58±0.12 | 88.58±0.00 | 91.10±0.00 |
| | | BiomedCLIP | 78.16±0.46 | 85.27±0.56 | 28.62±0.86 | 5.58±1.50 | 2.95±0.91 | 2.74±2.24 | 3.84±1.39 | 88.62±3.15 | 77.42±0.45 | 80.37±0.89 | 5.98±1.42 | 6.20±2.83 |
| | | CLIP | 61.23±0.00 | 76.96±0.00 | 88.43±0.00 | 52.77±0.00 | 2.85±0.00 | 1.74±0.00 | 3.08±0.00 | 95.50±0.00 | 59.93±0.00 | 62.78±0.00 | 51.29±0.00 | 53.03±0.00 |
| | | MedCLIP | 52.18±7.78 | 54.80±40.43 | 67.79±4.02 | 41.58±2.25 | 6.64±6.07 | 2.63±0.05 | 1.54±1.18 | 96.53±4.17 | 53.89±6.58 | 49.97±11.28 | 40.26±2.24 | 42.89±2.24 |
| | | PubMedCLIP | 50.33±0.00 | 63.28±0.00 | 192.25±0.00 | 79.51±0.00 | 9.73±0.00 | 2.38±0.00 | 4.16±0.00 | 89.27±0.00 | 54.15±0.00 | 44.42±0.00 | 78.39±0.00 | 80.77±0.00 |
| | CLIP-Adapt | BLIP | 89.39±0.02 | 94.48±0.07 | 16.59±0.01 | 1.81±0.06 | 3.02±0.04 | 1.38±0.03 | 1.96±0.14 | 96.70±0.42 | 88.31±0.02 | 91.33±0.02 | 1.76±0.02 | 3.14±0.01 |
| | | BLIP2 | 85.01±0.03 | 91.87±0.21 | 20.58±0.02 | 3.10±0.00 | 2.76±0.08 | 0.25±0.21 | 1.35±0.13 | 99.47±0.37 | 83.94±0.05 | 86.70±0.03 | 2.50±0.03 | 2.75±0.23 |
| | | BiomedCLIP | 88.08±0.09 | 94.60±0.40 | 16.58±0.05 | 0.70±0.14 | 1.20±0.18 | 0.64±0.26 | 1.85±0.24 | 97.51±0.26 | 87.62±0.16 | 88.83±0.03 | 1.80±0.20 | 1.17±0.08 |
| | | CLIP | 87.76±0.00 | 92.57±0.07 | 18.51±0.00 | 1.65±0.03 | 1.74±0.00 | 0.26±0.01 | 2.29±0.00 | 99.00±0.14 | 88.64±0.00 | 86.89±0.00 | 2.27±0.02 | 2.01±0.03 |
| | | MedCLIP | 89.29±0.51 | 94.11±0.62 | 17.97±0.13 | 3.36±0.08 | 2.78±0.08 | 0.60±0.38 | 1.89±0.24 | 98.04±1.09 | 88.21±0.49 | 91.00±0.52 | 2.97±0.22 | 3.57±0.29 |
| | | PubMedCLIP | 83.44±0.00 | 93.10±0.00 | 19.98±0.00 | 1.99±0.02 | 2.69±0.03 | 0.58±0.02 | 1.25±0.00 | 96.57±0.00 | 82.38±0.01 | 85.08±0.01 | 2.12±0.02 | 2.70±0.01 |
| COVID | Linear Probe | BLIP | 50.67±13.90 | 54.17±7.41 | 69.35±0.63 | 17.05±1.71 | 15.73±12.24 | 7.40±5.34 | 2.78±2.41 | 6.76±2.67 | 45.69±19.99 | 54.43±15.06 | 26.93±5.46 | 19.53±2.59 |
| | | BLIP2 | 59.84±5.78 | 59.03±7.67 | 69.12±0.36 | 18.11±3.34 | 9.56±6.31 | 7.46±3.27 | 20.83±15.02 | 20.40±13.25 | 65.50±6.61 | 62.00±6.61 | 25.55±3.63 | 20.79±2.84 |
| | | BiomedCLIP | 60.47±3.71 | 60.42±2.95 | 69.28±1.42 | 18.82±3.35 | 10.14±11.34 | 8.63±5.70 | 15.28±16.84 | 18.53±11.31 | 56.99±9.52 | 67.13±5.72 | 20.38±2.13 | 29.01±2.68 |
| | | C2L | 57.03±1.56 | 51.39±1.96 | 69.26±0.02 | 15.66±4.73 | 12.82±10.23 | 9.56±2.52 | 8.33±0.00 | 9.56±2.02 | 55.83±7.13 | 61.89±10.58 | 29.32±0.15 | 24.99±8.86 |
| | | CLIP | 50.58±4.06 | 52.78±4.28 | 69.45±0.22 | 13.84±2.42 | 23.43±18.65 | 4.52±3.45 | 11.11±6.36 | 11.66±7.25 | 61.89±12.71 | 38.46±6.07 | 22.78±4.32 | 22.33±2.58 |
| | | DINOv2 | 50.38±2.22 | 53.47±3.93 | 74.38±6.31 | 19.57±4.32 | 17.83±12.45 | 10.77±5.50 | 6.94±4.81 | 7.58±3.59 | 55.24±9.09 | 39.74±7.39 | 22.34±10.64 | 27.45±0.43 |
| | | MedCLIP | 45.46±6.96 | 49.31±0.98 | 69.70±0.44 | 18.87±2.11 | 9.79±9.25 | 11.90±8.11 | 6.94±2.41 | 7.58±2.72 | 40.33±14.41 | 48.25±2.80 | 24.45±7.26 | 23.01±6.28 |
| | | MedLVM | 43.72±5.38 | 52.08±1.70 | 70.81±0.48 | 21.29±6.72 | 7.93±2.46 | 5.59±1.92 | 6.94±2.41 | 7.34±2.45 | 42.77±9.84 | 45.57±1.58 | 27.74±4.74 | 25.29±5.53 |
| | | MedMAE | 45.57±1.58 | 47.92±0.00 | 70.17±0.56 | 16.68±1.77 | 11.42±9.45 | 7.23±2.28 | 4.17±0.00 | 3.85±0.00 | 44.96±4.59 | 53.38±5.05 | 25.41±3.66 | 21.27±3.58 |
| | | MoCo-CXR | 45.31±10.21 | 52.08±2.95 | 69.62±0.27 | 17.81±3.50 | 12.94±11.04 | 6.35±6.44 | 9.72±6.36 | 11.31±6.54 | 38.34±13.47 | 51.28±8.11 | 18.55±2.40 | 21.22±6.15 |
| | | PubMedCLIP | 47.74±7.21 | 47.92±1.70 | 69.42±0.09 | 19.02±1.59 | 3.73±3.85 | 2.73±2.27 | 6.94±2.41 | 7.34±2.45 | 48.60±8.22 | 44.87±7.19 | 22.71±4.00 | 19.98±2.18 |
| | CLIP-ZS | BLIP | 56.36±0.00 | 66.67±0.00 | 88.44±0.00 | 26.92±0.00 | 33.21±0.00 | 7.53±0.00 | 4.17±0.00 | 97.06±0.00 | 72.27±0.00 | 39.06±0.00 | 23.20±0.00 | 30.72±0.00 |
| | | BLIP2 | 63.43±0.00 | 75.00±0.00 | 78.25±0.00 | 23.95±0.00 | 11.04±0.00 | 8.25±0.00 | 12.50±0.00 | 80.46±0.00 | 68.07±0.00 | 57.03±0.00 | 23.08±0.00 | 31.34±0.00 |
| | | BiomedCLIP | 91.78±1.54 | 84.03±2.41 | 57.67±6.37 | 21.46±1.75 | 5.42±0.82 | 1.98±0.93 | 5.56±4.81 | 94.52±2.59 | 89.64±1.28 | 95.05±1.19 | 21.80±1.10 | 21.77±1.54 |
| | | CLIP | 50.51±0.00 | 66.67±0.00 | 192.06±0.00 | 31.06±0.00 | 35.31±0.00 | 4.07±0.00 | 4.17±0.00 | 96.88±0.00 | 31.09±0.00 | 66.41±0.00 | 29.08±0.00 | 33.16±0.00 |
| | | MedCLIP | 51.18±7.29 | 68.06±1.20 | 66.94±2.50 | 17.24±3.96 | 15.61±3.91 | 4.96±2.19 | 6.94±2.41 | 92.30±4.18 | 56.02±14.95 | 48.70±11.55 | 24.77±2.73 | 27.42±2.73 |
| | | PubMedCLIP | 81.21±0.00 | 77.08±0.00 | 76.95±0.00 | 26.57±0.00 | 8.25±0.00 | 3.17±0.00 | 16.67±0.00 | 85.16±0.00 | 84.03±0.00 | 75.78±0.00 | 27.13±0.00 | 30.30±0.00 |
| | CLIP-Adapt | BLIP | 65.25±10.53 | 71.53±5.24 | 64.80±8.86 | 15.86±2.31 | 9.03±6.54 | 7.38±0.18 | 8.33±4.17 | 91.28±6.42 | 60.50±13.99 | 69.53±8.23 | 23.99±5.66 | 21.40±2.82 |
| | | BLIP2 | 66.60±15.35 | 69.44±3.18 | 62.49±3.90 | 17.69±1.77 | 12.41±8.95 | 2.65±3.05 | 6.94±2.41 | 91.70±4.73 | 72.83±17.83 | 60.42±17.01 | 23.32±2.28 | 20.88±3.56 |
| | | BiomedCLIP | 94.61±1.35 | 88.19±1.20 | 25.18±3.28 | 6.43±0.24 | 2.53±0.41 | 4.10±1.88 | 8.33±4.17 | 92.17±3.30 | 94.96±0.84 | 94.27±2.39 | 7.94±1.84 | 10.67±2.88 |
| | | CLIP | 64.51±14.24 | 68.75±3.61 | 65.51±8.40 | 21.79±4.29 | 11.79±9.19 | 5.69±0.61 | 5.56±2.41 | 93.91±5.46 | 62.75±22.71 | 65.89±7.09 | 27.04±4.02 | 28.49±3.78 |
| | | MedCLIP | 64.38±18.56 | 73.61±6.70 | 61.69±4.93 | 18.52±3.10 | 12.30±3.22 | 5.03±4.51 | 6.94±2.41 | 91.90±3.61 | 68.35±20.86 | 61.98±16.09 | 22.25±2.71 | 27.28±1.88 |
| | | PubMedCLIP | 87.00±2.37 | 78.47±2.41 | 47.34±1.97 | 15.74±3.49 | 3.07±1.16 | 4.45±3.79 | 25.00±0.00 | 71.99±2.84 | 88.52±4.63 | 86.72±2.07 | 18.83±2.92 | 20.16±3.60 |

Table 8: Classification results for CheXpert, FairVLMed10k, and GF3300 datasets, *sex* as the sensitive attribute. All experiments are repeated three times and mean ± std are reported (%).

| Dataset | Usage | Model | AUC | ACC | BCE | ECE | ΔAUC | ΔECE | DP | EOD | AUC$_{Male}$ | AUC$_{Female}$ | ECE$_{Male}$ | ECE$_{Female}$ |
|---|---|---|---|---|---|---|---|---|---|---|---|---|---|---|
| CheXpert | Linear Probe | BLIP | 77.90±0.00 | 81.69±0.02 | 56.10±0.00 | 31.22±0.00 | 0.76±0.00 | 2.33±0.00 | 3.27±0.03 | 5.28±0.12 | 78.36±0.00 | 77.60±0.00 | 32.38±0.00 | 30.05±0.00 |
| | | BLIP2 | 82.91±0.00 | 84.55±0.04 | 51.63±0.00 | 28.48±0.00 | 0.07±0.00 | 0.50±0.00 | 1.20±0.01 | 2.50±0.04 | 82.96±0.00 | 82.88±0.00 | 28.72±0.00 | 28.23±0.00 |
| | | BiomedCLIP | 83.60±0.00 | 84.81±0.02 | 50.30±0.00 | 26.29±0.00 | 0.77±0.00 | 1.59±0.01 | 1.72±0.02 | 2.16±0.02 | 83.23±0.00 | 84.01±0.00 | 27.08±0.01 | 25.49±0.00 |
| | | C2L | 81.88±0.00 | 84.94±0.00 | 65.53±0.00 | 38.26±0.00 | 1.45±0.00 | 0.11±0.00 | 0.04±0.00 | 2.00±0.00 | 81.18±0.00 | 82.63±0.00 | 38.31±0.00 | 38.20±0.00 |
| | | CLIP | 76.61±0.00 | 81.01±0.00 | 59.32±0.00 | 34.16±0.00 | 0.91±0.00 | 2.65±0.00 | 2.71±0.02 | 4.83±0.05 | 77.25±0.00 | 76.34±0.00 | 35.49±0.00 | 32.84±0.00 |
| | | DINOv2 | 83.49±0.00 | 85.54±0.02 | 50.12±0.01 | 27.16±0.01 | 0.55±0.01 | 0.73±0.01 | 0.77±0.02 | 1.16±0.02 | 83.22±0.00 | 83.78±0.01 | 26.79±0.01 | 27.52±0.01 |
| | | MedCLIP | 89.77±0.00 | 89.33±0.02 | 43.04±0.14 | 21.90±0.09 | 1.11±0.00 | 0.23±0.01 | 1.04±0.01 | 2.77±0.04 | 89.21±0.00 | 90.32±0.00 | 21.78±0.10 | 22.01±0.09 |
| | | MedLVM | 83.15±0.01 | 86.44±0.01 | 51.27±0.02 | 28.03±0.01 | 0.76±0.00 | 0.81±0.02 | 1.21±0.03 | 1.40±0.06 | 82.78±0.01 | 83.54±0.01 | 28.44±0.00 | 27.63±0.02 |
| | | MedMAE | 84.39±0.00 | 86.30±0.45 | 51.14±0.01 | 28.67±0.01 | 1.08±0.00 | 0.00±0.00 | 1.63±0.01 | 2.05±0.07 | 83.87±0.00 | 84.96±0.00 | 28.67±0.00 | 28.67±0.01 |
| | | MoCo-CXR | 82.04±0.22 | 84.61±0.52 | 54.27±0.19 | 30.96±0.16 | 0.69±0.09 | 2.64±0.26 | 5.40±1.24 | 7.81±1.89 | 81.85±0.28 | 82.53±0.20 | 32.28±0.26 | 29.64±0.07 |
| | | PubMedCLIP | 78.51±0.00 | 83.92±0.17 | 56.81±0.00 | 32.38±0.00 | 0.38±0.00 | 2.33±0.00 | 2.13±0.03 | 3.77±0.14 | 78.81±0.00 | 78.43±0.00 | 33.55±0.00 | 31.22±0.00 |
| | CLIP-ZS | BLIP | 56.08±0.00 | 38.14±0.00 | 79.90±0.00 | 45.73±0.00 | 2.44±0.00 | 0.62±0.00 | 5.89±0.00 | 96.38±0.00 | 57.33±0.00 | 54.89±0.00 | 45.42±0.00 | 46.04±0.00 |
| | | BLIP2 | 63.65±0.00 | 64.14±1.41 | 49.85±0.00 | 25.71±0.00 | 3.57±0.01 | 0.27±0.00 | 9.11±0.08 | 88.19±0.35 | 61.87±0.00 | 65.44±0.01 | 25.58±0.00 | 25.85±0.00 |
| | | BiomedCLIP | 72.50±2.33 | 81.90±2.47 | 155.70±8.82 | 48.73±3.22 | 2.62±0.52 | 4.87±1.17 | 2.26±1.20 | 97.19±1.58 | 71.20±2.46 | 73.82±2.24 | 46.32±3.77 | 51.18±2.61 |
| | | CLIP | 54.05±0.00 | 29.15±0.00 | 120.38±0.00 | 61.80±0.00 | 1.61±0.00 | 0.66±0.00 | 2.97±0.00 | 96.74±0.00 | 53.18±0.00 | 54.79±0.00 | 61.47±0.00 | 62.13±0.00 |
| | | MedCLIP | 48.27±14.32 | 30.77±35.66 | 72.37±7.10 | 41.48±4.02 | 3.71±1.01 | 1.09±0.29 | 4.25±7.29 | 94.68±9.16 | 47.61±16.65 | 49.23±12.60 | 42.02±3.88 | 40.93±4.16 |
| | | PubMedCLIP | 67.45±14.00 | 76.79±0.00 | 66.14±0.00 | 38.26±0.00 | 0.81±0.00 | 0.57±0.00 | 3.78±0.00 | 95.85±0.00 | 67.03±0.00 | 67.84±0.00 | 37.97±0.00 | 38.54±0.00 |
| | CLIP-Adapt | BLIP | 79.94±0.00 | 85.22±0.03 | 27.04±0.00 | 0.75±0.01 | 0.36±0.00 | 0.05±0.00 | 0.37±0.01 | 99.25±0.01 | 79.77±0.00 | 80.12±0.00 | 0.78±0.00 | 0.83±0.00 |
| | | BLIP2 | 81.65±0.74 | 85.32±0.51 | 26.43±0.43 | 1.14±0.46 | 0.55±0.26 | 0.25±0.12 | 0.33±0.12 | 99.54±0.20 | 81.37±0.61 | 81.92±0.87 | 1.21±0.53 | 1.24±0.22 |
| | | BiomedCLIP | 84.07±0.00 | 86.06±0.31 | 25.20±0.00 | 0.68±0.01 | 0.88±0.00 | 0.37±0.07 | 1.51±0.09 | 98.00±0.07 | 83.65±0.00 | 84.53±0.00 | 0.53±0.03 | 0.91±0.04 |
| | | CLIP | 81.23±0.00 | 84.83±0.03 | 26.52±0.00 | 0.76±0.01 | 0.16±0.00 | 0.14±0.01 | 0.87±0.04 | 98.00±0.04 | 81.17±0.00 | 81.32±0.00 | 0.85±0.00 | 0.72±0.01 |
| | | MedCLIP | 81.50±0.10 | 85.35±0.64 | 26.39±0.05 | 0.69±0.06 | 0.56±0.18 | 0.27±0.01 | 0.20±0.08 | 99.44±0.10 | 81.21±0.18 | 81.78±0.07 | 0.63±0.10 | 0.91±0.10 |
| | | PubMedCLIP | 82.37±0.00 | 85.27±0.01 | 25.92±0.00 | 0.55±0.01 | 0.54±0.00 | 0.21±0.01 | 1.50±0.01 | 97.47±0.04 | 82.12±0.00 | 82.66±0.00 | 0.70±0.01 | 0.91±0.01 |
| FairVLMed10k | Linear Probe | BLIP | 69.90±0.03 | 61.47±0.87 | 63.18±0.01 | 3.19±0.06 | 5.77±0.07 | 4.28±0.33 | 2.18±0.64 | 3.04±0.99 | 72.82±0.05 | 67.05±0.03 | 6.61±0.30 | 2.33±0.05 |
| | | BLIP2 | 74.14±0.00 | 64.97±1.66 | 59.88±0.02 | 2.94±0.13 | 3.27±0.04 | 2.35±0.19 | 1.53±0.13 | 1.76±0.78 | 75.78±0.01 | 72.52±0.03 | 5.65±0.07 | 3.30±0.10 |
| | | BiomedCLIP | 71.33±0.02 | 59.58±0.77 | 61.99±0.02 | 3.27±0.62 | 5.03±0.10 | 1.93±0.47 | 2.37±0.54 | 2.26±0.52 | 73.86±0.08 | 68.83±0.02 | 2.76±0.32 | 4.69±0.31 |
| | | C2L | 69.66±0.01 | 60.02±0.05 | 66.33±0.01 | 10.84±0.07 | 3.15±0.01 | 1.96±0.30 | 3.09±0.11 | 2.96±0.11 | 71.24±0.02 | 68.09±0.01 | 12.10±0.15 | 10.14±0.10 |
| | | CLIP | 68.47±0.06 | 63.12±0.26 | 64.23±0.02 | 5.77±0.15 | 4.26±0.02 | 1.64±0.74 | 3.78±0.64 | 3.57±0.64 | 70.51±0.07 | 66.25±0.06 | 7.09±0.48 | 5.44±0.47 |
| | | DINOv2 | 76.63±0.05 | 66.02±1.03 | 57.33±0.02 | 2.87±0.35 | 1.45±0.11 | 0.51±0.49 | 1.49±0.23 | 1.46±0.08 | 77.34±0.04 | 75.89±0.10 | 3.86±0.04 | 3.56±0.54 |
| | | MedCLIP | 72.44±0.09 | 63.90±1.23 | 62.05±0.04 | 5.57±0.14 | 4.94±0.09 | 2.18±0.61 | 3.28±0.63 | 3.06±0.60 | 74.91±0.10 | 69.98±0.11 | 7.34±0.25 | 5.16±0.40 |
| | | MedLVM | 73.12±0.01 | 64.76±0.48 | 60.90±0.01 | 2.95±0.48 | 4.42±0.21 | 2.20±0.88 | 1.11±0.13 | 3.92±0.92 | 75.32±0.11 | 70.90±0.09 | 3.50±0.16 | 5.70±0.83 |
| | | MedMAE | 68.72±0.01 | 60.92±0.38 | 64.29±0.01 | 4.73±0.14 | 2.17±0.06 | 3.00±0.14 | 3.51±0.74 | 3.35±0.73 | 69.76±0.04 | 67.59±0.03 | 6.50±0.04 | 3.50±0.15 |
| | | MoCo-CXR | 69.22±0.45 | 61.72±0.82 | 64.26±0.25 | 5.52±0.49 | 3.70±1.08 | 3.01±0.21 | 3.44±1.16 | 3.51±1.00 | 70.99±0.84 | 67.29±0.53 | 7.48±0.25 | 4.48±0.38 |
| | | PubMedCLIP | 67.84±0.07 | 60.94±0.08 | 64.44±0.02 | 5.38±0.23 | 4.55±0.02 | 2.30±0.56 | 5.23±0.07 | 5.05±0.06 | 70.08±0.07 | 65.53±0.07 | 6.76±0.57 | 4.46±0.12 |
| | CLIP-ZS | BLIP | 44.44±0.00 | 51.49±0.00 | 81.84±0.00 | 19.67±0.00 | 4.48±0.00 | 3.13±0.00 | 0.00±0.00 | 99.54±0.00 | 42.18±0.00 | 46.67±0.00 | 21.20±0.00 | 18.06±0.00 |
| | | BLIP2 | 43.83±0.01 | 51.32±0.00 | 78.02±0.00 | 18.05±0.00 | 2.95±0.01 | 0.61±0.04 | 0.11±0.00 | 99.89±0.00 | 42.47±0.01 | 45.42±0.01 | 18.23±0.00 | 18.84±0.04 |
| | | BiomedCLIP | 54.33±2.39 | 51.32±0.00 | 110.30±25.16 | 29.48±9.47 | 2.12±1.58 | 2.07±1.42 | 0.11±0.00 | 99.89±0.00 | 53.24±2.94 | 55.37±1.89 | 30.59±10.08 | 28.52±8.93 |
| | | CLIP | 60.19±0.00 | 52.52±0.00 | 72.61±0.00 | 15.13±0.00 | 0.28±0.00 | 0.40±0.00 | 0.00±0.00 | 99.51±0.00 | 60.04±0.00 | 60.33±0.00 | 15.03±0.00 | 15.43±0.00 |
| | | MedCLIP | 54.23±6.33 | 51.49±0.15 | 69.91±0.27 | 6.51±2.53 | 1.91±0.97 | 0.37±0.54 | 0.15±0.07 | 99.77±0.11 | 54.30±7.19 | 54.20±5.65 | 7.39±3.30 | 7.10±2.69 |
| | | PubMedCLIP | 50.02±0.00 | 51.32±0.00 | 74.95±0.00 | 14.49±0.00 | 0.49±0.00 | 1.20±0.00 | 0.11±0.00 | 99.89±0.00 | 49.81±0.00 | 50.30±0.00 | 13.84±0.00 | 15.04±0.00 |
| | CLIP-Adapt | BLIP | 75.35±0.01 | 67.60±0.35 | 58.98±0.01 | 1.62±0.04 | 3.81±0.01 | 0.23±0.19 | 0.61±0.13 | 96.02±0.36 | 77.29±0.02 | 73.48±0.01 | 3.64±0.21 | 3.41±0.05 |
| | | BLIP2 | 73.28±0.13 | 67.16±0.44 | 60.80±0.10 | 3.48±0.49 | 3.14±0.42 | 0.78±0.66 | 1.49±0.30 | 98.16±0.49 | 74.81±0.33 | 71.67±0.12 | 4.66±0.16 | 4.88±1.15 |
| | | BiomedCLIP | 72.62±0.08 | 65.71±0.75 | 61.15±0.08 | 3.53±0.05 | 3.05±0.16 | 0.73±0.61 | 2.48±0.17 | 97.73±0.17 | 74.11±0.04 | 71.06±0.16 | 4.51±0.29 | 3.78±0.35 |
| | | CLIP | 74.38±0.02 | 65.23±0.06 | 59.60±0.02 | 2.83±0.29 | 1.91±0.05 | 1.40±0.31 | 0.84±0.24 | 97.28±0.18 | 75.32±0.05 | 73.41±0.00 | 4.28±0.12 | 2.88±0.24 |
| | | MedCLIP | 75.29±0.41 | 65.77±2.55 | 58.94±0.34 | 2.85±0.91 | 3.51±0.15 | 0.72±0.21 | 1.91±0.13 | 97.99±0.16 | 77.02±0.33 | 73.51±0.47 | 4.24±0.77 | 3.52±0.59 |
| | | PubMedCLIP | 73.69±0.00 | 66.65±0.34 | 60.37±0.01 | 2.36±0.06 | 3.94±0.01 | 1.47±0.12 | 2.52±0.20 | 97.34±0.17 | 75.60±0.01 | 71.66±0.01 | 3.36±0.02 | 4.83±0.11 |
| GF3300 | Linear Probe | BLIP | 77.45±0.16 | 69.47±0.21 | 57.01±0.20 | 8.48±0.80 | 2.74±0.24 | 2.30±1.56 | 4.15±0.85 | 5.28±0.84 | 78.92±0.16 | 76.18±0.22 | 8.73±1.62 | 10.62±0.28 |
| | | BLIP2 | 80.09±0.08 | 70.54±1.67 | 53.81±0.06 | 7.61±0.50 | 1.64±0.22 | 3.99±0.88 | 3.59±2.72 | 4.84±2.85 | 81.00±0.05 | 79.36±0.23 | 9.88±0.59 | 5.89±0.31 |
| | | BiomedCLIP | 80.13±0.18 | 75.25±0.50 | 52.97±0.34 | 7.67±0.38 | 2.44±0.31 | 2.62±0.31 | 5.16±0.70 | 6.63±0.68 | 81.64±0.29 | 79.19±0.08 | 9.94±0.43 | 7.32±0.54 |
| | | C2L | 82.33±0.21 | 72.28±0.08 | 63.19±0.27 | 20.46±0.01 | 0.66±0.39 | 0.65±0.24 | 1.01±0.58 | 2.35±0.55 | 82.60±0.46 | 82.08±0.17 | 21.20±0.45 | 21.23±0.23 |
| | | CLIP | 80.23±0.15 | 74.02±0.16 | 58.20±0.02 | 12.55±0.02 | 2.22±0.14 | 1.42±0.31 | 7.41±0.00 | 8.96±0.00 | 79.23±0.23 | 81.45±0.09 | 12.65±0.15 | 14.07±0.40 |
| | | DINOv2 | 80.79±0.45 | 74.24±0.24 | 53.02±0.49 | 6.64±0.23 | 1.67±0.24 | 3.77±0.83 | 2.92±0.97 | 1.96±0.35 | 81.69±0.63 | 80.02±0.39 | 9.14±0.25 | 5.38±0.92 |
| | | MedCLIP | 76.39±0.15 | 67.23±0.40 | 62.09±0.00 | 10.43±0.17 | 1.58±0.03 | 3.74±0.94 | 1.23±0.19 | 2.82±0.81 | 77.26±0.15 | 75.68±0.12 | 12.63±0.45 | 8.89±0.31 |
| | | MedLVM | 82.54±0.03 | 75.31±0.40 | 49.73±0.07 | 5.61±0.13 | 1.42±0.04 | 0.54±0.60 | 3.03±0.58 | 4.65±0.48 | 83.40±0.03 | 81.98±0.07 | 7.71±0.19 | 7.17±0.30 |
| | | MedMAE | 80.60±0.01 | 71.04±0.00 | 54.78±0.04 | 10.65±0.14 | 2.14±0.04 | 2.51±0.51 | 6.62±0.39 | 7.95±0.38 | 81.73±0.02 | 79.59±0.05 | 12.72±0.16 | 10.21±0.26 |
| | | MoCo-CXR | 82.34±0.19 | 74.45±0.48 | 55.42±0.81 | 12.84±1.52 | 0.93±0.11 | 1.98±0.99 | 3.48±0.39 | 5.03±0.39 | 82.51±0.75 | 82.28±0.36 | 13.74±0.24 | 12.31±1.83 |
| | | PubMedCLIP | 78.40±0.01 | 64.42±0.21 | 57.33±0.03 | 11.86±0.17 | 2.86±0.02 | 3.72±1.22 | 5.27±0.19 | 6.12±0.18 | 79.97±0.01 | 77.11±0.01 | 13.84±0.47 | 10.11±0.53 |
| | CLIP-ZS | BLIP | 42.47±0.00 | 50.84±0.00 | 107.25±0.00 | 33.83±0.00 | 0.95±0.00 | 0.36±0.00 | 0.34±0.00 | 99.68±0.00 | 41.88±0.00 | 42.83±0.00 | 34.24±0.00 | 33.88±0.00 |
| | | BLIP2 | 42.75±0.01 | 50.84±0.00 | 81.84±0.01 | 20.87±0.00 | 1.31±0.07 | 3.09±0.00 | 0.34±0.00 | 99.68±0.00 | 43.44±0.05 | 42.13±0.02 | 19.26±0.00 | 22.35±0.00 |
| | | BiomedCLIP | 56.65±1.90 | 50.84±0.00 | 214.87±67.59 | 43.02±5.68 | 1.98±1.51 | 1.82±0.79 | 0.34±0.00 | 99.67±0.01 | 55.73±2.50 | 57.72±1.65 | 44.18±6.17 | 42.36±5.51 |
| | | CLIP | 70.64±0.00 | 62.79±0.00 | 215.28±0.00 | 47.68±0.00 | 1.63±0.00 | 3.05±0.00 | 3.37±0.00 | 95.83±0.00 | 71.69±0.00 | 70.06±0.00 | 49.17±0.00 | 46.12±0.00 |
| | | MedCLIP | 46.27±16.46 | 54.32±5.88 | 70.70±2.52 | 12.52±4.54 | 4.42±3.74 | 2.74±3.31 | 2.81±3.99 | 96.97±4.38 | 44.98±19.35 | 47.54±13.85 | 13.92±6.77 | 11.81±2.90 |
| | | PubMedCLIP | 39.87±0.00 | 50.84±0.00 | 87.67±0.00 | 23.90±0.00 | 3.31±0.00 | 2.31±0.00 | 0.34±0.00 | 99.68±0.00 | 38.22±0.00 | 41.53±0.00 | 25.97±0.00 | 23.66±0.00 |
| | CLIP-Adapt | BLIP | 83.52±0.03 | 74.80±0.26 | 49.41±0.04 | 4.93±0.49 | 1.65±0.09 | 2.84±0.38 | 3.37±0.34 | 95.44±0.68 | 84.14±0.07 | 82.49±0.03 | 6.96±0.17 | 4.13±0.42 |
| | | BLIP2 | 71.36±7.48 | 59.93±9.50 | 61.88±6.75 | 6.89±2.12 | 5.85±3.24 | 5.04±2.04 | 2.58±0.19 | 97.70±0.75 | 74.45±5.79 | 68.59±9.02 | 10.44±2.15 | 5.40±0.38 |
| | | BiomedCLIP | 83.12±0.14 | 74.97±2.04 | 49.77±0.16 | 4.72±1.86 | 3.15±0.41 | 3.07±0.84 | 2.36±2.05 | 94.91±1.79 | 81.64±0.28 | 84.79±0.16 | 5.88±1.24 | 7.17±2.78 |
| | | CLIP | 82.82±0.03 | 72.95±0.26 | 50.19±0.04 | 6.50±0.44 | 0.46±0.07 | 0.89±0.24 | 3.03±0.89 | 98.15±0.86 | 83.02±0.01 | 82.57±0.07 | 7.36±0.54 | 6.47±0.72 |
| | | MedCLIP | 81.93±0.44 | 72.62±0.83 | 51.66±0.38 | 5.95±1.88 | 1.23±0.36 | 3.53±2.79 | 3.25±1.85 | 95.38±1.87 | 82.60±0.64 | 81.38±0.34 | 8.29±1.60 | 4.75±1.20 |
| | | PubMedCLIP | 80.57±0.03 | 71.21±0.29 | 52.70±0.04 | 4.32±0.23 | 2.10±0.08 | 1.91±0.61 | 5.39±1.21 | 93.34±1.26 | 81.75±0.05 | 79.64±0.04 | 6.76±0.28 | 4.86±0.60 |

Table 9: Classification results for HAM10000, MIMIC-CXR, and PAPILA datasets, *sex* as the sensitive attribute. All experiments are repeated three times and mean ± std are reported (%).

| Dataset | Usage | Model | AUC | ACC | BCE | ECE | ΔAUC | ΔECE | DP | EOD | AUC$_{Male}$ | AUC$_{Female}$ | ECE$_{Male}$ | ECE$_{Female}$ |
|---|---|---|---|---|---|---|---|---|---|---|---|---|---|---|
| HAM10000 | Linear Probe | BLIP | 86.25±0.01 | 81.10±0.11 | 48.81±0.04 | 24.26±0.04 | 0.11±0.06 | 1.58±0.05 | 11.71±0.35 | 7.75±0.87 | 85.77±0.04 | 85.85±0.08 | 23.47±0.02 | 25.05±0.06 |
| | | BLIP2 | 90.14±0.04 | 86.78±0.53 | 43.54±0.04 | 20.72±0.05 | 2.34±0.06 | 0.66±0.06 | 9.74±0.61 | 12.05±0.19 | 90.79±0.04 | 88.44±0.07 | 21.04±0.05 | 20.39±0.07 |
| | | BiomedCLIP | 85.70±0.08 | 80.99±0.75 | 49.13±0.11 | 21.21±0.14 | 2.83±0.08 | 0.78±0.10 | 8.18±0.83 | 7.92±1.34 | 86.52±0.14 | 83.69±0.10 | 20.81±0.14 | 21.60±0.16 |
| | | C2L | 78.72±0.02 | 71.42±1.91 | 62.84±0.30 | 33.12±0.21 | 3.06±0.12 | 6.01±0.04 | 8.73±0.57 | 8.59±0.75 | 79.43±0.06 | 76.37±0.06 | 30.11±0.19 | 36.12±0.23 |
| | | CLIP | 84.85±0.06 | 82.64±1.16 | 51.81±0.07 | 26.05±0.04 | 0.23±0.05 | 1.80±0.01 | 9.64±1.80 | 6.95±1.63 | 84.14±0.08 | 84.37±0.08 | 25.15±0.03 | 26.95±0.04 |
| | | DINOv2 | 85.88±0.11 | 84.02±0.55 | 46.19±0.05 | 20.65±0.06 | 3.48±0.15 | 0.22±0.02 | 9.45±0.11 | 14.50±1.40 | 87.03±0.06 | 83.54±0.19 | 20.76±0.05 | 20.54±0.06 |
| | | MedCLIP | 81.09±0.22 | 77.45±1.21 | 56.21±0.09 | 28.60±0.10 | 6.44±0.27 | 3.01±0.20 | 9.81±1.93 | 13.06±3.18 | 83.11±0.29 | 76.66±0.09 | 27.09±0.15 | 30.10±0.10 |
| | | MedLVM | 87.26±0.06 | 85.57±0.38 | 47.50±0.10 | 22.95±0.07 | 2.80±0.07 | 0.11±0.04 | 13.28±0.17 | 18.11±0.54 | 87.99±0.06 | 85.19±0.13 | 22.89±0.07 | 23.01±0.07 |
| | | MedMAE | 85.34±0.03 | 82.88±0.32 | 50.20±0.01 | 24.97±0.00 | 4.78±0.03 | 2.91±0.03 | 9.16±0.07 | 14.23±0.45 | 86.77±0.02 | 81.99±0.04 | 23.51±0.01 | 26.42±0.02 |
| | | MoCo-CXR | 82.92±1.10 | 79.27±1.43 | 55.62±0.85 | 28.64±0.37 | 5.97±1.78 | 1.15±0.65 | 12.42±1.15 | 16.73±2.81 | 84.87±0.64 | 78.90±2.14 | 28.07±0.28 | 29.22±0.58 |
| | | PubMedCLIP | 85.89±0.09 | 79.90±0.16 | 51.26±0.08 | 26.13±0.03 | 1.93±0.01 | 2.25±0.05 | 11.98±0.34 | 7.24±0.27 | 84.62±0.10 | 86.55±0.09 | 25.00±0.02 | 27.25±0.05 |
| | CLIP-ZS | BLIP | 42.85±0.00 | 17.56±0.00 | 70.89±0.00 | 33.43±0.00 | 4.59±0.00 | 13.28±0.00 | 3.88±0.00 | 96.85±0.00 | 46.79±0.00 | 42.20±0.00 | 26.78±0.00 | 40.06±0.00 |
| | | BLIP2 | 44.36±0.01 | 14.47±0.00 | 59.90±0.00 | 25.68±0.00 | 0.73±0.00 | 9.76±0.00 | 0.11±0.00 | 99.70±0.00 | 45.42±0.01 | 44.69±0.00 | 21.01±0.00 | 30.77±0.00 |
| | | BiomedCLIP | 67.26±5.49 | 64.84±7.75 | 112.30±31.30 | 45.56±8.66 | 1.19±0.25 | 5.89±1.59 | 3.61±3.27 | 96.90±1.49 | 66.71±5.34 | 67.90±5.42 | 42.61±7.87 | 48.49±9.44 |
| | | CLIP | 72.05±0.00 | 74.16±0.00 | 73.86±0.00 | 37.55±0.00 | 5.19±0.00 | 5.03±0.00 | 6.16±0.00 | 91.98±0.00 | 73.58±0.00 | 68.39±0.00 | 35.03±0.00 | 40.06±0.00 |
| | | MedCLIP | 54.11±11.53 | 32.91±29.79 | 69.05±4.07 | 35.16±3.24 | 8.29±3.72 | 7.87±0.86 | 1.81±1.91 | 97.92±2.10 | 56.11±13.81 | 51.34±9.71 | 31.23±3.56 | 39.10±2.91 |
| | | PubMedCLIP | 62.38±0.00 | 63.78±0.00 | 99.15±0.00 | 50.08±0.00 | 8.29±0.00 | 2.05±0.00 | 16.58±0.00 | 79.98±0.00 | 64.06±0.00 | 55.77±0.00 | 49.06±0.00 | 51.11±0.00 |
| | CLIP-Adapt | BLIP | 87.44±0.01 | 82.35±0.11 | 28.60±0.01 | 1.64±0.06 | 2.12±0.02 | 0.39±0.08 | 10.53±0.23 | 89.39±0.43 | 87.88±0.01 | 85.76±0.01 | 2.80±0.18 | 2.40±0.10 |
| | | BLIP2 | 86.97±1.26 | 81.00±1.05 | 29.05±1.23 | 2.10±0.39 | 2.55±0.77 | 0.70±0.48 | 8.15±1.09 | 91.93±0.28 | 87.65±1.54 | 85.09±0.78 | 3.48±0.44 | 2.78±0.18 |
| | | BiomedCLIP | 87.12±0.03 | 84.63±3.38 | 28.89±0.02 | 1.92±0.09 | 0.79±0.16 | 1.32±0.67 | 6.98±1.37 | 94.58±0.98 | 87.16±0.11 | 86.37±0.06 | 3.80±0.62 | 2.47±0.10 |
| | | CLIP | 87.94±0.00 | 84.17±0.11 | 28.19±0.00 | 1.58±0.04 | 2.84±0.01 | 0.43±0.09 | 7.39±0.06 | 91.13±0.04 | 88.83±0.00 | 86.00±0.01 | 3.41±0.05 | 2.98±0.05 |
| | | MedCLIP | 84.80±0.06 | 83.80±0.75 | 30.77±0.08 | 1.84±0.32 | 0.67±0.20 | 0.86±0.20 | 10.70±1.53 | 89.25±3.01 | 84.12±0.27 | 84.49±0.44 | 3.47±0.23 | 2.61±0.28 |
| | | PubMedCLIP | 86.93±0.01 | 83.25±0.11 | 29.59±0.01 | 2.79±0.09 | 2.03±0.03 | 1.27±0.06 | 11.30±0.17 | 89.60±0.71 | 87.37±0.02 | 85.34±0.01 | 4.14±0.11 | 2.87±0.05 |
| MIMIC-CXR | Linear Probe | BLIP | 76.14±0.00 | 68.45±0.01 | 59.08±0.00 | 10.34±0.00 | 0.99±0.00 | 4.67±0.00 | 4.16±0.01 | 1.97±0.01 | 75.60±0.00 | 76.59±0.00 | 12.85±0.00 | 8.18±0.00 |
| | | BLIP2 | 80.79±0.00 | 73.45±0.00 | 54.21±0.00 | 9.30±0.00 | 1.84±0.00 | 1.47±0.00 | 8.23±0.01 | 5.54±0.01 | 79.69±0.00 | 81.53±0.00 | 10.04±0.00 | 8.57±0.00 |
| | | BiomedCLIP | 81.25±0.00 | 73.58±0.45 | 53.18±0.00 | 8.53±0.00 | 1.58±0.00 | 3.91±0.00 | 3.25±0.12 | 1.17±0.14 | 80.39±0.00 | 81.97±0.00 | 10.49±0.00 | 6.58±0.00 |
| | | C2L | 80.20±0.00 | 72.12±0.00 | 66.91±0.00 | 22.24±0.00 | 1.53±0.00 | 2.22±0.00 | 5.06±0.00 | 2.61±0.00 | 79.30±0.00 | 80.83±0.00 | 23.49±0.00 | 21.27±0.00 |
| | | CLIP | 74.83±0.00 | 66.41±0.01 | 61.69±0.00 | 11.95±0.00 | 2.48±0.00 | 2.48±0.00 | 4.27±0.01 | 2.52±0.00 | 73.39±0.00 | 75.87±0.00 | 13.49±0.00 | 10.60±0.01 |
| | | DINOv2 | 81.00±0.00 | 72.68±0.00 | 53.90±0.00 | 9.13±0.00 | 1.48±0.00 | 1.44±0.00 | 8.10±0.01 | 5.32±0.01 | 80.10±0.00 | 81.58±0.00 | 9.85±0.00 | 8.41±0.00 |
| | | MedCLIP | 89.38±0.00 | 82.23±0.01 | 41.98±0.01 | 7.61±0.01 | 0.27±0.00 | 0.22±0.00 | 7.57±0.01 | 3.64±0.01 | 89.17±0.00 | 89.44±0.00 | 7.72±0.01 | 7.50±0.01 |
| | | MedLVM | 80.68±0.00 | 72.06±0.03 | 54.33±0.00 | 9.37±0.01 | 1.60±0.00 | 2.39±0.00 | 7.06±0.02 | 4.41±0.02 | 79.72±0.00 | 81.32±0.00 | 10.57±0.00 | 8.17±0.01 |
| | | MedMAE | 80.73±0.00 | 72.88±0.09 | 54.65±0.00 | 9.97±0.00 | 1.83±0.00 | 1.41±0.01 | 6.91±0.02 | 4.35±0.01 | 79.66±0.00 | 81.50±0.00 | 10.69±0.00 | 9.28±0.01 |
| | | MoCo-CXR | 79.50±0.37 | 72.07±0.57 | 56.87±0.33 | 10.92±0.33 | 1.45±0.10 | 2.94±0.18 | 2.50±1.74 | 1.39±0.47 | 78.77±0.37 | 80.22±0.43 | 12.94±0.41 | 10.00±0.37 |
| | | PubMedCLIP | 77.46±0.00 | 70.43±0.00 | 58.64±0.00 | 10.75±0.00 | 1.67±0.00 | 3.03±0.00 | 5.68±0.00 | 3.33±0.00 | 76.47±0.00 | 78.14±0.00 | 12.49±0.00 | 9.45±0.00 |
| | CLIP-ZS | BLIP | 52.54±0.00 | 39.07±0.00 | 73.32±0.00 | 18.43±0.00 | 0.63±0.00 | 4.09±0.00 | 1.55±0.00 | 98.62±0.00 | 51.51±0.00 | 52.15±0.00 | 20.47±0.00 | 16.38±0.00 |
| | | BLIP2 | 58.68±0.00 | 40.84±0.00 | 65.72±0.00 | 3.84±0.00 | 5.41±0.00 | 1.25±0.00 | 3.48±0.00 | 96.69±0.00 | 55.32±0.00 | 60.73±0.00 | 5.73±0.00 | 4.48±0.00 |
| | | BiomedCLIP | 64.97±2.11 | 49.02±2.59 | 137.02±13.53 | 32.02±2.62 | 0.70±0.65 | 0.60±0.43 | 7.38±1.46 | 93.70±1.26 | 64.36±1.82 | 65.06±2.33 | 32.33±2.82 | 31.74±2.45 |
| | | CLIP | 54.08±0.00 | 37.87±0.00 | 97.19±0.00 | 36.55±0.00 | 0.71±0.00 | 5.23±0.00 | 0.00±0.00 | 99.95±0.00 | 53.47±0.00 | 54.18±0.00 | 39.16±0.00 | 33.93±0.00 |
| | | MedCLIP | 50.03±7.72 | 38.97±2.09 | 71.73±3.74 | 15.70±5.27 | 0.63±0.41 | 6.29±0.11 | 2.02±3.49 | 98.07±3.33 | 50.36±7.96 | 50.74±8.58 | 19.04±4.93 | 12.74±4.99 |
| | | PubMedCLIP | 64.18±0.00 | 51.44±0.00 | 67.26±0.00 | 12.47±0.00 | 0.06±0.00 | 5.12±0.00 | 3.50±0.00 | 97.79±0.00 | 63.97±0.00 | 64.04±0.00 | 15.03±0.00 | 9.91±0.00 |
| | CLIP-Adapt | BLIP | 79.15±0.00 | 71.68±0.03 | 53.30±0.00 | 2.22±0.01 | 1.51±0.00 | 0.19±0.16 | 6.94±0.01 | 95.49±0.01 | 78.26±0.00 | 79.77±0.00 | 2.46±0.12 | 2.65±0.05 |
| | | BLIP2 | 80.89±0.07 | 72.99±0.12 | 51.59±0.07 | 2.64±0.05 | 1.36±0.02 | 0.07±0.05 | 8.53±0.01 | 94.22±0.01 | 80.04±0.05 | 81.40±0.08 | 2.68±0.16 | 2.67±0.06 |
| | | BiomedCLIP | 81.59±0.00 | 74.20±0.16 | 50.80±0.00 | 1.86±0.01 | 1.17±0.00 | 0.17±0.06 | 4.20±0.03 | 98.53±0.02 | 80.93±0.00 | 82.10±0.00 | 2.72±0.04 | 2.54±0.02 |
| | | CLIP | 80.62±0.00 | 73.06±0.01 | 51.80±0.00 | 2.11±0.02 | 1.64±0.00 | 0.27±0.19 | 6.49±0.02 | 96.09±0.03 | 79.64±0.00 | 81.28±0.00 | 2.42±0.13 | 2.15±0.06 |
| | | MedCLIP | 80.21±0.08 | 72.35±0.65 | 52.15±0.08 | 1.97±0.06 | 1.57±0.09 | 0.06±0.03 | 7.90±0.31 | 94.74±0.34 | 79.24±0.12 | 80.82±0.05 | 2.12±0.05 | 2.18±0.06 |
| | | PubMedCLIP | 81.51±0.00 | 73.63±0.43 | 50.77±0.00 | 1.66±0.01 | 1.55±0.00 | 0.33±0.10 | 6.11±0.01 | 96.59±0.01 | 80.59±0.00 | 82.14±0.00 | 2.24±0.08 | 1.92±0.02 |
| PAPILA | Linear Probe | BLIP | 82.19±1.14 | 72.62±4.69 | 59.03±0.48 | 30.20±0.36 | 5.68±2.11 | 2.19±2.05 | 10.71±10.71 | 14.03±8.79 | 78.70±1.57 | 84.38±1.04 | 31.55±2.50 | 29.96±0.36 |
| | | BLIP2 | 87.98±2.95 | 88.69±2.23 | 58.01±2.32 | 33.43±2.22 | 12.37±5.02 | 3.25±3.36 | 11.90±5.46 | 18.79±18.27 | 93.62±0.50 | 81.25±5.51 | 36.02±2.40 | 34.53±3.24 |
| | | BiomedCLIP | 69.11±7.87 | 69.64±9.56 | 54.07±2.84 | 25.81±1.56 | 17.04±9.16 | 1.49±1.55 | 4.76±2.06 | 11.19±4.34 | 64.06±0.50 | 76.39±16.74 | 27.37±1.30 | 28.86±1.64 |
| | | C2L | 67.22±3.22 | 68.45±8.91 | 69.15±1.73 | 34.09±1.21 | 15.44±13.07 | 5.68±2.02 | 7.14±7.14 | 19.23±7.55 | 65.22±6.96 | 68.23±15.89 | 34.68±3.40 | 38.10±1.95 |
| | | CLIP | 82.78±9.66 | 82.74±2.23 | 65.48±0.93 | 34.19±1.84 | 4.40±1.54 | 3.95±3.75 | 14.29±12.88 | 25.07±17.66 | 84.06±11.04 | 81.42±6.47 | 38.63±1.68 | 34.69±1.41 |
| | | DINOv2 | 83.06±4.88 | 80.36±6.36 | 51.29±2.39 | 25.98±1.48 | 7.25±5.76 | 2.69±2.42 | 4.76±2.06 | 15.48±13.00 | 80.00±3.98 | 86.81±7.39 | 28.64±2.09 | 26.71±1.15 |
| | | MedCLIP | 78.68±1.54 | 79.76±4.69 | 62.94±0.40 | 32.61±1.25 | 8.50±8.35 | 6.39±1.10 | 8.33±5.46 | 20.04±12.70 | 72.75±2.19 | 81.25±6.51 | 31.67±1.06 | 38.06±0.24 |
| | | MedLVM | 81.32±0.41 | 79.17±3.04 | 58.29±0.95 | 30.53±0.90 | 6.19±3.65 | 4.64±2.04 | 2.38±2.06 | 7.90±2.28 | 82.90±3.29 | 78.12±2.76 | 30.53±3.47 | 33.07±3.16 |
| | | MedMAE | 63.28±8.36 | 54.17±12.14 | 67.32±0.33 | 32.67±0.18 | 24.23±7.72 | 2.61±2.06 | 9.52±4.12 | 17.65±3.77 | 74.93±5.08 | 50.69±12.55 | 33.21±1.80 | 35.67±1.59 |
| | | MoCo-CXR | 54.65±20.94 | 53.57±29.59 | 69.03±1.08 | 35.80±0.77 | 29.04±26.66 | 3.25±2.83 | 3.57±0.00 | 15.30±4.85 | 41.45±23.66 | 70.49±23.73 | 37.07±1.87 | 40.32±0.80 |
| | | PubMedCLIP | 73.05±5.50 | 75.00±9.56 | 65.18±0.30 | 33.77±2.54 | 20.66±4.96 | 2.53±1.44 | 5.95±5.46 | 11.22±6.92 | 64.06±5.51 | 84.72±5.35 | 33.56±3.00 | 35.47±1.78 |
| | CLIP-ZS | BLIP | 35.46±0.00 | 26.79±0.00 | 219.91±0.00 | 75.03±0.00 | 30.69±0.00 | 2.37±0.00 | 0.00±0.00 | 85.05±0.00 | 23.48±0.00 | 54.17±0.00 | 75.20±0.00 | 77.57±0.00 |
| | | BLIP2 | 69.11±0.14 | 64.29±0.00 | 310.17±0.01 | 81.43±0.00 | 26.81±0.60 | 3.60±0.00 | 3.57±0.00 | 74.93±0.00 | 54.78±0.00 | 81.60±0.60 | 79.64±0.00 | 83.23±0.00 |
| | | BiomedCLIP | 78.33±2.07 | 76.79±4.72 | 43.57±6.74 | 13.48±6.13 | 15.33±2.70 | 8.99±3.50 | 14.29±3.57 | 73.88±6.66 | 72.17±2.30 | 87.50±1.04 | 16.01±2.37 | 21.68±7.29 |
| | | CLIP | 78.25±0.00 | 85.71±0.00 | 43.52±0.00 | 16.58±0.00 | 9.86±0.00 | 1.15±0.00 | 3.57±0.00 | 95.33±0.00 | 81.74±0.00 | 71.88±0.00 | 21.26±0.00 | 20.10±0.00 |
| | | MedCLIP | 50.51±16.80 | 60.12±36.70 | 65.58±2.84 | 32.18±1.65 | 23.78±14.78 | 4.32±1.43 | 7.14±3.57 | 84.61±3.27 | 37.68±10.73 | 61.46±23.87 | 30.50±1.81 | 34.82±0.81 |
| | | PubMedCLIP | 43.26±0.00 | 33.93±0.00 | 101.25±0.00 | 51.10±0.00 | 36.74±0.00 | 1.17±0.00 | 7.14±0.00 | 78.03±0.00 | 61.74±0.00 | 25.00±0.00 | 51.80±0.00 | 52.97±0.00 |
| | CLIP-Adapt | BLIP | 84.63±0.63 | 80.95±1.03 | 33.74±0.66 | 10.52±0.93 | 24.99±1.33 | 1.77±0.29 | 1.19±2.06 | 74.16±5.72 | 71.88±1.33 | 96.88±0.00 | 19.77±0.23 | 18.00±0.35 |
| | | BLIP2 | 90.62±0.59 | 89.88±1.03 | 37.73±0.33 | 17.08±1.65 | 9.42±2.31 | 8.29±1.76 | 16.67±2.06 | 79.20±1.26 | 92.75±1.00 | 83.33±3.12 | 23.17±0.85 | 14.89±2.61 |
| | | BiomedCLIP | 83.85±1.66 | 86.90±1.03 | 32.91±1.14 | 7.65±3.75 | 8.72±1.38 | 3.40±1.27 | 10.71±6.19 | 73.88±8.07 | 79.13±1.51 | 87.85±2.62 | 9.01±2.01 | 12.40±3.27 |
| | | CLIP | 82.51±0.47 | 82.74±1.03 | 38.51±0.52 | 11.69±1.25 | 20.29±0.60 | 3.33±4.31 | 19.05±2.06 | 75.19±1.20 | 88.70±0.00 | 68.40±0.60 | 20.15±0.91 | 17.95±4.44 |
| | | MedCLIP | 88.49±2.12 | 86.90±2.73 | 39.30±0.34 | 17.61±2.56 | 13.69±1.28 | 0.65±0.47 | 5.95±7.43 | 81.39±13.74 | 81.45±1.33 | 95.14±2.17 | 19.31±2.48 | 19.09±3.07 |
| | | PubMedCLIP | 84.00±0.36 | 75.60±1.03 | 36.13±0.19 | 14.30±1.55 | 11.78±1.00 | 2.46±1.00 | 4.76±2.06 | 79.09±0.63 | 78.84±1.00 | 90.62±0.00 | 17.35±3.34 | 17.14±0.23 |

Table 10: Segmentation results on the FairSeg dataset with 2D FMs.

| Dataset | Model | Prompt | DSC$_{Avg}$ | DSC$_{min}$ | DSC$_{max}$ | DSC$_{\Delta}$ | DSC$_{STD}$ | DSC$_{ES}$ |
|---|---|---|---|---|---|---|---|---|
| FairSeg | SAM | center | 45.01 | 44.83 | 45.27 | 0.45 | 0.22 | 36.82 |
| | | rand | 45.22 | 44.74 | 45.87 | 1.12 | 0.56 | 28.94 |
| | | rands | 55.02 | 54.42 | 55.85 | 1.43 | 0.71 | 32.13 |
| | | bbox | 63.35 | 63.26 | 63.42 | 0.16 | 0.08 | 58.52 |
| | MobileSAM | center | 34.59 | 33.97 | 35.47 | 1.50 | 0.75 | 19.74 |
| | | rand | 32.41 | 31.84 | 33.22 | 1.38 | 0.69 | 19.21 |
| | | rands | 45.86 | 45.47 | 46.41 | 0.94 | 0.47 | 31.20 |
| | | bbox | 65.64 | 65.34 | 65.86 | 0.52 | 0.26 | 52.20 |
| | TinySAM | center | 43.47 | 43.40 | 43.55 | 0.15 | 0.07 | 40.53 |
| | | rand | 48.06 | 48.00 | 48.13 | 0.13 | 0.07 | 45.02 |
| | | rands | 59.57 | 59.44 | 59.77 | 0.33 | 0.16 | 51.24 |
| | | bbox | 67.47 | 67.14 | 67.70 | 0.56 | 0.28 | 52.60 |
| | MedSAM | center | 2.48 | 2.41 | 2.58 | 0.17 | 0.08 | 2.29 |
| | | rand | 1.72 | 1.71 | 1.74 | 0.03 | 0.02 | 1.69 |
| | | rands | 17.88 | 17.84 | 17.92 | 0.08 | 0.04 | 17.23 |
| | | bbox | 44.67 | 43.80 | 45.28 | 1.48 | 0.74 | 25.67 |
| | SAM-Med2D | center | 32.12 | 31.49 | 32.56 | 1.07 | 0.54 | 20.92 |
| | | rand | 34.29 | 33.65 | 34.75 | 1.10 | 0.55 | 22.12 |
| | | rands | 59.69 | 59.21 | 60.03 | 0.82 | 0.41 | 42.33 |
| | | bbox | 49.09 | 48.66 | 49.38 | 0.72 | 0.36 | 36.09 |
| | FT-SAM | center | 16.35 | 16.02 | 16.59 | 0.57 | 0.29 | 12.70 |
| | | rand | 13.51 | 13.16 | 13.76 | 0.60 | 0.30 | 10.39 |
| | | rands | 31.42 | 31.01 | 31.71 | 0.70 | 0.35 | 23.27 |
| | | bbox | 49.34 | 48.77 | 49.76 | 0.99 | 0.50 | 32.95 |

Table 11: Segmentation results on the HAM10000 dataset with 2D FMs.

| Dataset | Model | Prompt | DSC$_{Avg}$ | DSC$_{min}$ | DSC$_{max}$ | DSC$_{\Delta}$ | DSC$_{STD}$ | DSC$_{ES}$ |
|---|---|---|---|---|---|---|---|---|
| HAM10000 | SAM | center | 59.01 | 57.34 | 61.01 | 3.67 | 1.83 | 20.81 |
| | | rand | 51.31 | 50.87 | 51.85 | 0.98 | 0.49 | 34.44 |
| | | rands | 63.39 | 62.18 | 64.84 | 2.66 | 1.33 | 27.21 |
| | | bbox | 51.09 | 50.44 | 51.88 | 1.44 | 0.72 | 29.70 |
| | MobileSAM | center | 48.73 | 48.17 | 49.41 | 1.24 | 0.62 | 30.08 |
| | | rand | 45.51 | 45.34 | 45.64 | 0.30 | 0.15 | 39.57 |
| | | rands | 50.49 | 50.38 | 50.62 | 0.24 | 0.12 | 45.08 |
| | | bbox | 75.13 | 71.69 | 79.25 | 7.56 | 3.78 | 15.72 |
| | TinySAM | center | 58.83 | 56.19 | 61.99 | 5.80 | 2.90 | 15.08 |
| | | rand | 53.38 | 51.16 | 56.04 | 4.88 | 2.44 | 15.52 |
| | | rands | 63.71 | 61.47 | 66.39 | 4.92 | 2.46 | 18.41 |
| | | bbox | 83.46 | 82.21 | 84.96 | 2.75 | 1.38 | 35.14 |
| | MedSAM | center | 0.29 | 0.29 | 0.30 | 0.01 | 0.01 | 0.29 |
| | | rand | 0.97 | 0.95 | 0.99 | 0.04 | 0.02 | 0.95 |
| | | rands | 14.63 | 14.52 | 14.77 | 0.25 | 0.12 | 13.00 |
| | | bbox | 74.78 | 74.43 | 75.22 | 0.79 | 0.39 | 53.61 |
| | SAM-Med2D | center | 86.65 | 85.39 | 88.16 | 2.77 | 1.38 | 36.33 |
| | | rand | 87.12 | 85.80 | 88.69 | 2.89 | 1.45 | 35.63 |
| | | rands | 89.48 | 88.66 | 90.46 | 1.80 | 0.90 | 47.09 |
| | | bbox | 90.92 | 90.30 | 91.67 | 1.37 | 0.69 | 53.96 |
| | FT-SAM | center | 11.54 | 10.32 | 12.99 | 2.67 | 1.33 | 4.94 |
| | | rand | 6.22 | 5.38 | 7.23 | 1.85 | 0.93 | 3.23 |
| | | rands | 19.01 | 17.04 | 21.37 | 4.33 | 2.17 | 6.01 |
| | | bbox | 69.95 | 67.10 | 73.38 | 6.28 | 3.14 | 16.90 |

Table 12: Segmentation results on the TUSC dataset with 2D FMs.

| Dataset | Model | Prompt | $DSC_{Avg}$ | $DSC_{min}$ | $DSC_{max}$ | $DSC_{\Delta}$ | $DSC_{STD}$ | $DSC_{ES}$ |
|---------|-------|--------|-------------|-------------|-------------|-------------|-------------|------------|
| TUSC | SAM | center | 27.13 | 26.85 | 28.60 | 1.75 | 0.88 | 14.47 |
| | | rand | 26.96 | 26.52 | 29.28 | 2.76 | 1.38 | 11.33 |
| | | rands | 32.23 | 32.13 | 32.77 | 0.64 | 0.32 | 24.42 |
| | | bbox | 87.06 | 86.41 | 87.19 | 0.78 | 0.39 | 62.63 |
| | MobileSAM | center | 24.70 | 24.00 | 28.36 | 4.36 | 2.18 | 7.77 |
| | | rand | 24.81 | 24.03 | 28.93 | 4.90 | 2.45 | 7.19 |
| | | rands | 27.53 | 26.81 | 31.26 | 4.45 | 2.23 | 8.54 |
| | | bbox | 82.84 | 81.12 | 83.16 | 2.04 | 1.02 | 41.01 |
| | TinySAM | center | 27.18 | 26.93 | 28.51 | 1.58 | 0.79 | 15.18 |
| | | rand | 25.50 | 24.96 | 28.30 | 3.34 | 1.67 | 9.55 |
| | | rands | 30.69 | 30.61 | 31.15 | 0.54 | 0.27 | 24.17 |
| | | bbox | 88.33 | 87.81 | 88.43 | 0.62 | 0.31 | 67.43 |
| | MedSAM | center | 0.92 | 0.88 | 0.93 | 0.05 | 0.03 | 0.90 |
| | | rand | 1.14 | 0.56 | 1.25 | 0.69 | 0.34 | 0.85 |
| | | rands | 17.75 | 16.79 | 17.93 | 1.14 | 0.57 | 11.31 |
| | | bbox | 69.07 | 68.88 | 70.07 | 1.19 | 0.59 | 43.30 |
| | SAM-Med2D | center | 18.35 | 15.35 | 18.92 | 3.57 | 1.79 | 6.59 |
| | | rand | 14.93 | 12.81 | 15.34 | 2.53 | 1.26 | 6.59 |
| | | rands | 54.07 | 51.05 | 54.65 | 3.60 | 1.80 | 19.31 |
| | | bbox | 57.38 | 53.20 | 58.18 | 4.98 | 2.49 | 16.44 |
| | FT-SAM | center | 2.26 | 2.17 | 2.75 | 0.58 | 0.29 | 1.75 |
| | | rand | 1.71 | 1.70 | 1.76 | 0.06 | 0.03 | 1.66 |
| | | rands | 9.94 | 9.74 | 10.98 | 1.24 | 0.62 | 6.14 |
| | | bbox | 44.91 | 42.00 | 45.46 | 3.46 | 1.73 | 16.45 |

Table 13: Segmentation results on the Montgomery dataset with 2D FMs.

| Dataset | Model | Prompt | $DSC_{Avg}$ | $DSC_{min}$ | $DSC_{max}$ | $DSC_{\Delta}$ | $DSC_{STD}$ | $DSC_{ES}$ |
|---------|-------|--------|-------------|-------------|-------------|-------------|-------------|------------|
| Montgomery | SAM | center | 71.80 | 65.75 | 78.55 | 12.80 | 6.40 | 9.70 |
| | | rand | 66.33 | 59.15 | 74.33 | 15.19 | 7.59 | 7.72 |
| | | rands | 88.46 | 86.62 | 90.55 | 3.92 | 1.96 | 29.86 |
| | | bbox | 92.75 | 92.43 | 93.11 | 0.68 | 0.34 | 69.22 |
| | MobileSAM | center | 56.87 | 55.52 | 57.98 | 2.45 | 1.23 | 25.56 |
| | | rand | 51.26 | 49.48 | 52.73 | 3.24 | 1.62 | 19.54 |
| | | rands | 62.46 | 61.62 | 62.95 | 1.34 | 0.67 | 37.40 |
| | | bbox | 88.75 | 88.58 | 88.84 | 0.26 | 0.13 | 78.54 |
| | TinySAM | center | 78.62 | 77.86 | 79.33 | 1.47 | 0.73 | 45.38 |
| | | rand | 69.00 | 65.67 | 71.53 | 5.86 | 2.93 | 17.56 |
| | | rands | 79.82 | 78.71 | 80.60 | 1.89 | 0.95 | 41.04 |
| | | bbox | 90.17 | 90.08 | 90.19 | 0.11 | 0.05 | 85.47 |
| | MedSAM | center | 1.18 | 1.09 | 1.26 | 0.17 | 0.09 | 1.08 |
| | | rand | 2.65 | 1.96 | 3.11 | 1.15 | 0.58 | 1.68 |
| | | rands | 27.86 | 27.00 | 28.43 | 1.43 | 0.71 | 16.24 |
| | | bbox | 79.56 | 77.85 | 81.44 | 3.59 | 1.80 | 28.44 |
| | SAM-Med2D | center | 88.34 | 87.06 | 89.86 | 2.80 | 1.40 | 36.77 |
| | | rand | 82.50 | 78.52 | 87.44 | 8.92 | 4.46 | 15.10 |
| | | rands | 91.15 | 90.83 | 91.55 | 0.72 | 0.36 | 67.02 |
| | | bbox | 91.73 | 91.06 | 92.51 | 1.45 | 0.73 | 53.10 |
| | FT-SAM | center | 6.17 | 5.74 | 6.61 | 0.87 | 0.43 | 4.30 |
| | | rand | 4.66 | 4.58 | 4.79 | 0.20 | 0.10 | 4.24 |
| | | rands | 24.56 | 18.16 | 30.16 | 12.00 | 6.00 | 3.51 |
| | | bbox | 75.46 | 74.23 | 76.62 | 2.39 | 1.20 | 34.34 |

Table 14: Segmentation results on the CANDI dataset with 2D FMs.

| Dataset | Model | Prompt | $DSC_{Avg}$ | $DSC_{min}$ | $DSC_{max}$ | $DSC_\Delta$ | $DSC_{STD}$ | $DSC_{ES}$ |
|---|---|---|---|---|---|---|---|---|
| | SAM | center | 19.56 | 19.29 | 19.91 | 0.62 | 0.36 | 19.50 |
| | | rand | 27.50 | 27.25 | 27.84 | 0.59 | 0.48 | 27.32 |
| | | rands | 29.36 | 28.84 | 30.03 | 1.19 | 0.76 | 29.05 |
| | | bbox | 55.35 | 54.96 | 55.85 | 0.89 | 0.45 | 55.11 |
| | MobileSAM | center | 13.33 | 13.14 | 13.47 | 0.33 | 0.17 | 13.30 |
| | | rand | 13.87 | 13.61 | 14.07 | 0.47 | 0.23 | 13.85 |
| | | rands | 18.32 | 18.29 | 18.35 | 0.06 | 0.21 | 18.29 |
| | | bbox | 52.53 | 52.22 | 52.92 | 0.70 | 0.35 | 52.36 |
| | TinySAM | center | 22.03 | 21.55 | 22.65 | 1.10 | 0.55 | 21.93 |
| | | rand | 28.97 | 28.62 | 29.42 | 0.80 | 0.40 | 28.84 |
| | | rands | 24.84 | 24.46 | 25.34 | 0.87 | 0.52 | 24.72 |
| | | bbox | 52.69 | 52.45 | 52.99 | 0.54 | 0.27 | 52.55 |
| CANDI | MedSAM | center | 0.40 | 0.36 | 0.43 | 0.08 | 0.04 | 0.40 |
| | | rand | 16.19 | 16.15 | 16.23 | 0.08 | 0.06 | 16.16 |
| | | rands | 14.79 | 14.54 | 15.10 | 0.56 | 0.31 | 14.72 |
| | | bbox | 40.29 | 40.01 | 40.65 | 0.64 | 0.56 | 40.09 |
| | SAM-Med2D | center | 9.23 | 8.82 | 9.76 | 0.93 | 0.47 | 9.18 |
| | | rand | 14.68 | 14.23 | 15.27 | 1.04 | 0.52 | 14.56 |
| | | rands | 32.59 | 32.12 | 33.21 | 1.10 | 0.55 | 32.41 |
| | | bbox | 28.23 | 27.77 | 28.84 | 1.07 | 0.54 | 28.08 |
| | FT-SAM | center | 3.61 | 3.51 | 3.73 | 0.22 | 0.28 | 3.60 |
| | | rand | 8.47 | 8.39 | 8.58 | 0.19 | 0.34 | 8.43 |
| | | rands | 21.89 | 21.63 | 22.21 | 0.58 | 0.56 | 21.76 |
| | | bbox | 22.10 | 21.76 | 22.54 | 0.78 | 0.44 | 21.99 |

Table 15: Segmentation results on the IRCADb dataset with 2D FMs.

| Dataset | Model | Prompt | $DSC_{Avg}$ | $DSC_{min}$ | $DSC_{max}$ | $DSC_\Delta$ | $DSC_{STD}$ | $DSC_{ES}$ |
|---|---|---|---|---|---|---|---|---|
| | SAM | center | 26.43 | 23.59 | 28.13 | 4.54 | 6.14 | 24.73 |
| | | rand | 37.66 | 32.56 | 41.09 | 8.52 | 8.85 | 34.44 |
| | | rands | 43.10 | 39.60 | 45.27 | 5.67 | 9.51 | 39.27 |
| | | bbox | 57.51 | 54.14 | 61.45 | 7.31 | 7.54 | 53.46 |
| | MobileSAM | center | 18.24 | 16.61 | 19.32 | 2.71 | 3.27 | 17.37 |
| | | rand | 24.12 | 23.40 | 23.83 | 0.42 | 5.55 | 22.58 |
| | | rands | 29.80 | 28.21 | 29.60 | 1.39 | 7.12 | 27.38 |
| | | bbox | 55.14 | 51.45 | 58.45 | 7.01 | 6.54 | 51.84 |
| | TinySAM | center | 28.60 | 26.42 | 29.10 | 2.68 | 6.14 | 26.83 |
| | | rand | 40.95 | 36.07 | 43.84 | 7.77 | 9.69 | 37.23 |
| | | rands | 44.37 | 42.87 | 44.49 | 1.62 | 7.80 | 40.94 |
| | | bbox | 57.26 | 53.47 | 60.99 | 7.52 | 7.52 | 53.31 |
| IRCADb | MedSAM | center | 1.23 | 0.62 | 1.38 | 0.76 | 0.78 | 1.21 |
| | | rand | 1.12 | 0.85 | 1.28 | 0.44 | 0.33 | 1.12 |
| | | rands | 12.66 | 11.03 | 12.88 | 1.86 | 3.18 | 12.11 |
| | | bbox | 43.43 | 42.33 | 44.78 | 2.45 | 5.07 | 41.16 |
| | SAM-Med2D | center | 27.74 | 25.24 | 28.07 | 2.83 | 4.91 | 26.20 |
| | | rand | 36.05 | 33.23 | 37.71 | 4.48 | 3.48 | 34.59 |
| | | rands | 46.37 | 45.69 | 46.94 | 1.25 | 3.48 | 44.90 |
| | | bbox | 38.39 | 37.00 | 40.44 | 3.44 | 5.31 | 36.25 |
| | FT-SAM | center | 13.22 | 10.46 | 14.34 | 3.87 | 4.12 | 12.49 |
| | | rand | 16.68 | 12.65 | 19.16 | 6.51 | 4.48 | 15.63 |
| | | rands | 28.76 | 26.12 | 30.32 | 4.19 | 3.28 | 27.60 |
| | | bbox | 38.33 | 36.95 | 39.08 | 2.13 | 3.20 | 37.34 |

Table 16: Segmentation results on the KiTS dataset with 2D FMs.

| Dataset | Model | Prompt | DSC$_{Avg}$ | DSC$_{min}$ | DSC$_{max}$ | DSC$_\Delta$ | DSC$_{STD}$ | DSC$_{ES}$ |
|---|---|---|---|---|---|---|---|---|
| KiTS | SAM | center | 22.44 | 21.91 | 22.47 | 0.56 | 1.05 | 22.24 |
| | | rand | 34.02 | 31.62 | 35.06 | 3.44 | 1.72 | 33.42 |
| | | rands | 41.26 | 38.86 | 42.36 | 3.51 | 1.75 | 40.47 |
| | | bbox | 75.11 | 75.12 | 75.19 | 0.07 | 0.63 | 74.64 |
| | MobileSAM | center | 8.12 | 7.92 | 8.45 | 0.52 | 0.38 | 8.09 |
| | | rand | 11.22 | 11.13 | 11.22 | 0.09 | 0.36 | 11.18 |
| | | rands | 12.89 | 12.55 | 13.44 | 0.89 | 0.58 | 12.80 |
| | | bbox | 68.67 | 68.15 | 69.73 | 1.58 | 0.79 | 68.19 |
| | TinySAM | center | 19.00 | 18.29 | 19.09 | 0.80 | 1.17 | 18.80 |
| | | rand | 27.53 | 25.54 | 28.34 | 2.81 | 1.41 | 27.17 |
| | | rands | 31.82 | 31.62 | 31.86 | 0.24 | 1.06 | 31.50 |
| | | bbox | 72.19 | 71.81 | 72.99 | 1.17 | 0.59 | 71.78 |
| | MedSAM | center | 0.57 | 0.54 | 0.59 | 0.05 | 0.10 | 0.57 |
| | | rand | 1.17 | 1.13 | 1.22 | 0.09 | 0.12 | 1.16 |
| | | rands | 17.80 | 17.43 | 18.49 | 1.06 | 0.71 | 17.66 |
| | | bbox | 46.77 | 46.31 | 46.97 | 0.66 | 0.95 | 46.30 |
| | SAM-Med2D | center | 31.22 | 30.57 | 31.45 | 0.88 | 0.88 | 30.93 |
| | | rand | 40.83 | 38.02 | 42.09 | 4.07 | 2.04 | 40.05 |
| | | rands | 49.90 | 48.48 | 50.39 | 1.91 | 1.08 | 49.52 |
| | | bbox | 47.20 | 45.43 | 47.87 | 2.44 | 1.22 | 46.78 |
| | FT-SAM | center | 15.61 | 15.27 | 16.22 | 0.95 | 0.47 | 15.54 |
| | | rand | 18.60 | 18.24 | 18.73 | 0.49 | 0.27 | 18.56 |
| | | rands | 39.80 | 39.11 | 39.96 | 0.86 | 0.92 | 39.51 |
| | | bbox | 45.59 | 44.93 | 45.70 | 0.77 | 1.25 | 45.07 |

Table 17: Segmentation results on the SPIDER dataset with 2D FMs.

| Dataset | Model | Prompt | DSC$_{Avg}$ | DSC$_{min}$ | DSC$_{max}$ | DSC$_\Delta$ | DSC$_{STD}$ | DSC$_{ES}$ |
|---|---|---|---|---|---|---|---|---|
| SPIDER | SAM | center | 24.54 | 23.00 | 25.71 | 2.70 | 1.46 | 24.13 |
| | | rand | 24.62 | 23.03 | 25.82 | 2.79 | 1.40 | 24.22 |
| | | rands | 34.45 | 34.03 | 35.21 | 1.18 | 0.82 | 34.16 |
| | | bbox | 68.49 | 68.80 | 69.90 | 1.10 | 0.63 | 68.06 |
| | MobileSAM | center | 14.13 | 14.32 | 14.65 | 0.34 | 0.49 | 14.02 |
| | | rand | 13.03 | 13.01 | 13.54 | 0.53 | 0.60 | 12.91 |
| | | rands | 20.56 | 19.23 | 20.34 | 1.11 | 0.62 | 20.40 |
| | | bbox | 65.46 | 65.07 | 65.79 | 0.72 | 0.57 | 65.08 |
| | TinySAM | center | 16.87 | 16.40 | 16.82 | 0.42 | 0.65 | 16.75 |
| | | rand | 16.80 | 16.03 | 16.66 | 0.63 | 0.69 | 16.67 |
| | | rands | 27.40 | 25.82 | 26.89 | 1.07 | 0.98 | 27.16 |
| | | bbox | 67.20 | 67.11 | 67.69 | 0.58 | 0.46 | 66.90 |
| | MedSAM | center | 0.94 | 0.82 | 0.97 | 0.15 | 0.17 | 0.94 |
| | | rand | 1.06 | 0.87 | 1.01 | 0.14 | 0.15 | 1.05 |
| | | rands | 23.45 | 22.95 | 24.17 | 1.21 | 0.80 | 23.26 |
| | | bbox | 50.09 | 50.22 | 51.48 | 1.26 | 1.03 | 49.58 |
| | SAM-Med2D | center | 24.84 | 21.29 | 24.02 | 2.73 | 1.39 | 24.46 |
| | | rand | 24.05 | 20.73 | 23.49 | 2.76 | 1.41 | 23.70 |
| | | rands | 39.51 | 37.86 | 40.43 | 2.56 | 1.32 | 39.04 |
| | | bbox | 31.03 | 32.87 | 33.81 | 0.94 | 0.89 | 30.77 |
| | FT-SAM | center | 11.36 | 8.45 | 10.06 | 1.61 | 0.84 | 11.25 |
| | | rand | 10.93 | 7.98 | 9.60 | 1.62 | 0.81 | 10.82 |
| | | rands | 20.31 | 15.64 | 18.48 | 2.84 | 1.42 | 20.01 |
| | | bbox | 36.70 | 32.52 | 35.32 | 2.80 | 1.71 | 35.99 |

Table 18: Segmentation results on 3D datasets with 3D FMs.

| Dataset | Model | Prompt | $DSC_{Avg}$ | $DSC_{min}$ | $DSC_{max}$ | $DSC_{\Delta}$ | $DSC_{STD}$ | $DSC_{ES}$ |
|---------|-------|--------|-------------|-------------|-------------|----------------|-------------|------------|
| CANDI | FastSAM-3D | 1point | 16.21 | 15.47 | 17.13 | 1.66 | 0.83 | 8.87 |
| | | 5points | 29.93 | 29.43 | 30.55 | 1.12 | 0.56 | 19.21 |
| | SAM-Med3D | 1point | 21.19 | 20.66 | 21.85 | 1.19 | 0.60 | 13.29 |
| | | 5points | 25.33 | 25.20 | 25.48 | 0.29 | 0.14 | 22.15 |
| | SegVol | point | 17.92 | 17.85 | 17.97 | 0.12 | 0.06 | 16.89 |
| | | bbox | 25.92 | 25.89 | 25.95 | 0.06 | 0.03 | 25.18 |
| IRCADb | FastSAM-3D | 1point | 18.49 | 17.05 | 20.80 | 3.76 | 1.88 | 6.42 |
| | | 5points | 36.49 | 36.14 | 36.69 | 0.55 | 0.27 | 28.65 |
| | SAM-Med3D | 1point | 23.94 | 22.72 | 24.91 | 2.19 | 1.10 | 11.42 |
| | | 5points | 30.05 | 27.79 | 32.78 | 5.00 | 2.50 | 8.59 |
| | SegVol | point | 45.87 | 38.70 | 53.04 | 14.35 | 7.17 | 5.61 |
| | | bbox | 47.00 | 41.66 | 52.35 | 10.69 | 5.35 | 7.41 |
| KiTS | FastSAM-3D | 1point | 28.79 | 27.93 | 29.11 | 1.18 | 0.59 | 18.09 |
| | | 5points | 45.81 | 44.64 | 46.28 | 1.64 | 0.82 | 25.19 |
| | SAM-Med3D | 1point | 22.95 | 22.85 | 22.96 | 0.12 | 0.06 | 21.69 |
| | | 5points | 29.88 | 29.24 | 30.13 | 0.89 | 0.45 | 20.66 |
| | SegVol | point | 43.06 | 37.99 | 46.05 | 8.06 | 4.03 | 8.56 |
| | | bbox | 42.15 | 37.57 | 44.85 | 7.28 | 3.64 | 9.09 |
| SPIDER | FastSAM-3D | 1point | 27.53 | 27.48 | 27.63 | 0.15 | 0.07 | 25.61 |
| | | 5points | 38.26 | 37.94 | 38.78 | 0.84 | 0.42 | 26.94 |
| | SAM-Med3D | 1point | 15.37 | 14.41 | 16.93 | 2.52 | 1.26 | 6.80 |
| | | point | 33.10 | 31.57 | 35.57 | 4.00 | 2.00 | 11.03 |
| | SegVol | point | 33.10 | 31.57 | 35.57 | 4.00 | 2.00 | 11.03 |
| | | bbox | 35.15 | 33.38 | 38.01 | 4.63 | 2.31 | 10.60 |

# F Codebase

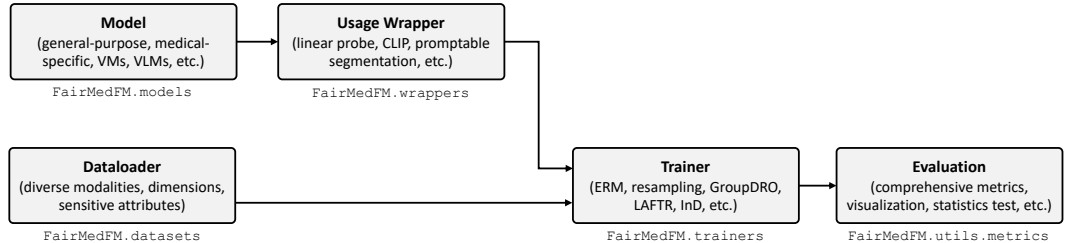

Figure 19: The structure of the open-source `FairMedFM` codebase.

As depicted in Fig. 19, the `FairMedFM` codebase captures comprehensive modules for benchmarking the fairness of foundation models in medical image analysis. We build the codebase using PyTorch. For more details, please refer to our open-sourced repository: https://github.com/FairMedFM/FairMedFM.

1. **Dataloader** provides a consistent interface for loading and processing imaging data across various modalities and dimensions, supporting both classification and segmentation tasks.

2. **Model** is a one-stop library that includes implementations of the most popular pre-trained foundation models for medical image analysis.

3. **Usage Wrapper** encapsulates foundation models for various use cases and tasks, including linear probe, zero-shot inference, PEFT, promptable segmentation, etc.

4. **Trainer** offers a unified workflow for fine-tuning and testing wrapped models, and includes state-of-the-art unfairness mitigation algorithms.

5. **Evaluation** includes a set of metrics and tools to visualize and analyze fairness across different tasks.

We note that all the modules are designed to be easily replicated and extended. The following example demonstrates how to implement the wrapper for CLIP-Adapt with simple modifications.

```python
class CLIPWrapper(BaseWrapper):
    def __init__(self, model, base_text_features):
        super().__init__(model)
        # zero-shot class prototypes
        self.base_text_features = base_text_features
        # class prototypes are trainable in CLIP-Adapt
        self.prototypes = nn.Parameter(base_text_features.clone())

        for param in self.model.parameters():
            param.requires_grad = False

    def forward(self, x):
        return self.model.forward_clip(x, self.prototypes)
```

