# OpenReview forum: "FairMedFM: Fairness Benchmarking for Medical Imaging Foundation Models"
_NeurIPS.cc/2024/Datasets_and_Benchmarks_Track — NeurIPS 2024 Track Datasets and Benchmarks Poster_

### Official Review · Reviewer_XNR2 · 2024-06-19

**Rating:** 8
**Confidence:** 2
**Correctness:** Yes.
**Clarity:** Yes.

**Review:**

In all, the intention and structure of this paper are good. There are some minor issues.

Could you link the equations of equity-scaled AUC and equity-scaled DSC to the supplementary materials? Usually, when people show the trade-off between utility and fairness, they can follow microeconomics to use X axis for fairness and Y axis for utility, and draw a curve to measure the trade-off between fairness and utility. If the curve of model A is above that of model B, it is a better model, as simple as that. The issue of incorporating both fairness and utility to a fixed metric is that, it allows a fairness model to tune some fairness hyperparameter to change that metric. If you are using a fixed metric, anyone can just manipulate that trade-off, making the trade-off metric less meaningful.

Despite the various experiments carried out, detailed results are hidden under the main results in the main manuscript. MEDFAIR, on the other hand, would list all the results with error bars in the tables in the appendix.

Typo in Line 230.

**Strengths:**

Very comprehensive experimental analysis.

**Additional Feedback:**

It would be great to record all the results in tables, so that readers can fully interpret the results by themselves.

**Documentation:**

Yes.

**Ethics:**

No.

**Limitations:**

Yes, limitations are discussed.

**Opportunities For Improvement:**

Some typos need to be addressed.

**Relation To Prior Work:**

Yes.

**Summary And Contributions:**

This paper introduces a fairness benchmark for FM research in medical imaging. It is a comprehensive benchmark covering various model types, functionalities, data types, and evaluation metrics.

---

> ### Author Rebuttal · Authors · 2024-08-17
>
> We are pleased that the reviewer enjoyed reading our paper, and acknowledged our intention and structure. We'd like to provide our response point by point below.
>
> > Review 1: Could you link the equations of equity-scaled AUC and equity-scaled DSC to the supplementary materials?
>
> Thank you for the insight suggestion. We will add the detailed equations in our final version.

---

> > ### Comment · Reviewer_XNR2 · 2024-08-17
> >
> > Thank you for the change. I have no more questions and will retain my good score.

---

> > > ### Author Response · Authors · 2024-08-19
> > > **Thank You**
> > >
> > > Thank you for your feedback and keeping your positive score. Your valuable comments are greatly appreciated!

---

> ### Author Rebuttal · Authors · 2024-08-17
>
> > Review 2: Usually, when people show the trade-off between utility and fairness, they can follow microeconomics to use X axis for fairness and Y axis for utility, and draw a curve to measure the trade-off between fairness and utility.
> ### **Visualization for fairness-utility tradeoff rather than metrics**
>
> We appreciate the reviewer’s insightful feedback. In our original submission, we used fixed equity-scaled metrics (as seen in [1-4]) to evaluate the trade-off between fairness and utility, as it allowed us to present a single, consolidated measure of model performance that was easy to interpret across various scenarios. Our primary goal was to provide a **straightforward** comparison among models, particularly for readers less familiar with the nuances of trade-off curves.
>
> However, we recognize that this approach can oversimplify the relationship between fairness and utility, as you pointed out. After considering your feedback, we agree that visualizing the trade-off in the fairness-utility plane offers a more transparent evaluation. **We have added figures visualizing the fairness-utility plane in the attached PDF**.
>
> In the PDF, we visualize the fairness-utility trade-offs of various foundation models across different datasets, using different tasks or prompts for classification and segmentation tasks. Our results demonstrate that careful selection and use of foundation models are crucial for achieving a favorable fairness-utility trade-off in both tasks. This observation is consistent with the results using fixed equity-scaling metrics, as shown in Figure 4 of our main paper. Additionally, the analysis from lines 236-247 is reflected in these new visualization results.
>
> [1] FairSeg: A Large-Scale Medical Image Segmentation Dataset for Fairness Learning Using Segment Anything Model with Fair Error-Bound Scaling (ICLR 2024)\
> [2] FairVision: Equitable Deep Learning for Eye Disease Screening via Fair Identity Scaling (ArXiv 2024)\
> [3] Harvard Glaucoma Fairness: A Retinal Nerve Disease Dataset for Fairness Learning and Fair Identity Normalization (IEEE TMI 2024)\
> [4] FairDomain: Achieving Fairness in Cross-Domain Medical Image Segmentation and Classification (ECCV 2024)

---

> ### Author Rebuttal · Authors · 2024-08-17
>
> > Review 3: Detailed results are hidden under the main results in the main manuscript.
>
> Thank you for the suggestion. We will add the full result table in the final version of the paper. Besides, we have presented part of the result table in the attachment of the global rebuttal, please refer to the **PDF** for details.

---

### Official Review · Reviewer_EbvA · 2024-07-04
**Review Comments**

**Rating:** 7
**Confidence:** 3
**Correctness:** It is constructed in a sound way in m…
**Clarity:** This paper is well-written.

**Review:**

Quality: The authors have conducted extensive experiments across multiple datasets, models, and tasks, providing a robust foundation for their findings.

Clarity: This work appears to be well-structured and clearly presented. The authors have used figures and tables effectively to summarize complex information and results.

Originality: FairMedFM represents a novel contribution to the field. While fairness in AI has been studied before, this work is unique in its focus on foundation models in medical imaging and its comprehensive scope.

Significance: 1. It addresses a critical issue in healthcare AI - fairness and bias.
2. It provides a benchmark that can guide future research and development of fair AI models in medical imaging.
3. The open-source nature of the project allows for community engagement and further development.

Pros:

1. The scope is comprehensive to cover multiple datasets, models, tasks, and fairness metrics.

2. It provides insights that can directly impact the development and deployment of AI in healthcare.

3. The provided codebase facilitates reproducibility and further research.

4. This work addresses an important ethical concern in AI applications for healthcare.

Cons:

1. The study focuses on binarized sensitive attributes, which may not capture the full complexity of real-world demographics.

2. While these are important tasks, other medical imaging tasks (like detection or reconstruction) are not covered.

3. The findings might be influenced by the specific datasets chosen, which may not be representative of all medical imaging scenarios.

4. The comprehensive nature of the study likely required significant computational resources, which may limit replication or extension by researchers with limited resources.

5. The paper contains some typos, e.g., the heading of Section 3.4, "Taxnonmy" should be "Taxononmy".

**Strengths:**

Contribution Significance: 1. It Introduces a comprehensive benchmark for evaluating fairness in foundation models for medical imaging.
2. It addresses a critical gap in understanding bias and fairness issues in advanced AI models applied to healthcare.
3. It provides insights into the prevalence of bias, fairness-utility trade-offs, and the effectiveness of current mitigation strategies.


Relevance to Broader Research Community: Applicable to both medical imaging and machine learning communities; Open-source codebase promotes reproducibility and enables further research; Findings can guide the development of fairer AI models in medical applications.


Quality of Research: It covers 17 datasets, 20 foundation models, multiple tasks and evaluation metrics.

Ethical and Social Implications: It addresses a crucial ethical concern in AI application to healthcare; It raises awareness about fairness issues in medical imaging AI; It has potential to improve equitable healthcare outcomes by promoting fairer AI models.

**Additional Feedback:**

Please refer to the cons in the section of Review

**Documentation:**

It provides the code for reproducing the experiments, but I did not try the code due to time constraints.

**Ethics:**

I didn't see any potential ethical issues with the submission.

**Limitations:**

It would be better to provide guidelines for responsible use of the benchmark to mitigate potential misuse.

**Opportunities For Improvement:**

Please refer to the cons in the section of Review

**Relation To Prior Work:**

It is clearly discussed how this work differs from previous contributions.

**Summary And Contributions:**

FairMedFM is a comprehensive benchmark introduced to evaluate the fairness of foundation models (FMs) in medical imaging. It encompasses 17 medical imaging datasets, 20 FMs, various tasks including classification and segmentation, and multiple fairness evaluation metrics. The study reveals prevalent bias in FMs for medical imaging tasks, with fairness-utility trade-offs influenced by both FM choice and usage. It identifies consistent disparities in sensitive attributes across different FMs on the same dataset and finds that existing bias mitigation strategies are not always effective for FMs in medical imaging. The authors provide an open-source codebase to facilitate further research, aiming to raise awareness of fairness issues in medical imaging and promote the development of fair algorithms in the machine learning community.

---

> ### Author Rebuttal · Authors · 2024-08-17
>
> We appreciate the reviewer's positive comment on our paper, which presents a comprehensive scope covering multiple datasets, models, tasks, and fairness metrics, provides insights that can directly impact the development and deployment of AI in healthcare, facilitates reproducibility and further research through the provided codebase, and addresses an important ethical concern in AI applications for healthcare.
>
> >Cons 1: The study focuses on binarized sensitive attributes, which may not capture the full complexity of real-world demographics.
>
> ### **Justification on the Selection of Sensitive Attributes**
>
> We appreciate the opportunity to highlight that **we actually included non-binary sensitive attributes** like BMI for the KiTS dataset and ethnicity, language, and marital status for the FairSeg dataset (see Fig. 8, Appendix B). In most of our tasks, our use of binarized sensitive attributes is **largely due to the limitations of most publicly available medical datasets**, which primarily provide binary sex attributes [1]. This constraint has influenced our ability to fully assess fairness across a broader range of non-binary sensitive attributes. Despite this, FairMedFM supports other categorical or non-binary attributes.
>
> Moreover, FairMedFM’s codebase is designed to facilitate this extension by allowing modifications to handle multiple sensitive attributes in the dataloader (see FairMedFM/dataset/base.py). As new datasets are integrated, we are committed to expanding fairness evaluations in medical imaging, encompassing various task types and sensitive attributes, in future work.
>
> [1] Addressing Fairness Issues in Deep Learning-Based Medical Image Analysis: A Systematic Review

---

> ### Author Rebuttal · Authors · 2024-08-17
>
> >Cons 2: While these are important tasks, other medical imaging tasks (like detection or reconstruction) are not covered
>
> ### **Justification on the Current Focus of Medical Imaging Tasks**
>
> We appreciate the reviewer’s comment and would like to first highlight that, in the field of deep learning-based medical imaging analysis, there is a greater body of research focused on classification and segmentation compared to detection or reconstruction tasks [1-2].
>
> Additionally, our primary aim in this work is to benchmark fairness in medical imaging foundation models, particularly within the tasks most commonly studied in the current literature. To the best of our knowledge, there are few open-source foundation models that emphasize medical detection [3-5] or reconstruction, limiting our ability to evaluate fairness in these areas.
>
> For clarification, we interpret "detection" in this context as identifying the location of an object (e.g., using bounding boxes or coordinates), while tasks that involve detecting pathologies based on categorical variables are treated as classification tasks in our study (as seen in [6-7]).
>
> Additionally, there are merely studies about fairness in medical detection/reconstruction tasks, except for [8][9]. Therefore, there is a lack of consensus about the definition of fairness in these two tasks. In our point of view, it is better to decide the evaluation metrics before benchmarking them considering FM.
>
>
> [1] A Survey on Deep Learning in Medical Image Analysis. (Med. Image Anal. 2017)\
> [2] Recent advances and clinical applications of deep learning in medical image analysis. (Med. Image Anal. 2022)\
> [3] LVM-Med: Learning Large-Scale Self-Supervised Vision Models for Medical Imaging via Second-order Graph Matching (NeurIPS 2023)\
> [4] Foundation Model for Endoscopy Video Analysis via Large-scale Self-supervised Pre-train (MICCAI 2023)\
> [5] Contrastive self-supervised learning from 100 million medical images with optional supervision (J. Med. Imaging 2022)\
> [6] A foundation model for generalizable disease detection from retinal images. (Nature 2024)\
> [7] CheXpert: A Large Chest Radiograph Dataset with Uncertainty Labels and Expert Comparison. (AAAI 2019)\
> [8] Unveiling Fairness Biases in Deep  Learning-Based Brain MRI Reconstruction. (MICCAI 2023 workshop)\
> [9] Bias in Unsupervised Anomaly Detection in  Brain MRI. (MICCAI 2023 workshop)

---

> ### Author Rebuttal · Authors · 2024-08-17
>
> >Cons 3: The findings might be influenced by the specific datasets chosen, which may not be representative of all medical imaging scenarios.
>
> ### **Inclusion of Representative dataset**
> We appreciate the reviewer's insights regarding the representativeness of the datasets in our study. We have made significant efforts to include a wide range of open-source medical datasets that contain sensitive attribute information. These datasets include **large-scale X-ray datasets and those frequently utilized to train medical foundation models**, such as MIMIC and CheXpert for MedCLIP, MedMAE, PubMedCLIP, and C2L. Although these datasets may not fully encompass the diversity of patient populations, **they are among the most comprehensive available**. Additionally, we have incorporated recently proposed **datasets specifically designed to investigate medical fairness**, including GF3300 and FairVLMed10k. These datasets are widely used in various medical image analysis studies focused on fairness.
>
> Moreover, a key strength of FairMedFM is its extendable nature, allowing it to evolve alongside advancements in foundation models. In this rapidly evolving field, both datasets and models are continually improving. We have designed FairMedFM to accommodate new datasets as they become available.
> We are dedicated to the continuous improvement of our work and would greatly appreciate any specific dataset recommendations the reviewer may have. We are keen to incorporate additional datasets into our package to further enhance its comprehensiveness and relevance as our continuous efforts.

---

> ### Author Rebuttal · Authors · 2024-08-17
>
> >Cons 4: The comprehensive nature of the study likely required significant computational resources, which may limit replication or extension by researchers with limited resources.
>
> Thank you for acknowledging the comprehensive nature of our study and the significant effort that went into building this benchmark. However, we would like to clarify that the computational resources required to replicate or extend our work are far more accessible than one might assume.
>
> While FairMedFM indeed covers a broad range of tasks, researchers can use our provided codebase to perform evaluation using pre-trained foundation models or efficient tuning methods, without the need for extensive retraining. For example, all of our segmentation experiments were carried out using official pre-trained weights, eliminating the need for large-scale computational resources. Additionally, most of our classification tasks employed parameter-efficient fine-tuning techniques such as Linear Probe, CLIP-adapter, and LoRA. These methods are designed to minimize the computational burden and can be easily executed on a single NVIDIA RTX 3080 GPU. This setup strikes a balance between thorough experimentation and accessibility, ensuring that researchers with limited resources can replicate, deploy, and extend our work.
>
> Moreover, to further support the community, we will be including our raw experimental results in the final version of the paper. This will allow researchers working with similar configurations to directly reference our findings without having to repeat the entire computational process themselves.
>
> In conclusion, we are confident that FairMedFM is not only comprehensive but also scalable and accessible, empowering a wide range of researchers to contribute to this important field.
>
> >Typos
>
> We thank the reviewer for pointing out our typos. We will make corresponding revisions in the final version of the paper.
>
> >Guidelines for responsible use of the benchmark
>
> We appreciate the reviewer's suggestion. I have updated our codebase with a guideline for the responsible use of the benchmark under the acknowledgment section.

---

### Official Review · Reviewer_ppic · 2024-07-17
**brief**

**Rating:** 4
**Confidence:** 5
**Correctness:** Yes
**Clarity:** yes

**Review:**

Clarity: The paper is well-written.
Originality: The development of FairMedFM is an original contribution, providing a new tool for evaluating fairness in medical imaging FMs.
Significance: The findings might have implications for improving the fairness of medical imaging tools.

**Strengths:**

1.	The paper introduces FairMedFM, a benchmark for assessing fairness in medical imaging Foundation Models. This addresses a critical gap in the field, enhancing the understanding and development of equitable AI in healthcare.
2.	The framework's extensive evaluation across multiple datasets, FMs, and tasks provides a robust and thorough analysis, demonstrating the depth and breadth of the research.

**Additional Feedback:**

Please see “Opportunities For Improvement”.

**Documentation:**

Yes, this paper discusses the collection of the dataset and publishes the code, which enhances its usability.

**Limitations:**

Please see “Opportunities For Improvement”.

**Opportunities For Improvement:**

1.	The datasets included in the study may not fully represent the global diversity in patient populations, which might potentially limit the generalizability of the findings.
2.	The framework's relevance might be constrained by its current focus on specific medical imaging tasks and may have difficulty to adopt to other tasks or domains.
3.	The scalability of the framework to accommodate growing data sizes and complexities in medical imaging could be a limitation that warrants further investigation.

**Relation To Prior Work:**

Not sufficient.

**Summary And Contributions:**

This paper presents FairMedFM, a novel benchmarking framework designed to evaluate the fairness of Foundation Models (FMs) in medical imaging. The paper addresses a significant gap in the literature regarding fairness in AI applications within healthcare. The authors have provided a comprehensive analysis and an open-source tool that could greatly benefit the research community.

---

> ### Author Rebuttal · Authors · 2024-08-17
>
> We appreciate the reviewer’s feedbacks and suggestios on our paper. We provide our response point-to-point below in the following rebuttal blocks.

---

> ### Author Rebuttal · Authors · 2024-08-17
>
> >Improvement 1: The datasets included in the study may not fully represent the global diversity in patient populations, which might potentially limit the generalizability of the findings.
>
> ### **Explanation of the dataset’s diversity**
> We fully acknowledge the importance of ensuring that patient populations are appropriately diverse, and this consideration was a core motivation in designing our benchmark that includes extensive public medical datasets and an evaluation pipeline that can be easily adapted to other datasets.
>
> **We have taken significant steps to include a wide range of datasets**—17 in total—spanning different institutions and patient demographics. These datasets, such as the **large-scale MIMIC dataset**, include sensitive attribute information to ensure a broad representation. Additionally, we have incorporated datasets like GF3300 and FairVLMed10k, which have been **specifically developed to investigate fairness in medical image analysis** and are widely used in fairness-related studies.
>
> That said, it is important to recognize that our work **represents one of the first systematic efforts to evaluate the fairness of using foundation models for medical imaging tasks**. Prior to this study, there had been no comprehensive evaluation of fairness in this domain. We believe **our study represents a pioneering effort to raise awareness about fairness in medical imaging tasks in the era of foundation models and could inspire more collaborative efforts in the community to evaluate more diverse datasets**.
>
>
> ### **Generalizability of the findings**
>
> Regarding reviewers' concern about the potential limitations in the generalizability of our findings, while we acknowledge the inherent challenges in achieving global diversity and complete generalizability, the datasets we utilized represent a meaningful step toward addressing this gap. **Our findings are consistently robust across the 17 datasets and 20 foundation models** evaluated (e.g., Figures 2 and 3). We agree that there may be differences when evaluating models on private, inaccessible datasets. However, **the primary goal of FairMedFM is to provide the community with an extensible tool for investigating fairness in foundation models**, with the flexibility to accommodate future advancements in dataset size and diversity.
>
> We are fully committed to enhancing the comprehensiveness of our study and welcome any suggestions for additional publicly accessible datasets that the reviewer may recommend. If such datasets are available, we would be glad to integrate them into our evaluation framework to further strengthen the generalizability and applicability of our findings.

---

> ### Author Rebuttal · Authors · 2024-08-17
>
> > Improvement 2: The framework's relevance might be constrained by its current focus on specific medical imaging tasks and may have difficulty to adopt to other tasks or domains.
>
> ### **The focus of the work**
>
> We thank the reviewer for their thoughtful comments regarding the scope of our framework. As stated in both the abstract (line 9) and introduction (line 39) of our paper, our primary focus is on addressing the fairness of foundation models (FMs) specifically within medical image analysis and the tasks of classification and segmentation (line 14 in abstract and Table 1 in introduction). These two downstream tasks consist of the greater body of deep-learning-based medical image analysis, especially in the field of health equity [1-2].  This focus is intentional, given the rapid proliferation of FMs in both tasks in medical imaging and the critical need to systematically evaluate their implications for health equity in clinical settings.
>
> While we believe FairMedFM has the potential to be adapted for other medical imaging tasks, we did not include other tasks such as detection or reconstruction in this study. This decision was primarily due to **the current lack of pre-trained foundation models and large-scale datasets with sensitive attributes in these areas**. Without these essential components, it is challenging to conduct a thorough benchmark and draw generalizable conclusions about fairness issues in these additional tasks.
>
> Extending the scope beyond medical machine learning to encompass other domains would **risk diluting the specificity and relevance of our study**. By concentrating on medical imaging, particularly in tasks like classification and segmentation—both central to the current fairness literature—we are able to offer deeper, more actionable insights that directly address the unique challenges in this area.
>
> ### **Justification of the feasibility of adopting to other tasks or domains**
>
> Conversely, a key strength of FairMedFM is its **extendable design, enabling it to grow with advancements in foundation models**. In this fast-paced field, datasets and models are constantly changing. FairMedFM supports new tasks, models, or data from various domains or modalities through easily extensible modules such as Usage Wrapper and Dataloaders. For instance, section F in the appendix details FairMedFM's structure and includes an example of integrating CLIP-based model wrappers for zero-shot classification and adaptation in just a few lines of code.
>
> [1] A Survey on Deep Learning in Medical Image Analysis. (Med. Image Anal. 2017)\
> [2] Recent advances and clinical applications of deep learning in medical image analysis. (Med. Image Anal. 2022)

---

> ### Author Rebuttal · Authors · 2024-08-17
>
> > Improvement 3: The scalability of the framework to accommodate growing data sizes and complexities in medical imaging could be a limitation that warrants further investigation.'
>
> ### **Clarification on the scalability of the framework**
>
> We humbly disagree with the concern about “The scalability of the framework to accommodate growing data sizes and complexities in medical imaging could be a limitation”. FairMedFM has already integrated large-scale datasets, such as CheXpert and MIMIC, which are commonly used to train foundation models like MedCLIP and MedMAE. To the best of our knowledge, these datasets represent the current standard for large-scale medical imaging data used in fairness studies.
>
> Again, the core contribution of FairMedFM is its extendibility, particularly through our dataloader extension. We have designed the framework to be adaptable, anticipating future increases in data size and complexity. Users can seamlessly incorporate new datasets by following a straightforward process:
>
> 1. Define the disease labels;
>
> 2. Define the sensitive attributes, and;
>
> 3. Specify the method to retrieve the images.
>
> This flexibility ensures that FairMedFM remains scalable and relevant as new datasets emerge and the field of medical imaging evolves. Should the reviewer have specific suggestions or additional concerns regarding scalability, we are open to further refining our framework to better address these challenges.

---

> ### Author Rebuttal · Authors · 2024-08-23
>
> Dear Reviewer ppic,
>
>
> Thank you for dedicating your valuable time to reviewing our paper and our responses. We have carefully reviewed and thoughtfully addressed each of your comments in detail. We are encouraged by your the overall positive feedback from the reviewers. As the discussion period comes to a close, we would greatly appreciate your feedback on whether our responses satisfactorily address your questions and concerns.
>
>
> Best regards,
>
> FairMedFM Authors

---

> > ### Comment · Reviewer_ppic · 2024-09-01
> >
> > Thanks for the careful rebuttal, which solves some of my previous concerns, but major limitation remains. Thus, I will keep my score unchanged.

---

> > > ### Author Rebuttal · Authors · 2024-09-01
> > >
> > > Thank you for your feedback. We genuinely regret that some concerns remain unresolved, especially as we only became aware of them at the very end of the discussion period. If you could kindly specify the particular questions or issues still outstanding, we would be more than willing to address them. While the timing has limited our opportunity to respond comprehensively, we are committed to doing our best to resolve any remaining concerns.

---

### Official Review · Reviewer_Lo9r · 2024-07-20
**Fairness Benchmark of Foundational models on Medical Data**

**Rating:** 10
**Confidence:** 4
**Correctness:** Yes

**Review:**

The authors address a crucial problem within the medical image analysis community, garnering widespread interest. Their proposed solution aims to streamline researchers' analysis processes, ultimately facilitating more efficient and effective work. The authors take care to thoroughly outline the importance of the problem at hand, providing a comprehensive explanation as well as a detailed description of the benchmark setup. Notably, the benchmark encompasses 17 medical datasets and 20 FMs, showcasing the authors' commitment to comprehensiveness and thoroughness. The benchmark provides an extensive amount of tools to gauge the fairness of the models, making it possible for the researchers to take a look at different aspects of Fairness.

Besides introducing the benchmark setup and its content, the authors also present some exemplary results on the application of the benchmark on fairness in gender. The benchmark also contains some mitigation strategies and metrics on their success.

Overall, the presented benchmark is thorough regarding data, models, and metrics, and it is open for further development.

**Strengths:**

The benchmark is extensive in terms of provided datasets, models and metrics. The paper is well written and easy to follow to help the reader understand the different aspects of the provided tools.
The authors provide detailed information on datasets and models in the supplementary materials. They also provide additional test results in the supplementary.
Overall, the paper is extensive, and the benchmark provided will greatly help the research community.

**Additional Feedback:**

none

**Clarity:**

The presentation is clear. There are some minor typos in the text and the figure, that does not effect the understandability of the paper in any means.

**Documentation:**

yes

**Limitations:**

None to highlight

**Opportunities For Improvement:**

There are a few typos in the paper and the figures.

**Relation To Prior Work:**

yes

**Summary And Contributions:**

The authors present a benchmark setup consisting of 17 medical image datasets and 20 foundational models. The benchmark consists of several adaptation/fine-tuning techniques and, more importantly, fairness metrics to gauge the fairness of the models over different datasets.

The authors present their findings on the fairness of the models on the "gender" attribute (and more on the supplementary data), but more importantly, they provide the means for the users to conduct their preferred test on their selected attributes and models.

---

> ### Author Rebuttal · Authors · 2024-08-17
>
> Thank you for your positive recognition and comments.
>
> We are glad to hear that the reviewer enjoyed reading our paper. Also, we appreciate you for pointing us to the minor typos in the paper and figure. We will make sure to correct them and fully polish our final version of the paper.

---

### Author Rebuttal · Authors · 2024-08-17

We appreciate that the reviewers are aware of the importance (Lo9r, EbvA), comprehensiveness (Lo9r, ppic, EbvA, XNR2), novelty (ppic, EbvA), usefulness (Lo9r, ppic, EbvA) and well-written (Lo9r, ppic, EbvA, XNR2) of our FairMedFM.

We have corrected the typos in the revised manuscript and added the full result table to the Appendix. We also provide part of the full result table (classification, *sex* as the sensitive attribute) in the **attached PDF**.

---

### Decision · Program_Chairs · 2024-09-26

**Decision:**

Accept (Poster)

**Comment:**

This paper introduces a new benchmark for assessing fairness in medical imagining foundation models. The benchmark is extensive in terms of datasets, models, and metrics. The paper is well-written and easy to follow. All reviewers agree that this study addresses a crucial ethical problem in AI applications in healthcare, which will benefit the medical image analysis and machine learning communities. It also provides insights into the prevalence of bias, fairness trade-offs, and the effectiveness of current mitigation strategies.

Reviewers also raise concerns such as the complexity of demographics and scalability of the dataset. The authors' response has clarified most of the points raised by the reviewers. In light of this, the authors are strongly encouraged to incorporate the feedback into the final version.